# Process-based modelling of NH₃ exchange with grazed grasslands

Andrea Móring[1,2,3], Massimo Vieno[2], Ruth M. Doherty[3], Celia Milford[4,5], Eiko Nemitz[2], Marsailidh M. Twigg[2], László Horváth[6], Mark A. Sutton[2]

[1]University of Edinburgh, High School Yards, Edinburgh, EH8 9XP, United Kingdom
[2]Centre for Ecology & Hydrology, Bush Estate, Penicuik, EH26 0QB, United Kingdom
[3]University of Edinburgh, The King's Buildings, Alexander Crum Brown Road, Edinburgh, EH9 3FF, United Kingdom
[4]Associate Unit CSIC University of Huelva "Atmospheric Pollution", CIQSO, University of Huelva, Huelva, E21071, Spain
[5]Izaña Atmospheric Research Center, AEMET, Joint Research Unit to CSIC "Studies on Atmospheric Pollution", Santa Cruz de Tenerife, Spain
[6]Hungarian Meteorological Service, Gilice tér 39, Budapest, 1181, Hungary

*Correspondence to*: Mark A. Sutton (ms@ceh.ac.uk)

**Abstract.** In this study the GAG model, a process-based ammonia (NH₃) emission model for urine patches was extended and applied for the field scale. The new model (GAG_field) was tested over two modelling periods, for which micrometeorological NH₃ flux data were available. Acknowledging uncertainties in the measurements, the model was able to simulate the main features of the observed fluxes. The temporal evolution of the simulated NH₃ exchange flux was found to be dominated by NH₃ emission from the urine patches, offset by simultaneous NH₃ deposition to areas of the field not affected by urine. The simulations show how NH₃ fluxes over a grazed field in a given day can be affected by urine patches deposited several days earlier, linked to the interaction of volatilization processes with soil pH dynamics. Sensitivity analysis showed that GAG_field was more sensitive to soil buffering capacity ($\beta$), field capacity ($\theta_{fc}$) and permanent wilting point ($\theta_{pwp}$) than the patch scale model. The reason for these different sensitivities is dual. Firstly, the difference originates from the different scales. Secondly, the difference can be explained by the different initial soil pH and physical, which determine the maximum volume of urine that can be stored in the NH₃ source layer. It was found that in the case of urine patches with a higher initial soil pH and higher initial soil water content, the sensitivity of NH₃ exchange to $\beta$ was stronger. Also, in the case of a higher initial soil water content, NH₃ exchange was more sensitive to the changes in $\theta_{fc}$ and $\theta_{pwp}$. The sensitivity analysis showed that the nitrogen content of urine ($c_N$) is associated with high uncertainty in the simulated fluxes. However, model experiments based on $c_N$ values randomized from an estimated statistical distribution indicated that this uncertainty is considerably smaller in practice. Finally, GAG_field was tested with a constant soil pH of 7.5. The variation of NH₃ fluxes simulated in this way showed a good agreement with those from the simulations with the original approach, accounting for a dynamically changing soil pH. These results suggest a way for model simplification when GAG_field is applied later for regional scale.

## List of Symbols

| | |
|---|---|
| $A_{non}$ (m²) | Area of the field unaffected by urine (non-urine area) |
| $AD$ (ha⁻¹) | Animal density |
| $A_{field}$ (m²) | Field area |

| | |
|---|---|
| $A_{patch}$ (m$^2$) | Area of a urine patch |
| $B_{H2O}$ (dm$^3$) | Water budget in the source layer |
| $B_{H2O}(max)$ (dm$^3$) | Maximal water amount in the source layer |
| $B_{H2O}^j$ (dm$^3$) | Water budget in the source layer under the urine patches deposited in the $j^{th}$ time step |
| $c_N$ (g N dm$^{-3}$) | N content of the urine (assumed to be in the form of urea) |
| $c_N^{Ave}$ | Average urinary N concentration (assumed to be in the form of urea) in urine patches deposited in the same time step |
| $c_N^{Dil}$ (g N dm$^{-3}$) | Urine N content (assumed to be in the form of urea) after dilution in the soil |
| $c_N^k$ (g N dm$^{-3}$) | Urinary N concentration (assumed to be in the form of urea) in the $k^{th}$ urine patch |
| $d$ (m) | Displacement height |
| $D(c_N)$ | Distribution function of urinary nitrogen content |
| $D_t$ | Proportion of the urine-covered area over a $t$ time period on field if there is no overlap between the urine patches |
| $F_{non}$ (μg N m$^{-2}$ s$^{-1}$) | Net NH$_3$ exchange flux over the non-urine area |
| $F_g$ (μg N m$^{-2}$ s$^{-1}$) | NH$_3$ exchange flux over the ground |
| $F_{net}$ (μg N m$^{-2}$ s$^{-1}$) | Net NH$_3$ exchange flux for the whole field |
| $F_{patch}^j$ (μg N m$^{-2}$ s$^{-1}$) | NH$_3$ emission flux from the urine patches deposited in the $j^{th}$ time step |
| $F_t$ (μg N m$^{-2}$ s$^{-1}$) | Total NH$_3$ exchange flux over the canopy above a single urine patch |
| $F_\chi$ (μg N m$^{-2}$ s$^{-1}$) | NH$_3$ exchange flux derived based on measurements with AMANDA |
| $h$ (m) | Canopy height |
| $H$ (J m$^{-2}$ s$^{-1}$) | Sensible heat flux |
| $K$ | Karman constant |
| $K$ | Parameter representing the uniformity of the excretal distribution on a field |
| $L$ (m) | Monin-Obukhov length |
| $LAI$ (m$^2$ m$^{-2}$) | Leaf area index |
| $n(t_j)$ | Number of urine patches deposited in the $j^{th}$ time step |
| $N_t$ | Total number of urine patches deposited over a $t$ time period on a field |
| $p$ (kPa) | Surface atmospheric pressure |
| $pH(t_0)$ | Soil pH before urine patch deposition |
| $P$ (mm) | Precipitation amount |
| $PAR$ (μmol m$^2$ s$^{-1}$) | Photosynthetically active radiation |
| $P_t$ | Proportion of the field covered by urine patches after a $t$ time period |
| $Q$ | Parameter in the calculation of $P_t$ |
| $R_{ac}$ (s m$^{-1}$) | Aerodynamic resistance in the canopy |
| $R_g$ (s m$^{-1}$) | Resistance on the ground |
| $REW$ (mm) | Readily evaporable water in the soil |
| $R_{glob}$ (MJ m$^2$h$^{-1}$) | Global radiation / solar radiation |
| $RH$ (%) | Relative humidity |

| | |
|---|---|
| $Sens_{net}$ (%) | Sensitivity of the total $NH_3$ exchange over the whole field |
| $Sens_{patch}$ (%) | Sensitivity of the total $NH_3$ exchange over the urine patches on the field |
| $t_i, t_j$ | $i^{th}$ and $j^{th}$ time steps |
| $T_{air}$ (°C) | Air temperature at 2 m |
| $T_{soil}$ (°C) | Soil temperature |
| $u$ (m s$^{-1}$) | Wind speed |
| $u_{dir}$ (°) | Angle of the wind direction |
| $u_*$ (m s$^{-1}$) | Friction velocity |
| $U_{add}$ (g N) | Urea added to the source layer |
| $UF$ (animal$^{-1}$ day$^{-1}$) | Urination frequency |
| $W_{rain}$ (dm$^3$) | Water input as rain water over the urine patch |
| $W_{urine}$ (dm$^3$) | Volume of urine |
| $z$ (m) | Height of the $NH_3$ concentration measurements |
| $z_w$ (m) | Height of wind measurement |
| $\beta$ (mol H$^+$ (pH unit)$^{-1}$ dm$^{-3}$) | Soil buffering capacity |
| $\beta_{patch}$ (mol H$^+$ (pH unit)$^{-1}$) | Buffering capacity of the source layer |
| $\Gamma_g$ | $NH_3$ emission potential on the soil surface |
| $\Gamma_{sto}$ | $NH_3$ emission potential from the stomata |
| $\Delta z$ (mm) | Thickness of the source layer |
| $\theta(t_0)$ (m$^3$ m$^{-3}$) | Soil volumetric water content before urine patch deposition |
| $\theta_{fc}$ (m$^3$ m$^{-3}$) | Field capacity |
| $\theta_{por}$ (m$^3$ m$^{-3}$) | Porosity |
| $\theta_{pwp}$ (m$^3$ m$^{-3}$) | Permanent wilting point |
| $\theta_{urine}$ (m$^3$ m$^{-3}$) | Proportion of the source layer that can be filled up by urine |
| $\sigma, \mu$ | Scale parameters of the log-normal distribution (the arithmetic standard deviation and the arithmetic mean of the normal distribution of $\log(c_N)$, respectively) |
| $\Sigma F_{net}$ | Total $NH_3$ exchange over the grazed field |
| $\Sigma F_{non}$ | Total $NH_3$ exchange over the non-urine area |
| $\Sigma F_{patch}$ | Total $NH_3$ emission from the urine patches on a grazed field |
| $X$ | Air concentration of $NH_3$ in the measurement heights of AMANDA |
| $\chi_a$ (µg N m$^{-3}$) | Air concentration of $NH_3$ at 1 m height |
| $\chi_g$ (µg N m$^{-3}$) | Compensation point on the ground |
| $\chi_p$ (µg N m$^{-3}$) | Compensation point in the soil pores |
| $\chi_{z0}$ (µg N m$^{-3}$) | Canopy compensation point |
| $\Psi_H$ | Stability function for heat |

# 1 Introduction

The global nitrogen (N) cycle has been substantially altered by the emission of reactive nitrogen compounds ($N_r$), which is dominated by the emission of ammonia ($NH_3$) (Galloway et al., 2008, Fowler et al., 2013). As a result of the strong emission of $N_r$, five key environmental threats have been identified: water, air and soil quality, greenhouse balance and ecosystems (Sutton et al., 2011). The main global source of $NH_3$ emission to the atmosphere is agriculture (EDGAR, 2011), specifically, the breakdown of animal excreta and fertilizers containing ammonium ($NH_4^+$). The volatilization of $NH_3$ is dependent on meteorology, especially temperature (Flechard et al., 2013, Sutton et al., 2013), which raises the question: how will $NH_3$ emission be influenced by climate change? A way to address this question and predict the environmental consequences is to design meteorology-driven $NH_3$ emission models for each agricultural source (Sutton et al., 2013). This study represents a step

toward this goal by describing an $NH_3$ exchange model for grazed fields, accounting for the relevant meteorological drivers.

As confirmed by both laboratory and field studies (Farquhar et al., 1980, Sutton et al., 1995), the exchange of $NH_3$ between surface and atmosphere is bidirectional. The direction of the net $NH_3$ exchange is controlled by the difference in the relative magnitude of atmospheric $NH_3$ concentration at two heights above the surface: the so-called 'compensation point' (atmospheric $NH_3$ concentration right above the surface) and the ambient atmospheric $NH_3$ concentration (high above the

surface). If the compensation point is the larger of the two $NH_3$ is emitted to the atmosphere, whilst if the ambient air concentration is the larger, net deposition takes place, transferring $NH_3$ to the surface. The state-of-the-art modelling technique for this bidirectional behaviour is the application of a 'canopy compensation point' model (Sutton et al., 1995, Nemitz et al., 2001, Burkhardt et al., 2009, Flechard et al., 2013). These models derive the net $NH_3$ emission flux over a canopy by taking into account the $NH_3$ exchange with the different sources and sinks within the canopy (e.g. stomata, leaf surface, soil, litter,

etc.) as well as the effect of meteorological variables and the canopy on these component $NH_3$ fluxes.

Over a grazed field the dominant source of $NH_3$ is urine rather than dung (Petersen et al., 1998, Laubach et al., 2013). Therefore, the $NH_3$ exchange over a grazed field is determined by two main components: the $NH_3$ emission from the urine patches and the $NH_3$ exchange with the area on the field that is not affected by urine ("non-urine area"). The GAG model (Generation of Ammonia from Grazing, Móring et al., 2016) is a special application of a canopy compensation point model that derives $NH_3$

volatilization from a unit of $NH_3$ source on a grazed field: a single urine patch. GAG calculates $NH_3$ emission from a urine patch in a process-based way, simulating the total ammoniacal nitrogen (TAN) and water content under the urine patch as well as the evolution of soil pH. The present paper describes an extension of the GAG model, so that it accounts for the $NH_3$ emission from all of the urine patches deposited over a time interval on a grazed field and the $NH_3$ exchange with the non-urine area.

The primary goal of this model development was to construct a tool that can be used in further studies to gain insights on the effects of meteorological variables on $NH_3$ emission from grazing. Furthermore, our aim was to design a model that can be applied to an atmospheric chemistry transport model. Such a model application would serve as a base for future research, investigating how altered climate can affect $NH_3$ emission, dispersion and deposition on a larger scale, i.e., regional or global

scale. Therefore, simplicity was a key aspect in the model development presented here, while taking into account physical and chemical processes that can be relevant over these larger scales.

In the following, firstly, the theoretical background of the field-scale model application is presented (Section 2). Secondly, the equations required for upscaling to a field are provided, as well as the data used in the model evaluation and the methods applied in the sensitivity analysis are introduced (Section 3). This is followed by presentation of the model simulations for two experimental periods and the outcomes of the sensitivity analysis (Section 4). Finally, we conclude the paper with the discussion of the results and our conclusions (Section 5 and 6).

## 2 Theoretical background

### 2.1 Description of the GAG model

The GAG model, applied and extended to the field scale in this study, is a process-based $NH_3$ emission model for a single urine patch. An in-depth description of the model, together with a comprehensive sensitivity analysis can be found in Móring et. al (2016) and Móring (2016). The GAG model is capable of simulating the driving soil chemistry by accounting for: the TAN and the water content of the soil under the urine patch (in Fig. 1. TAN budget and water budget, respectively) and the variation of soil pH ($H^+$ ion budget in Fig. 1). Following the considerations of Móring et al. (2016), the model handles urine as a water solution of urea, i.e. the urinary N content is assumed to be in the form of urea. The TAN and the $H^+$ ion budgets are controlled by the hydrolysis of the urea content of urine, as well as the $NH_3$ emission from the soil. In the water budget, apart from the liquid content of the incoming urine, precipitation acts as a source term, whilst soil evaporation is considered as the only sink term.

The GAG_model is a single layer model, which means that the effective $NH_3$ emission occurs only from the urine that a thin top soil layer, the so-called "ammonia source layer" can hold. Since during the development of the GAG model simplicity was a key aspect, the effect of the vertical movement of the liquid within the soil (leaching and capillary rise) as well as the mixing of urea and the products of its hydrolysis within the solution was neglected. Hereafter, the original GAG model for patch scale and its extended version for field scale are referred to as GAG_patch and GAG_field, respectively.

### 2.2 Assumptions for the model application at field scale

Among all the naturally varying factors related to urination events during grazing, the following subsections describe those that are likely to be the most relevant from the point of view of $NH_3$ exchange over a grazed field. Firstly, the possible overlap of the patches is examined (Section 2.2.1), then further parameters are discussed that can vary among urination events, such as the area of the patches ($A_{patch}$), the frequency of urination events ($UF$) and the nitrogen content of urine ($c_N$) (Section 2.2.2). Finally, model assumptions for calculating the total $NH_3$ net flux for the field are identified (Section 2.2.3).

### 2.2.1 Exclusion of the overlap of the urine patches

According to observations (e.g. Betteridge et al., 2010, Moir et al., 2011, Dennis et al., 2013), urine patches over a grazed paddock may overlap. It was found that the overlap can have a large effect on N leaching (Pleasants et al., 2007, Shorten and Pleasants, 2007); however, no studies are available that investigate the effect of overlap in particular on $NH_3$ emission from urine patches.

It is reasonable to assume that the emission flux from the area of the overlap will differ from both the previously and the newly deposited patches due to the differences in the soil chemical properties (Fig. 2). Since urea hydrolysis is in a different stage in the two urine patches, the soil chemistry under them will be different, and their mixture under the overlap is likely to result in a third, different chemical composition. In addition, if patches partly cover each other, the total source area will be smaller than if they were completely separate, which may influence the total $NH_3$ emission from the field. Therefore, it is likely that the possible overlap of the patches affects $NH_3$ emission. However, to predict in every time step of the model which patches will cover each other, and what size the overlap will be, is very difficult. Thus, it would be preferable to neglect the overlap of the patches if the error from this simplification can be shown to be small. To assess the resulting error arising from such a simplification, the difference in the field proportion covered by urine patches was investigated between the two cases: when overlap is assumed and when it is excluded.

A way to estimate the temporal evolution of the urine-covered proportion of the field is to use a negative binomial distribution function for the time-space distribution of the urine patches as suggested by Petersen et al. (1956), or the Poisson distribution tested by Romera et al. (2012). Based on the distribution suggested by Petersen et al. (1956), Pakrou and Dillon (2004) determined the proportion of the paddock covered by urine patches ($P_t$) after a $t$ time period as:

$$P_t = 1 - q^{-K}, \tag{1}$$

where $K$ is a parameter that represents the uniformity of the excretal distribution. Following Pakrou and Dillon (2004), a representative value of K=7 was used. The value of $q$ is calculated as:

$$q = \frac{D_t + K}{K}, \tag{2}$$

in which $D_t$ is the proportion of the urine-covered area over a $t$ time period if there is no overlap (Eq. 3), i.e. the total number of the patches ($N_t$) deposited over $t$ multiplied by $A_{patch}$ and divided by the field area ($A_{field}$).

$$D_t = \frac{N_t A_{patch}}{A_{field}} \tag{3}$$

Using $D_t$, Romera et al. (2012) derived $P_t$ assuming a Poisson distribution as follows:

$$P_t = 1 - e^{-D_t}, \tag{4}$$

where $e$ is Euler's constant (~2.718).

To investigate the highest possible difference that the exclusion of overlap can cause, in the following calculation a "worst case scenario" was assumed with the highest possible coverage by urine, i.e. the highest realistic animal density over a field,

the largest $A_{patch}$ and the highest $UF$. The ranges of all these parameters are listed in Table 1 for sheep and cattle, together with their references.

According to the agricultural statistics of the European Commission for 2010 (EC, 2015), the maximal grazing animal densities on the agricultural holdings Europe-wide were higher than 10 LSU ha$^{-1}$ (where LSU stands for livestock unit, which equals to 1 dairy cow or 10 sheep). Since no higher values than 10 were identified, 10 LSU ha$^{-1}$ was assumed as the maximum. The value of $N_t$ was calculated as the product of animal density over a hectare ($A_{field}$ = 10 000 m$^2$) and the maximum daily $UF$ (urination events per animal per day, Table 1).

Fig. 3 shows $P_t$, using the two different equations, Eq. (1) and (4). These results are very close to each other, with slightly smaller values from Eq. (1). Therefore, for further investigation the $P_t$ values from Pakrou and Dillon (2004) (Eq. 1) were taken and compared with the no overlap case ($P_t = D_t$). In the case of sheep (Fig. 3a), the difference between $P_t$ and $D_t$ became higher than 5% after the eighth day (and exceeds 10% after the 16$^{th}$ day – not shown here), whilst in the case of cattle (Fig. 3b) the same occurred after the 17$^{th}$ day.

The great majority of NH$_3$ is emitted in the first 8 days after the deposition of a urine patch (Sherlock and Goh, 1985). This means that after the eighth day the NH$_3$ exchange flux over the urine patches will be very close to that of the unaffected area of the field. Presumably, (as suggested by the model results in Móring et al. 2016) at this stage the chemical composition of the soil solution in the source layer under these patches will be also close to that of the initial, unaffected soil. Thus, practically, the patches deposited eight or more days before the given time step can be treated as part of the unaffected area of the field, or in other words, these patches disappear from the field. As a consequence, the total area of the patches grows in the first eight days, then it remains constant while the animals are on the field. Therefore, the probability of overlap after the eighth day will be the same as on the eighth day, since the total area of the patches prone to overlap with the new patches does not change after the eighth day.

Finally, it has to be noted that the results in Fig. 3 illustrate an extreme situation (the "worst case scenario"), and in reality $P_t$ is much likely to grow rather more slowly. This allows a longer time before the exceedance of the 5% difference in $P_t$ between the overlap and no-overlap case. Hence, for field-scale application of GAG the effect of overlap between the patches was concluded to be negligible, assuming completely separated urine patches in every time step.

It should be stressed that in the above calculation the case of rotational or intensive grazing was not taken into account when the grazing density can be above 20-40 LSU/ha (e.g. Bell et al., 2016), whilst the animals are typically on the field only for only a few days. If it is assumed that an intensive grazing period typically lasts for a maximum of 3 days, using Eq. 1 and 3, with the maximum $A_{patch}$ values from Table 1, in case of cows, 57 and 113 LSU/ha can be on the field to keep the error – originating from the neglect of the overlap between the urine patches - under 5% and 10%, respectively. In case of sheep the same numbers will be 26.1 and 51.7 LSU/ha, respectively. For cows, the resulting grazing densities are above the 40 LSU/ha, therefore, even in the worst case, the error will be under 5%. For sheep, calculating with 44 LSU/ha, the highest grazing density in Bell et al. (2016), the error will be 8%, which can be still considered reasonable for the worst case scenario. While patch

overlap can therefore be generally neglected for continuous grazing and short periods of rotational grazing, we acknowledge that there may be some extreme cases of intense extended grazing where patch overlap could become relevant.

### 2.2.2 Assumptions for $A_{ptach}$, UF and $c_N$

As shown in the previous subsection, the parameters that regulate the extent of the field covered by urine are (i) the number of the animals on the field, (ii) $A_{patch}$ and (iii) $UF$. The first parameter at a field-scale model application is easy to obtain, but the observations of the area of every single urine patch, as well as the number of urinations on an hourly basis, are rather difficult (see the overview of the observation techniques in Dennis et al., 2013).

Therefore, for GAG_field, a constant $A_{patch}$ for every individual urination event and a constant $UF$ were assumed. There are values reported for $A_{patch}$ in the literature (Table 1), whose average was used in the baseline simulations and with a sensitivity test an estimation was given for the uncertainty resulting from this simplification (Section 4.2.5).

In the literature observational data can be also found for $UF$ (as shown in Table 1), but the temporal resolution of these data is usually a day. Based on personal communication with farmers, the hourly number of urine patches deposited over a field varies between the grazing and rumination periods and also between day and night. However, for the current modelling study an even distribution of urination events was assumed over the day, dividing the reported average daily $UF$ by 24 hours. As for $A_{patch}$, a sensitivity analysis was carried out for this parameter as well (Section 4.2.5).

Another feature of the individual urination events that strongly influences the subsequent $NH_3$ volatilization is $c_N$. This parameter ranges widely ($2 - 20$ g N $dm^{-3}$, Whitehead, 1995), not just amongst different animals, but also for different urination events by the same animal (Betteridge et al., 1986, Hoogendoorn et al., 2010). In the baseline simulation a constant average N content was applied. In Section 4.2.5, the response of the model was analysed to this choice of $c_N$ and also to the uncertainty originating from the temporal variation of this parameter.

### 2.2.3 Assumptions for the calculation of the net $NH_3$ flux

With all the above assumptions, two types of area can be distinguished over a grazed field: (a) area covered by urine, and (b) area that is not affected by urine, referred to hereafter as "non-urine area" (as shown on Fig. 4). Therefore, it was assumed that the total flux over the field is the sum of the emission from the urine affected area and the exchange with the non-urine area. Over the urine affected area the GAG model was applied to every single urine patch and for the non-urine area a modified version of the GAG model was used, assuming constant emission potentials, as explained later, in Section 3.1. One of the challenges of simulating bi-directional exchange at the field scale is that fluxes are both driven by atmospheric concentrations ($\chi_a$) -especially for deposition - and affect atmospheric concentrations -especially for emission (e.g. Loubet et al., 2009). In addition, due to the urine patches, a grazed field is not a uniform source of $NH_3$. One of the consequences is that the atmospheric concentration of $NH_3$ is not homogenous over the field (see e.g. Bell et al., 2016). Both effects result in a horizontal advection of $NH_3$, neglecting which leads to an error of the estimation of the total $NH_3$ flux. At the field scale, this effect can be explored by explicit consideration of horizontal gradients (Loubet et al., 2009) or by sensitivity analysis to the values of $\chi_a$. The purpose

of this work is to construct a model that can be applied for regional scale, where the overall effect of bi-directional exchange can be incorporated as emission/deposition feeds back to the simulated value of $\chi_a$. Therefore, the model was kept at this, lower level of complexity, neglecting the horizontal advection of $NH_3$. To investigate the effect of $\chi_a$ on the simulated $NH_3$ flux, a sensitivity analysis for $\chi_a$ was carried out (Section 4.2.2).

Finally, the field was assumed to have spatially homogenous physical and soil chemical properties before urine application. This assumption in tandem with the exclusion of the overlap of the urine patches and the horizontal dispersion of $NH_3$, leads to the consequence that the total flux over the field is independent of the placement of the patches on the surface.

# 3 Material and methods

## 3.1 Model equations for the field-scale application

Based on the considerations outlined in the previous subsections, for GAG_field we assumed that physically and chemically identical urine patches are deposited in every time step over the modelling period. To capture the effect of all of the urine patches, in calculating the net $NH_3$ flux for the whole field ($F_{net}$), an $m \times m$ matrix can be considered (see Fig. 5, where $m$ is the number of the time steps in the modelling period). In this matrix $i$ index denotes the time step for which the given flux is derived and $j$ shows the time step when the patches were deposited. In this way, $F_{net}$ in the $i^{th}$ time step ($t_i$) can be expressed

by Eq. (5).

The first term in the numerator of Eq. (5) represents the $NH_3$ emitted by the non-urine area: the $NH_3$ exchange flux over the non-urine area ($F_{non}$) multiplied by the size of this area ($A_{non}$). While the second term in the numerator equals to the total $NH_3$ emitted from the urine patches, where $F_{patch}^j$ is the emission flux from the urine patches deposited in the $j^{th}$ time step, and $n(t_j)$ is the number of the patches deposited in the same time step. To calculate $F_{net}$, the sum of the two has to be divided by $A_{field}$

(Eq. 5).

$$F_{net}(t_i) = \frac{F_{non}(t_i)A_{non}(t_i) + \sum_{j=1}^{m} F_{patch}^j(t_i)n(t_j)A_{patch}}{A_{field}} \tag{5}$$

In the non-urine area, in the absence of any considerable nitrogen input, the soil chemistry is practically undisturbed. Thus, for the non-urine area a modified version of GAG_patch was applied in which constant soil chemistry was assumed. Based on this, $F_{non}$ was derived in the same way as $F_t$, the net $NH_3$ flux over a urine patch in GAG_patch, described by Eq. (1)-(7) in

Móring et al. (2016), together with the following simplifications:

- Since over the non-urine area undisturbed soil chemistry is assumed, the dynamic simulation of soil chemistry in GAG_field is not needed. Therefore, the original version of the two-layer canopy compensation point model by Nemitz et al. (2001) is used. While dynamic simulation of undisturbed soil chemistry would be a useful avenue for further research, it is not addressed in the present study. The model by Nemitz et al. (2001) includes only the original

compensation point on the ground ($\chi_g$), instead of the soil resistance and compensation point in the soil assumed for GAG_patch. As a consequence, for the non-urine area the equation in GAG_patch for the NH$_3$ emission from the soil ($F_g$) changes to:

$$F_g = \frac{\chi_g - \chi_{z_0}}{R_{ac} + R_{bg}} \, ,\tag{6}$$

where $\chi_{z0}$ represents the canopy compensation point, and $R_{ac}$ and $R_{bg}$ stand for the aerodynamic resistance within the canopy and the quasi-laminar resistance at the ground, respectively (see the applied resistance model in the supplementary material on Fig. S1). For the parametrization of these variables in GAG_patch see Móring et al. (2016).

- The value of $\chi_g$ (Eq. 7) for the non-urine area was calculated similarly to that of the compensation point in the soil pore in GAG_patch ($\chi_p$), except that the NH$_3$ emission potential for the ground ($\Gamma_g$) was handled as a constant (Section 3.2.3) instead of being modelled dynamically as in GAG_patch in the soil pore. In Eq. (7) $T_{soil}$ represents the soil temperature.

$$\chi_g = \frac{161500}{T_{soil}} \times \exp\left(\frac{-10380}{T_{soil}}\right) \times \Gamma_g \tag{7}$$

- Since over the non-urine area no N input is assumed, for the emission potential of the stomata ($\Gamma_{sto}$), instead of applying a decay function, like in GAG_patch, it was treated as constant (Section 3.2.3).

The size of $A_{non}$ in the given $t_i$ time step is the area of the field that is not covered by any urine patches (assuming no overlap):

$$A_{non}(t_i) = A_{field} - \sum_{j=1}^{i} n(t_j) A_{patch} \, ,\tag{8}$$

where $n(t_j)$ (Eq. 9) is the number of the urine patches deposited in the $j^{th}$ (hourly) time step. This can be expressed as the product of the animal density on the field in $t_j$ ($AD(t_j)$, animals ha$^{-1}$), $A_{field}$ (ha) and the daily $UF$ (urinations day$^{-1}$ animal$^{-1}$), divided by 24 hours.

$$n(t_j) = \frac{(AD(t_j) \times A_{field} \times UF)}{24} \tag{9}$$

Finally, $F_{patch}^j(t_i)$ was determined by Eq. (10), which expresses that before the deposition of the urine patch, the area is handled as non-urine area (first condition), and afterwards GAG_patch calculates the net NH$_3$ flux over the urine patch ($F_t(t_i)$, second condition).

$$F_{patch}^j(t_i) = \begin{cases} F_{non}(t_i) & \text{if} \quad i < j \\ F_t(t_i) & \text{otherwise} \end{cases} \tag{10}$$

When calculating $F_t(t_i)$ a slight modification is also required compared with the GAG_patch model, regarding the urea added with a single urination ($U_{add}$). At field scale it has to be considered that during the modelling period urine patches may be deposited at the same time as a rain event occurs. A rain event

i) will dilute the incoming urea solution, and

ii) may lead to the maximal water content ($B_{H2O}(max)$) in the $NH_3$ source layer. In the formulation of GAG_patch this means that for the incoming liquid there is no more soil pore to fill, i.e. there is no infiltration. Therefore, when a urine patch is deposited while the water content is at $B_{H2O}(max)$, will result in no N input to the system and consequently, no $NH_3$ emission from the soil.

To address point i), it has to be noted that although over the non-urine area GAG_field does not simulate the dynamic, temporal evolution of the TAN budget and the soil pH (a constant $\Gamma_g$ is used as noted above), it does account for the changes in water budget ($B_{H2O}$) in the source layer. Therefore, the water budget for the non-urine area (simulated by the modified version of the GAG_patch as described above), right before the $j^{th}$ patch deposition ($B_{H2O}{}^j(t_i = (j - 1))$) can be updated by GAG_patch in the next time step ($B_{H2O}{}^j(t_i = j)$). Although the effect of dilution is treated in GAG_patch, it is defined only for the first time step, when urine is applied to the surface. This means that in Móring et al. (2016) $U_{add}$ was not defined as a function of time. Therefore, in the field-scale model, where urine patches are deposited in every time step, $U_{add}$ was calculated for all of the urine patches deposited in every $t_j$ as:

$$U_{add}(t_j) = c_N^{Dil}(t_j)\left(B_{H_2O}^j\left(t_{i=(j)}\right) - B_{H_2O}^j\left(t_{i=(j-1)}\right)\right), \tag{11}$$

where the diluted N concentration in the mixture of rain water and urine ($c_N{}^{Dil}$, Eq. 12) equals to the total amount of N in the urine ($c_N \times W_{urine}$) divided by the sum of the volume of the liquid phase ($W_{urine} + W_{Rain}(t_i = j)$, where $W_{rain}$ denotes the volume of the infiltrating rain water).

$$c_N^{Dil}(t_j) = \left(\frac{c_N W_{urine}}{W_{urine} + W_{Rain}(t_{i=j})}\right) \tag{12}$$

To avoid the possible error resulting from the second point, it was assumed that instead of no infiltration, a small amount of water is always allowed to penetrate to the soil. This amount was chosen to be the 5% of $B_{H2O}(max)$, as shown in Eq. 13. This assumption is necessary since in reality in most of the cases there is infiltration to the soil (except after heavy rain or an elongated rain event), therefore, there is $NH_3$ emission from the soil even if the urine patch deposited to a very wet soil. However, in this case, the $NH_3$ emission flux from the soil might be weaker for two reasons: 1) due to the soil wetness, the urine might dilute after its deposition, leading to a lower $\chi_p$ and 2) the high water content is associated with large soil resistance, leading to a weaker $NH_3$ emission flux. Therefore, the choice of 5% of $B_{H2O}(max)$ could be reasonably large to avoid zero soil emission, but reasonably small to represent the described effects.

$$\left(B_{H_2O}^j\left(t_{i=(j)}\right) - B_{H_2O}^j\left(t_{i=(j-1)}\right)\right) \geq 0.05 \times B_{H_2O}(max) \tag{13}$$

## 3.2 Dataset used in the baseline simulations and model evaluation

### 3.2.1 Measurements

GAG_field was evaluated (Section 4.1) using measurements taken at a grassland site near Easter Bush, UK (see the field specific data in Table 2) by CEH (Centre for Ecology & Hydrology). The field is divided into two halves, the North Field and the South Field, and the instruments were placed on the boundary of the two (Fig. 6). For the site, $NH_3$ flux measurements are available for a number of years (2001-2007). These fluxes were derived using the aerodynamic gradient method, which calculates the fluxes ($F_\chi$) based on measurements of the vertical gradient of $NH_3$ air concentration and micrometeorological variables (Eq. 14). In Eq. (14) $\chi$ denotes the $NH_3$ air concentration measured at a $z$ height, whilst $k$, $u_*$, $d$, $\Psi_H$ and $L$ stand for the Karman constant, friction velocity, displacement height, the stability function for heat, and the Monin-Obukhov length, respectively.

$$F_\chi = -ku_* \frac{\partial\chi}{\partial\left[\ln(z-d) - \Psi_H\left(\frac{z-d}{L}\right)\right]}, \tag{14}$$

Ammonia concentration measurements were conducted by using a high-resolution $NH_3$ analyser, AMANDA (Ammonia Measurement by ANnular Denuder sampling with online Analysis, Wyers et al., 1993). During the sampling, gaseous $NH_3$ is captured in a continuous flow rotating annular wet denuder applying a stripping solution of 3.6 mM sodium hydrogen sulphate ($NaHSO_4$). The technique determines the air concentration of $NH_3$ online by conductivity detection (Milford et al., 2001). The concentration gradients were obtained from concentration measurements at three heights: 0.44, 0.96 and 2.06 m.

The meteorological input variables that are required for a simulation with GAG_field are the same as for GAG_patch. From these, air and soil temperature ($T_{air}$ and $T_{soil}$), relative humidity ($RH$), precipitation ($P$), atmospheric pressure ($p$), global radiation ($R_{glob}$), wind speed ($u$), wind direction ($u_{dir}$) and sensible heat flux ($H$) were observed at Easter Bush. For further details on instrumentation see Milford et al. (2001). Since photosynthetically active radiation ($PAR$, μmol m$^2$ s$^{-1}$) was not measured at the site, it was calculated from $R_{glob}$ as shown in Eq. (15). According to Emberson et al. (2000), $PAR$ is 45-50% of $R_{glob}$ (0.475 in Eq. 15), and it is expressed in μmol m$^{-2}$ s$^{-1}$ (to the unit of $R_{glob}$, Wm$^{-2}$, a conversion factor of 4.57 should be applied). The measured input data is illustrated in the supplementary material, in Fig. S2 and S3.

$$PAR = R_{glob} \times 0.475 \times 4.57 \tag{15}$$

### 3.2.2 Processing of the measured data for model application

For the baseline simulation and model evaluation (Section 4.1), a subset of the measurement data for 2001-2007 was selected that fulfilled the following criteria:

1. there were animals on the field;
2. grazing started at the beginning of the modelling period;
3. there had been no grazing, fertilizer spreading or grass cutting in the week before the grazing started;

4. there are no significant gaps in the meteorological input data;

5. flux measurements are available for validation.

The second criterion is important because NH$_3$ fluxes over the field can be affected by emission from urine patches deposited earlier. If the model does not account for these, it may underestimate the fluxes. The management practices listed in the third

5   criterion can also affect the NH$_3$ exchange in a given time step, as well as fertilization can considerably affect the chemical balance of the soil. The latter would conflict with the model assumption that urine patches are deposited to a non-affected soil. The fourth criterion is necessary, because a continuous input dataset is needed for a simulation, since within GAG_patch the TAN, the water and the H$^+$ budgets in a given time step are dependent on the values in the previous time.

As a result of the filtering, two suitable time periods were found: 26/08/2002 00:00 - 03/09/2002 06:00 and 20/06/2003 00:00

10   - 25/06/2003 05:00. These periods are referred herby to as P2002 and P2003, respectively. In both time intervals cattle were grazing on the South Field. Their number over the two modelling periods is indicated in Table 2.

To prepare the measured datasets for the hourly model application, firstly, the flux measurements were assessed for stability of the AMANDA instrumentation record with periods of obvious instrument malfunction and gaps in data removed (Móring, 2016). All data were then averaged too an hourly time resolution. The time resolution of the ambient air concentration ($\chi_a$), $u$,

$T_{air}$ and $F_\chi$ (all at 1 m height) as well as $T_{soil}$ was 15 minutes, whilst it was 30 minutes for $p$, $R_{glob}$ and $RH$. Secondly, in the resulted averaged time series (except in $F_\chi$) gap-filling was carried out. Data were missing from the $\chi_a$ dataset for the simulation for P2002:

- over 27/08 13:00 – 28/08 13:00,
- on 02/09 at 23:00.

The individual gap was interpolated from the values from the previous and next time step, whilst over the long period of missing data in $\chi_a$ (25 consecutive hourly time steps), the values were replaced by the average of the measured values of $\chi_a$ over P2002 (1.71 µg m$^{-3}$). In P2003 a single, hourly wind speed was missing at 01:00 on 25/06, which was interpolated based on the data in the neighbouring two time steps.

In the third step of data processing, the measured fluxes were filtered according to the wind direction. As mentioned above,

animals were grazing on the South Field and the fluxes were measured at the border line of the two fields (Fig. 6). Therefore, to distinguish the fluxes over the investigated part of the field, only the fluxes were used in the comparison that were associated with wind from the direction of the South Field, between 135° and 315°. The wind blew from this direction in most of the time. In the two modelling periods in P2002 and P2003 the wind direction was the opposite in the 7% and 15% of the hourly time steps, respectively.

In addition, a quality check was carried out on the measured flux dataset, distinguishing the time periods with low wind and strong stability. A flux measurement was considered robust if it met all of the following criteria:

-     according to the footprint analysis, the field contributed at least 67% to the measured flux,
-     $u_* > 0.15$ m s$^{-1}$ for at least 45 minutes,
-     $L^{-1} < 0.2$ m$^{-1}$, and

- $u > 1$ m s$^{-1}$.

The fluxes failing to meet one or more of the above criteria were considered as less robust. The robust and less robust data determined in this way, can be seen in Section 4.1.1 on Fig. 9.

Finally, although the NH$_3$ concentrations measured in the time steps with $u_{dir}$ from the North Field represents the concentration in the North Field, in order to keep the continuity in the input data, these values were kept in the dataset. If they were substituted with zeros (similarly as it was handled in the gap-filling of $\chi_a$), another type of error would have been added to the input data. Considering the relatively small number of $u_{dir}$ values from the direction of the North Field, this choice is not anticipated to result in large errors in the NH$_3$ flux simulations.

### 3.2.3 Model constants

The main urine-patch-specific constants defined by Móring et al., (2016) for GAG_patch, are the soil buffering capacity ($\beta =$ 0.021 mol H$^+$(pH unit)$^{-1}$ dm$^{-3}$) and the thickness of the NH$_3$ source layer ($\Delta z = 4$ mm), were not changed in the model experiments with GAG_field. The other field, urine and site specific constants together with their sources are listed in Table 2.

For the constant $\Gamma_{sto}$, for the non-urine area of the field, where no considerable N input is assumed, the values from the emission potential inventory by Massad et al. (2010) for unfertilized grasslands were averaged. Since in the referenced inventory there were no $\Gamma_g$ estimates for non-fertilized grasslands, it was defined during preliminary simulations with GAG_field over a time interval when the grassland was not disturbed by any kind of management practice (grazing, fertilizer spreading or grass cutting). The period of 01/06/2003 00:00 – 08/06/2003 16:00 fulfilled these criteria. These preliminary model experiments indicated a reasonable agreement between the measured and simulated NH$_3$ fluxes with a $\Gamma_g$ of 3000 (see Fig. S4 in the supplementary material). Therefore, this value of $\Gamma_g$ was applied in the baseline simulations with GAG_field. To investigate the model sensitivity to this choice of $\Gamma_g$, a sensitivity analysis was carried out in Section 4.2.2.

## 3.3 Methods used in the sensitivity analysis

### 3.3.1 Perturbation experiments

Similarly to the model perturbation experiments carried out with GAG_patch (Móring et al., 2016), a sensitivity analysis of GAG_field to the regulating model parameters (Section 4.2.1-4.2.4) was performed. In addition to the parameters that were investigated for GAG_patch ($\Delta z$, $\beta$, $REW$ – readily evaporable water, $\theta_{fc}$ – field capacity, $\theta_{pwp}$ – permanent wiling point), $\Gamma_{sto}$, $\Gamma_g$, pH($t_0$) (soil pH before urine deposition), $\chi_a$, $LAI$ (leaf area index) and $h$ (canopy height) were also examined. The value of $\theta_{fc}$ and $\theta_{pwp}$ express the maximum and the minimum volumetric water content in the soil, including the NH$_3$ source layer. For a detailed description of $REW$, see Móring et. al (2016).

The perturbation experiments were carried out as follows: the investigated parameter was modified with ±10% and ±20%, whilst the other parameters were kept the same. In the case of the perturbation experiments for $\chi_a$, $\chi_a$ was modified by the

±10% and ±20% of its average over both periods. These average concentrations in P2002 and P2003 were 1.73 µg $NH_3$ m$^{-3}$ and 1.51 µg $NH_3$ m$^{-3}$, respectively. At the end of every simulation, the total $NH_3$ exchange ($\Sigma F_{net}$) was calculated by summing the modelled hourly $NH_3$ fluxes in the given modelling period. The difference compared with the baseline simulations was expressed in two ways. Firstly, it was calculated as the percentage of $\Sigma F_{net}$ in the baseline model integrations (127 g N and 403 g N net emission for the whole field in the baseline simulations for P2002 and P2003, respectively), denoted as $Sens_{net}$. Secondly, the differences were derived as the absolute average hourly change, i.e. $\Sigma F_{net}$ in the actual perturbation experiment minus $\Sigma F_{net}$ in the baseline simulation, divided by the length of the modelling periods (199 hours and 126 hours in P2002 and P2003, respectively).

In addition to the percentage differences for the whole field, similarly, the proportional change ($Sens_{patch}$) in the total $NH_3$ emission was calculated separately for the area covered by the urine patches ($\Sigma F_{patch}$) as well. In the baseline simulations, the total $NH_3$ emission from the urine patches were 717g N and 846 g N in P2002 and P2003, respectively. Finally, for $\beta$, $\theta_{fc}$, and $\theta_{pwp}$, the percentage differences in the total $NH_3$ emission ($\Sigma F_{patch}^{single}$) were calculated for every single urine patch deposited over both modelling periods, denoted as $Sens_{patch}^{single}$.

When the results from the sensitivity analysis for GAG_field and GAG_patch is compared (latter carried out by Móring et al., 2016), differences can occur for three reasons:

1) in GAG_field the total net $NH_3$ exchange consists of not only the total $NH_3$ emission over the urine patches, but also the total net $NH_3$ exchange over the non-urine area,

2) in GAG_field multiple urine patches are deposited in every time step, whilst in GAG_patch a single urine patch is simulated,

3) and the two models were applied for two different sites with different circumstances: GAG_field was applied for a grazed grassland at Easter Bush, Scotland and GAG_patch was evaluated for a grassland at Lincoln, New-Zealand.

For point 1), an insight can be gained if $Sens_{net}$ and $Sens_{patch}$ is compared. The differences originating from point 2) can be investigated based on the comparison of $Sens_{patch}$ and $Sens_{patch}^{single}$ derived for the single urine patches deposited in each time step of P2002 and P2003. Finally, the differences between the results of the perturbation experiments with GAG_patch in Móring et. al (2016) and those calculated for every urine patch in P2002 and P2003 ($Sens_{patch}^{single}$) will reflect the effect of the different circumstances at the two sites GAG_field and GAG_patch were applied for (point 3).

### 3.3.2 Further methods used in the sensitivity analysis

As explained in Section 2.2.2, for $c_N$, $A_{patch}$ and $UF$ constant, average values were applied in the baseline simulations with GAG_field. However, in reality these parameters can vary amongst different animals, and amongst different urination events as well. To examine the model uncertainty caused by these model assumptions, firstly, a sensitivity analysis was carried out (Section 4.2.5) applying the minimum and the maximum of these parameters as suggested in the literature (Table 1 and 2 – 20 g N dm$^{-3}$ for $c_N$ from Whitehead, 1995).

Since the results indicated that the largest uncertainty is coupled with $c_N$ (Section 4.2.5), in the case of this parameter further examinations were carried out. In natural conditions, even within an hour, several different urine patches are deposited over the field. For example, calculating with the lowest animal number on the field in the baseline experiment with GAG_field (17 from Table 2) and the minimal $UF$ (8 urination day$^{-1}$ cattle$^{-1}$, from Table 1), there were at least 5 urine patches deposited in an hour. When the number of urine patches is high enough, it can be assumed that the overall $c_N$ of all the urine deposited in a given hour is characterized by the average of the $c_N$ values related to the individual urination events. This can be expressed by Eq. (16), in which $c_N^{Ave}(t_j)$ represents the average N concentration in the time step $t_j$, $c_N^k(t_j)$ stands for the N content associated with the $k^{th}$ urine patch in $t_j$, and n($t_j$) is the number of urine patches deposited in $t_j$.

In the baseline simulations with GAG_field, $c_N^{Ave}$ was assumed to be 11 g N dm$^{-3}$ over the whole modelling period, therefore, it was examined how the model responds to a value of $c_N^{Ave}$, which is calculated in every time step according to Eq. (16). To approach this task, firstly $c_N^k$ values have to be randomized for every urination event from an estimated statistical distribution of $c_N$.

$$c_N^{Ave}(t_j) = \frac{\sum_{k=1}^{n(t_j)} c_N^k(t_j)}{n(t_j)} \tag{16}$$

Li et al. (2012) fitted a log-normal distribution (Eq. 20) to a $c_N$ dataset, originating from the observation of two Aberdeen Angus steers over three 24 hour periods (Betteridge et al., 1986). In Eq. (17) $\sigma$ and $\mu$ are the scale parameters of the distribution. These, in the fitted distribution by Li et al. (2012), were $\sigma = 0.786$ and $\mu = 1.154$. The arithmetic mean of $c_N$ calculated from these values (Eq. 18) was 4.33 g N dm$^{-3}$. In the study of Li et al. (2012), the findings were applied for cows, assuming that the distribution of $c_N$ is similar with the same $\sigma$, but a higher mean $c_N$. Based on these, from Eq. (18), Li et al. (2012) derived $\mu$ of the new distribution for cows and from this they generated a series of samples for $c_N$.

$$D(c_N) = \frac{1}{c_N \sigma \sqrt{2\pi}} e^{-\frac{(\ln c_N - \mu)^2}{2\sigma^2}} \tag{17}$$

$$mean(c_N) = e^{\mu + \frac{\sigma^2}{2}} \tag{18}$$

To test the uncertainty coupled to $c_N$ in GAG_field, the following steps were carried out. Firstly, following the method described by Li et al. (2012), based on Eq. (18), a new distribution of $c_N$ was obtained, assuming a mean $c_N$ of 11 g N dm$^{-3}$, and $\sigma = 0.786$. In this way, the scale parameter $\mu$ was found to be 2.089. The resulted distribution of $c_N$ is depicted in Fig. 7. Secondly, in every time step $c_N^k$ values were randomized from the resulted distribution, and from these, $c_N^{Ave}$ was derived based on Eq. (16). This resulted in a time series of $c_N^{Ave}$ values. In total, 30 $c_N^{Ave}$ time series were generated for both experimental periods (P2002 and P2003) and simulations were performed with GAG_field, for all of these time series.

Finally, in order to investigate the model response of GAG_field to a constant value of soil pH, model experiments were performed with different constant values of soil pH (Section 4.2.6).

# 4 Results

## 4.1 Model results derived by GAG_field

The model results for P2002 and P2003 are illustrated in Fig. 4. These model experiments are regarded as the baseline simulations and are discussed in Sections 4.1.1. In addition to the general evaluation of these model results, in Section 4.1.2, the contribution of the $NH_3$ emission from the urine patches to the $NH_3$ exchange over the whole field is also investigated.

## 4.1.1 Baseline simulations and model evaluation

In the case of P2002, although the model statistics imply a weak model performance (Fig. 8a), the visual comparison of the modelled and measured $NH_3$ exchange (Fig. 9a) suggests a broad accordance between the two datasets. The model captures the characteristic daily variation of $NH_3$ exchange detected over 31/08-02/09, with the magnitudes of the modelled and measured generally within 50 ng $m^{-2}$ $s^{-1}$. A larger difference occurred on 02/09 when the model clearly underestimated the observations. Discrepancies between the simulated and measured values can be also seen in the first two days of the modelling period and on the fourth day. Nevertheless, on these days the bottom $NH_3$ concentration sensor did not work; therefore, the reliability of the flux calculated based only on the concentration measurements at the middle and top level is less certain. In addition, according to the metadata, on 27/08, before the gap in the observed fluxes (Fig. 9a), the stripping solution of the denuder ran out. This could explain the last 2-3 very high measured values beforehand. When the last 6 values before this event as well as the less robust data were removed from the dataset, the calculated statistics reflected a much promising model performance.

Similarly to P2002, the model statistics imply a relatively low model performance (Fig. 8b) for P2003 as well, however, according to Fig. 9b, the simulation generally agreed with the observations within 50 ng $m^{-2}$ $s^{-1}$. The removal of the less robust data from the dataset, resulted in improved model statistics (Fig. 8b), suggesting a better agreement between the model and the measurements. The match with the observed fluxes was especially close in the second half of 23/06. By contrast, the largest difference was found on 24/06, in the morning, when an emission peak was detected during the measurements at 04:00-08:00. Even though there was a midday peak also in the simulation, it occurred 6 hours later than the maximum in the observation. The increase in measured fluxes on 24/06 was linked to a period of high wind speed (with largest values between 04:00-08:00 AM, not shown here). Although wind speed is included in the model, the larger effect on measured fluxes could imply a proportionately larger effect of turbulence on the fluxes (through atmospheric and within canopy resistances, see the parametrization in Móring et al., 2016) than estimated by the model. In addition, it should be noted that on 20/06 between 11:00 and 15:00 the $NH_3$ concentration denuder in the bottom height was not functioning properly, and afterwards it was not operating until 23/06 13:00 PM (in these periods only the remaing two denuders were considered), suggesting uncertainty in the measured dataset.

### 4.1.2 Contribution of the urine patches to NH₃ exchange over the field

Figure 10 distinguishes the contribution of the urine patches and the non-urine area to the simulated $NH_3$ exchange flux for the two modelling periods. It can be seen that the temporal variation of the $NH_3$ fluxes over the whole field were dominated by the $NH_3$ emission from the urine patches, which was offset by simultaneous $NH_3$ deposition to the non-urine area. In the absence of the urine patches in both experiments, deposition would have occurred for most of the time. This illustrates the considerable effect of the presence of grazing animals on $NH_3$ exchange over grasslands.

The contribution to the $NH_3$ exchange flux was also investigated for the groups of patches deposited in the different time steps (Fig. 11). The ensemble of the fluxes from the different patches show a clear daily variation with $NH_3$ emission peaks at midday in both modelling periods. In P2002, these peaks became lower from the fourth day because after the third day instead of the initial 40 animals, only 17 cattle were grazing on the field, depositing fewer urine patches.

In the baseline experiment with GAG_patch, the first and highest peak in $NH_3$ emission occurred about 12 hours after the urine application (Móring et al., 2016). By contrast, in the current results using GAG_field (Fig. 11) it can be observed that in some cases the highest peak over an individually deposited urine patch emerges more slowly, only a day or two days after the urination event. For example, in P2002 (Fig. 11a) from the urine patches deposited on the third day (orange lines) the highest emission occurred on the fourth day, or from the patches deposited on the sixth day (dark green lines) the maximal flux was observed two days later. Further examples from P2003 (Fig. 11b) are the urination events on the second day (orange lines) from which the highest flux can be observed a day after.

It has to be also noted that $NH_3$ emission fluxes in a given day can be substantially affected by urine patches deposited several days earlier. For instance, in Fig. 11a, on 02/09 the fluxes originating from the urination events six days before (red lines) are comparable with those from urine patches deposited two days before (dark green lines).

### 4.2 Sensitivity analysis to the regulating model parameters

In the following, first, the results of the perturbation experiments (Table 3) with GAG_field are discussed (Section 4.2.1-4.2.4). Secondly, in Section 4.2.5 the uncertainty associated with $c_N$, $A_{patch}$ and $UF$ is investigated. Finally, model experiments are presented in which GAG_field was tested with different constant values of soil pH (Section 4.2.6).

### 4.2.1 General remarks

Based on Table 3, some preliminary, general conclusions can be drawn. Firstly, a near constant ratio of $Sens_{net}$ and $Sens_{patch}$ can be observed for the urine-patch related parameters. These are the parameters that are used in the formulation of GAG_field only for the urine patches: $\Delta z$, $\beta$, $REW$, $\theta_{fc}$, $\theta_{pwp}$, and $pH(t_0)$ (initial soil pH). These have an effect on the $NH_3$ exchange for the whole field only through the $NH_3$ emission from the urine patches.

The value of $\Delta z$, $REW$, $\theta_{fc}$, and $\theta_{pwp}$ influences the water budget, which is considered in the calculation of the stomatal resistance for both the non-urine area and the patches (Móring et al., 2016). However, preliminary results indicated that without the urine

patches (assuming only non-urine area), the change in the total $NH_3$ exchange over the field in response to the perturbations applied to these parameters were negligibly small (under 1% in absolute value). Therefore, the effect of $\Delta z$, $REW$, $\theta_{fc}$, and $\theta_{pwp}$ on the total $NH_3$ exchange over a grazed field through the non-urine area can be ignored.

In essence, when $\Delta z$, $\beta$, $REW$, $\theta_{fc}$, $\theta_{pwp}$, and $pH(t_0)$ perturbed, the changes of the total exchange flux are attributed exclusively to the changes in the emission flux over the urine patches. Therefore, as shown in the following, for these parameters the ratio of $Sens_{net}$ and $Sens_{patch}$ is close to constant. Since the net $NH_3$ exchange over the whole field equals to the sum of the $NH_3$ emission from the urine patches and the $NH_3$ exchange over the non-urine area (Fig. 4), the total $NH_3$ exchange over the whole field ($\Sigma F_{net}$, Eq. 19) over a time interval is equal to the sum of the total $NH_3$ exchange over the non-urine area ($\Sigma F_{non}$) and the total $NH_3$ emission from the urine patches ($\Sigma F_{patch}$). Therefore, based on Eq. (19), when a urine-patch-related parameter is perturbed, the resulting differences ($\Delta F$) in $\Sigma F_{patch}$ and $\Sigma F_{net}$ will be the same.

$$\sum F_{net} = \sum F_{non} + \sum F_{patch} \tag{19}$$

Using $\Delta F$, $Sens_{patch}$ and $Sens_{net}$ can be expressed as:

$$Sens_{patch} = \frac{\Delta F}{\sum F_{patch}} \tag{20}$$

$$Sens_{net} = \frac{\Delta F}{\sum F_{net}} \tag{21}$$

Based on these, it can be clearly seen that the ratio of $Sens_{net}$ and $Sens_{patch}$ equals to the ratio of $\Sigma F_{patch}$ and $\Sigma F_{net}$. These ratios are 5.6 and 2.1 for P2002 and P2003, respectively, which is in accordance with the $Sens_{net}$ and $Sens_{patch}$ values in Table 3.

Secondly, in Table 3 it can be also seen that the absolute hourly changes (values in brackets) for the patch-related parameters are about 2-3 times larger in P2003 than P2002. The main reason for this is that on an hourly basis in P2003 the deposition rate of the urine patches was larger than in P2002. On average, in P2003 and P2002, 21 and 8 urine patches were deposited in an hour, respectively. The ratio of the two, 2.625, is in agreement with the observed ratio in the hourly changes for P2002 and P2003.

Finally, based on the results of Table 3, it is clear that $Sens_{net}$ is substantially affected by $\Sigma F_{net}$. For example, when $\chi_a$ was perturbed by -20%, the absolute changes in $\Sigma F_{net}$ were similar in P2002 and P2003 (+1.06 and +1.16 g N hr$^{-1}$, respectively), however, there was an enormous difference in the resulted $Sens_{net}$ values (+166% and +36%). This suggests that when the model behaviour is compared for P2002 and P2003, the $Sens_{net}$ values can be interpreted only together with the hourly absolute changes of $\Sigma F_{net}$. To visually compare these absolute changes with the values on Fig. 9, the hourly average error of the measurements can be taken as a base: ±2.86 g N and ±2.46 g N in P2002 and P2003, respectively, after conversion from flux (ng $NH_3$ m$^{-2}$ s$^{-1}$) to total emission for the whole field.

### 4.2.2 Sensitivity to $\Delta z$, REW, pH($t_0$), $\Gamma_{sto}$ and $\Gamma_g$, $\chi_a$, LAI and $h$

According to Table 3, compared with the other patch-related parameters, for GAG_field, $\Sigma F_{net}$ turned out to be the least sensitive to the changes in $\Delta z$ and REW. The $Sens_{patch}$ values were similar in the case of the perturbation experiments with GAG_patch, with an overall, slightly stronger sensitivity than was found in the case of GAG_field.

In the case of pH($t_0$), $\Sigma F_{net}$ was found to be very sensitive to the ±10% and ±20% modifications (Table 3). However, it has to be pointed out that these changes in the value of pH($t_0$) (±0.5 unit for a ±10% modification and ±1 unit for ±20%), can be considered as a large increase in the soil pH, taking into account that during intensive urea hydrolysis 2-3 units change can be expected (Fig. 12).

The constant $\Gamma_{sto}$ and $\Gamma_{soil}$ affect NH$_3$ exchange over the whole field exclusively through its effect on the NH$_3$ exchange over

the non-urine area. As the results show (Table 3), the model is only slightly sensitive to $\Gamma_{sto}$, whilst $\Gamma_g$ can have a considerable effect on NH$_3$ exchange. As it can be seen, for $\Gamma_{sto}$ the resulted changes in $\Sigma F_{net}$, depending on the modelling period, are about 5-15% of the perturbations applied to $\Gamma_{sto}$. This means that if a 5 times larger $\Gamma_{sto}$ (+400% perturbation, assuming a soil richer in N) was used in the model runs, the resulted $\Sigma F_{net}$ would be about 20-60% larger, with an overall hourly difference of 0.4 g N.

As for $\chi_{air}$, in Table 3 the percentage differences for P2002 over the whole field suggest a significant effect on $\Sigma F_{net}$. However, comparing the absolute hourly change to that for P2003, it can be concluded that the absolute influence was similar for the two periods. It can be also clearly seen that the absolute hourly changes over the urine patches are negligibly small in both P2002 and P2003 compared to the absolute changes observed for the whole field, suggesting that $\chi_{air}$ affects $\Sigma F_{net}$ mainly through the non-urine area, rather than the urine patches.

The effect of LAI on $\Sigma F_{net}$ turned out to be weak. The resulting percentage differences are negligibly small compared to the extent of the perturbations applied. Similarly, a relatively weak sensitivity was found for $h$. However, in this case, it has to be noted that the resulting percentage differences are about half of the perturbations. This means that in the case of e.g., a canopy height of 5 cm, which is -83% shorter than the $h$ used in the baseline simulations, could lead to considerable changes in the NH$_3$ exchange flux, especially toward the end of the period when the grass is shorter on the field due to the continuous grazing.

### 25   4.2.3 Sensitivity to $\beta$

In the case of $\beta$, strong sensitivity was detected in $\Sigma F_{net}$ (Table 3), and the values of $Sens_{patch}$ were 10-20 times larger than the $Sens_{patch}^{single}$ values reported for GAG_patch. According to Fig. 13b and Fig. 14b, the sensitivity of the total NH$_3$ emission for the single urine patches in most of the cases were similar (close to the values of $Sens_{patch}$), except in the time steps where the values became scattered, in some cases with extremely high values. The scattered pattern largely disappeared when the

precipitation was assumed to be zero, leaving behind the high peaks associated with the events of dew fall (Figs. 13a and 14a). These results suggest that $Sens_{patch}^{single}$ is affected by the volumetric water content at the time of the deposition of the urine patch. Furthermore, comparing the patch sensitivities illustrated in Figs. 13b and 14b, with those in Table 3 reported by Móring

et al. (2016), a large difference occurs over the urine patches observed at the two different sites, Lincoln (NZ) and Easter Bush (UK). Therefore, in the following two questions are investigated:

- What causes the difference between the patches at the two different sites?
- What causes the high peaks in the sensitivity to $\beta$?

For both questions, the general model behaviour was examined through a series of model experiments with GAG_patch (Table 4).

In Móring et al. (2016), the $H^+$ ion budget depends on the $H^+$ ion consuming and producing processes related to the products of urea breakdown. On top of these, the effect of the buffers in the soil is expressed with an additional term: (pH($t_i$)-pH($t_{i-1}$)) × $\beta_{patch}$, where $\beta_{patch} = \beta \times A_{patch} \times \Delta z$. Based on these, the main factors that can regulate the governing role of buffering in the

evolution of soil pH in the $NH_3$ source layer and subsequently, $NH_3$ exchange, are

    1) pH($t_i$)-pH($t_{i-1}$), and

    2) $\beta_{patch}$.

Considering point 1), if pH($t_0$) is low, i.e. [$H^+$] is high, during urea hydrolysis more $H^+$ ion can be consumed. This results in a larger increase in soil pH shortly after the urine patch deposition. In the baseline simulations with GAG_patch and GAG_field

pH($t_0$) was 6.65 and 4.95, respectively. On Fig. 12 it can be observed that in most of the urine patches deposited in the baseline simulations with GAG_field, the difference between the initial and maximum soil pH was about 3 units, whilst in the case of the baseline experiment with GAG_patch (with the higher pH($t_0$)) it was only 2 (Móring et al., 2016).

These larger changes in soil pH generate a larger buffering effect ((pH($t_i$)-pH($t_{i-1}$)) × $\beta_{patch}$), i.e. a larger term in the $H^+$ budget. This means that in the GAG_field simulations, this term has a stronger effect in the $H^+$ budget, consequently, when $\beta$ is

modified (through $\beta_{patch}$), the system gives a stronger response, which means that the model is more sensitive to the perturbation of $\beta$. This was confirmed in the model experiment A (Table 4). In this simulation, GAG_patch was run with the initial pH of 4.95 used in the baseline simulation with GAG_field. Although the response of $NH_3$ exchange was relatively weak to the modifications of $\beta$, it was stronger than in the original perturbation experiment for GAG_patch (Table 3).

Regarding point 2): the definition of $\beta_{patch}$ expresses the buffering effect of the solid material of the soil on the liquid content.

As it can be seen from the formula $\beta_{patch} = \beta \times A_{patch} \times \Delta z$, $\beta_{patch}$ depends clearly on $\Delta z$, but it does not depend on the liquid content of the soil. This means that in the model, in a source layer with the same $\Delta z$, the same buffering effect takes place even if less urine stored in it. In a smaller amount of urine, the $H^+$ ion budget (expressed in mol $H^+$) and the variations in it are proportionally smaller too. Therefore, the governing role of the same buffering capacity in the case of a smaller amount of urine becomes stronger, resulting in a stronger model sensitivity to $\beta$.

The maximum volume of urine that can be stored in the $NH_3$ source layer ($\theta_{urine}$) can be calculated as the difference of $\theta_{fc}$ and $\theta_{pwp}$. The values of $\theta_{urine}$ in the baseline experiments with GAG_field and GAG_patch were 0.18 and 0.3, respectively. This, based on the above consideration, suggests a stronger response in $\Sigma F_{patch}^{single}$ to the perturbation of $\beta$ for the GAG_field experiments than the GAG_patch experiment. This effect was explored in the model experiment B (Table 4), in which the baseline simulation with GAG_patch was performed with $\theta_{fc}$ and $\theta_{pwp}$ applied from the baseline experiment with GAG_field

(Table 2). The results show a small difference in $\Sigma F_{patch}^{single}$ in response to the change of $\beta$, but it is still larger than in the sensitivity analysis carried out for the baseline simulation with GAG_patch (Table 3), supporting the effect described above. When the influence of pH($t_0$) and the soil water content characteristics were examined together (model experiment C, Table 4), their effect added up, reaching a ±10% difference in $\Sigma F_{patch}$ when $\beta$ was modified by ±20%.

The model was tested also with a higher $\theta_{pwp}$ (model experiment D, Table 4), assuming that half of the available space for urine in the model soil pore is filled with water, allowing only half of $\theta_{urine}$ to infiltrate. This can represent a situation on the field when a urine patch is deposited after a rain event, when only half of the soil pore is empty. As expected, due to the smaller amount of urine, with this modification the sensitivity to $\beta$ became even stronger.

Overall, these findings show that the difference in $Sens_{patch}^{single}$ in response to the perturbations of $\beta$ between the GAG_field

and GAG_patch simulations are mainly caused by the difference in $\theta_{fc}$ and $\theta_{pwp}$ as well as pH($t_0$) at the two different sites. Furthermore, the above results highlight that the sensitivity of $\Sigma F_{patch}^{single}$ to $\beta$ can vary between wide ranges over the individual urine patches on the same field, depending on the water content of the soil at the time of the given urination event.

### 4.2.4 Sensitivity to $\theta_{fc}$ and $\theta_{pwp}$

In the case of $\theta_{fc}$ and $\theta_{pwp}$, the perturbation experiments suggested an extremely strong sensitivity of $\Sigma F_{net}$ (Table 3), especially

in P2003, where the absolute changes exceeded the 2 g N hourly rate in several cases. Some of the changes in these parameters resulted in a $\Sigma F_{net}$ that was double or almost triple (+191% in P2003 when $\theta_{fc}$ was changed by +20%) of the $\Sigma F_{net}$ for the baseline simulation. Furthermore, $Sens_{net}$ was below -100% in many cases, suggesting that in response to the modifications of $\theta_{pwp}$ and $\theta_{fc}$ the originally positive total net exchange turned to deposition. The values of $Sens_{patch}$ for both P2002 and P2003 were less extreme than $Sens_{net}$, however these still suggest a substantially stronger sensitivity of $\Sigma F_{patch}^{single}$ to the modifications

of $\theta_{fc}$ and $\theta_{pwp}$ in the GAG_field model experiments than the GAG_patch experiments. Figs. 13c-d and Figs. 14c-d show a similar pattern in the $Sens_{patch}^{single}$ values for $\theta_{fc}$ and $\theta_{pwp}$ to those for $\beta$: most of the values are close to the corresponding $Sens_{patch}$ value, however, extreme values appear during the events of precipitation and dew fall, which affect the soil water content at time of the deposition of the urine patches. Similarly to $\beta$, the sensitivities observed in the GAG_patch experiment at the Lincoln site are significantly lower than those depicted on Figs. 13c-d and Figs. 14c-d for Easter Bush. In the following

these findings are further explored in additional model experiments with GAG_patch.

The value of $\theta_{fc}$ and $\theta_{pwp}$ influence $NH_3$ exchange over a urine patch predominantly through $\theta_{urine}$, affecting the amount of urea available for hydrolysis in the $NH_3$ source layer. Therefore, the difference in the response of $\Sigma F_{patch}^{single}$ to the changes in $\theta_{fc}$ and $\theta_{pwp}$ at the two sites, might be caused by the difference in the values of $\theta_{fc}$ and $\theta_{pwp}$. As it was pointed out above, in the baseline simulation with GAG_patch $\theta_{urine} = 0.4$, and over the field scale $\theta_{urine} = 0.18$. In the perturbation experiments, when

$\theta_{fc}$ and $\theta_{pwp}$ are modified this fillable space in the source layer is also affected. As it can be seen in Table 5, the ±10% and ±20% modifications of $\theta_{fc}$ and $\theta_{pwp}$ resulted in proportionally smaller differences in $\theta_{urine}$ in the case of the GAG_patch experiment at Easter Bush than the GAG_field simulations at Lincoln, suggesting a weaker response in $\Sigma F_{patch}^{single}$ for the Lincoln site.

This effect was explored within a series of model experiments with GAG_patch (Table 6), in which the $\theta_{fc}$ and $\theta_{pwp}$ used in the baseline simulation with GAG_patch (0.4 and 0.1, respectively) were changed to those applied in the baseline simulation with GAG_field (0.37 and 0.19, respectively). All the other parameters and input variables were kept the same as in the baseline simulation with GAG_patch. The experiments were carried out in two cases for both $\theta_{fc}$ and $\theta_{pwp}$: 1) when the initial water

content of the soil ($\theta(t_0)$) was assumed to be the $\theta_{pwp}$ ($\theta(t_0) = 0.19$) and 2) when half of the available space was filled by liquid ($\theta(t_0) = 0.28$), e.g. by rain water from a preceding rainfall.

As it can be seen in Table 6, with the $\theta(t_0) = \theta_{pwp}$ model setting the sensitivity to both $\theta_{fc}$ and $\theta_{pwp}$ became higher than in the case of the original perturbation experiment with GAG_patch (Table 3). This sensitivity became even stronger when urine was deposited to a half-filled source layer ($\theta(t_0) = 0.28$). These results suggest that one of the reasons for the large differences in

$Sens_{patch}^{single}$ between the GAG_field simulations and the GAG_patch simulation could be the different $\theta_{fc}$ and $\theta_{pwp}$ values over the two sites. In addition, the findings in Table 6 also imply that depending on the rain events and how they modify the initial water budget in the soil before a urination event, the sensitivity of $NH_3$ exchange to the perturbations of $\theta_{fc}$ and $\theta_{pwp}$ over the individual urine patches, deposited on the same field over the modelling period, can vary widely.

### 4.2.5 Sensitivity to $c_N$, $A_{patch}$ and UF

As explained in Section 2.2.2, for $c_N$, $A_{patch}$ and $UF$ constant, average values were applied in the baseline simulations with GAG_field. However, in reality these parameters can vary amongst different animals, and amongst different urination events as well. To examine the model uncertainty caused by these model assumptions, firstly, a sensitivity analysis was carried out applying the minimum and the maximum of these parameters as suggested in the literature (Table 1 for $A_{patch}$ and $UF$, and 2 – 20 g N dm$^{-3}$ for $c_N$ from Whitehead, 1995).

According to Table 7, whilst the uncertainty originating from the choice of a constant $A_{patch}$ and $UF$ is considerable, the uncertainty coupled with the value of $c_N$ is extremely large. Although the model shows a large uncertainty associated with $c_N$, the close agreement between GAG_field and the measurements (Fig. 9) suggests that using the same average value in every time step well represents reality. In the following, the reasons of this high uncertainty associated with $c_N$ is further examined. For this purpose, randomized $c_N$ time series were generated as described in Section 3.3.2 and using these simulations were

performed with GAG_field.

The ensemble of the simulations derived in this way can be seen in Fig. 15. In both years the largest uncertainty occurred at the peaks of the $NH_3$ fluxes. Overall, however, the uncertainties observed in Fig. 15 are much smaller than was suggested by the sensitivity analysis presented above (Table 7). This is because in the sensitivity analysis the two extremes of $c_N$ were tested, whilst the $c_N^{Ave}$ values generated from the log-normal distribution of $c_N$ resulted in a value close to 11 g N dm$^{-3}$ applied in the

30 baseline simulation with GAG_field.

**4.2.6 Sensitivity to a constant soil pH**

From the point of view of future application of the model for regional scale, computational time could be saved if a constant soil pH over the whole time period could be assumed instead

of simulating soil pH dynamically for every urine patch deposited in the different time steps. To investigate the effect of such

a simplification the baseline simulation with GAG_field was performed with a constant soil pH of 7.5 (GAGf_pH7.5). The reason for selecting this value, is that this is the approximate value where the curve of soil pH flattens out in the case of every urine patch deposited in the baseline simulations in GAG_field (Fig. 12).

With a fixed value of pH 7.5, the model produced a similar temporal variation in $NH_3$ flux as with the dynamically changing soil pH in the baseline simulation with GAG_field (Fig. 16), following relatively closely the fluxes in the baseline simulations.

The model was tested with further two constant soil pH values, 7.0 and 8.0 in the experiments GAGf_pH7.0 and GAGf_pH8.0, respectively. These simulations resulted in highly different $NH_3$ exchange fluxes compared to those in the baseline simulations, especially in the case of GAGf_pH8.0 (Fig. 16).

Although the results from GAGf_pH7.5 suggest a possible simplification of the model for larger scale application, GAGf_8.0 and GAGf7.0 implied that the $NH_3$ exchange fluxes are sensitive to the chosen constant value of soil pH. In GAGf_pH7.5 that

value was applied where the soil pH stabilized under a patch after the intense urea hydrolysis stopped. However, this value might not be the same in every situation. For example, in the case of the baseline experiment with GAG_patch the curve of soil pH flattened out around pH 7 (Móring et al., 2016). Therefore, further considerations are needed regarding the choice of a constant soil pH, which may also be expected to vary with soil type.

**5 Discussion**

**5.1 Model development and evaluation**

The main source of $NH_3$ emission from grazed fields - as mentioned above - is the urine patches (Laubach et al., 2013, Petersen et al., 1998). The GAG model (referred to as GAG_patch in this study), was constructed for a single urine patch by Móring et al. (2016). GAG_patch is capable of simulating the TAN and the water content of the soil under a urine patch and the variation of soil pH. At a larger scale, over a grazed field, $NH_3$ exchange is determined by the coupled effect of $NH_3$ emission from the

urine patches and $NH_3$ exchange with the area of the field that is not affected by urine (non-urine area). Therefore, in this study GAG_patch was extended and applied at the field scale, by employing it for the urine patches and using a modified version of it for the non-urine area.

As shown by Móring et al. (2016), the simulations with GAG_patch with the incorporation of an assumed restart of urea hydrolysis and $CO_2$ emission resulted in a considerably better representation of the measurements than in the baseline

simulation, where these processes were excluded. However, the assumptions for the restart of urea hydrolysis and $CO_2$ emission were hypothetical or specific for the experimental site. For a general model application these processes would need

to be further investigated. Therefore, the possible restart of urea hydrolysis and $CO_2$ emission were concluded not to be implemented to GAG_field.

Regarding the model structure and functionality, Móring et al. (2016) provided a comparison for GAG_patch with the earlier modelling studies for urea affected soils (Sherlock and Goh, 1985, Rachhpal and Nye, 1986) and urine patches (Laubach et al., 2012). Its field scale application, GAG_field is novel among the field scale $NH_3$ exchange models, considering its dynamic approach for the modelling of soil pH under the urine patches. For the same purpose as GAG_field, the PaSim ecosystem model by Riedo et al. (2002) and the VOLT'AIR model by Génermont and Cellier (1997) could be used, the latter simulating $NH_3$ emission related to fertilizer and manure application. Both of these models, however, treat pH as a constant over the whole modelled area and do not account for the characteristics of the temporal development of the $NH_3$ emission form the individual urine patches. Furthermore, the framework of VOLT'AIR is more complex and requires more input data. Thus, for grazing situations, it is much easier to adapt GAG_field.

The ultimate goal of the development of GAG_field was to construct a modelling tool that could be applied to regional (i.e. national or continental) scale. Thus, simplicity was a key aspect of the model development, avoiding extra steps through model simplification during the up-scaling. For this reason, GAG_field operate with a single soil layer, neglecting the exchange of TAN and the movement of water between the soil layers. Even though the models mentioned above (Génermont and Cellier, 1997, Riedo et al., 2002) apply a more sophisticated, multi-layer approach for the soil, the model code of GAG_field enables the addition of new modules. For instance, a multi-layer approach for simulating the TAN budget or the water budget in the source layer.

Similarly to GAG_patch, GAG_field also accounts for the influence of meteorological variables on $NH_3$ exchange. This serves as a base of a further study, focusing on the investigation of the meteorological drivers of $NH_3$ exchange over grazed field. Also, in future work, linking GAG_field to an atmospheric chemistry transport model, these meteorological effects can be also explored in relation to $NH_3$ emission, dispersion and deposition on a larger (i.e. regional or global) scale.

Two baseline simulations were performed with GAG_field over two modelling periods based on data measured at Easter Bush, UK. The modelled and observed $NH_3$ fluxes were in a reasonably broad agreement. The formulation of GAG_field allowed us to investigate the $NH_3$ exchange separately for the urine-affected and unaffected areas, as well as for groups of patches deposited in different time intervals. The results suggested that the temporal evolution of the $NH_3$ exchange flux over a grazed field is dominated by the $NH_3$ emission from urine patches and its magnitude is substantially reduced by the simultaneous $NH_3$ deposition the non-urine area. It was also found that the temporal development of $NH_3$ emission can be considerably different in urine patches deposited in different time intervals. Moreover, the $NH_3$ flux over the field in a given day can be largely influenced by urine patches deposited several days earlier.

## 5.2 Sensitivity analysis

It was investigated how the total simulated $NH_3$ exchange flux responds to an assumed change in the model parameters that regulate $NH_3$ exchange over the whole field, as well as the TAN content and water content under the urine patches. A series

of perturbation experiments was carried out for $\Delta z$, $REW$, $\Gamma_{sto}$, $\Gamma_g$, $\beta$, pH($t_0$), $\theta_{fc}$, $\theta_{pwp}$. In addition to these analyses, we examined the uncertainty coupled with the selected value for $A_{ptach}$, $UF$ and $c_N$, and we also tested GAG_field with different constant values of soil pH. Although GAG_field was constructed so that it accounts for the effects of meteorology on the NH$_3$ exchange over a grazed field, the investigation of the influence of the meteorological variables will be the scope of a future study.

**5.2.1. General findings**

The results of the perturbation experiments were compared with those from Móring et al. (2016) for GAG_patch. In general, it can be concluded that the differences in the sensitivity of the two models can originate from three sources: 1) the effect of the non-urine area on the total net NH$_3$ exchange over the whole field, 2) the different response in the total NH$_3$ exchange of the urine patches as a group, and as individual urine patches, and 3) the different soil characteristics at the two experimental

sites, Easter Bush, UK (GAG_field) and Lincoln, NZ (GAG_patch).

For point 1) it was shown in general that if a patch-related parameter ($\Delta z$, $REW$, $\beta$, pH($t_0$), $\theta_{fc}$, $\theta_{pwp}$) is perturbed, even if the resulting change in the total NH$_3$ emission over the urine patches is the same, the percentage difference over the whole field will be larger if the deposition to the non-urine area is stronger. This is because a larger deposition term results in a smaller total net NH$_3$ exchange over the whole field, suggesting a proportionally larger change in the total over the whole field in

response to the perturbation of the given parameter.

Regarding point 2) a 3) additional perturbation experiments were carried out for $\theta_{fc}$, $\theta_{pwp}$, and $\beta$. Overall, these suggest that the sensitivity of the total NH$_3$ exchange of an individual urine patch is similar to the sensitivity of the urine patches as a group if the investigated urine patch is deposited when the water content of the source layer is minimal ($\theta_{pwp}$). However, over a urine patch, the total NH$_3$ exchange can be extremely sensitive to the perturbations of $\theta_{fc}$, $\theta_{pwp}$, $\beta$, if it is deposited shortly after an

event of rain fall (or dew fall), which increases the water content of the source layer at the time of urine deposition. Since in the baseline simulations with GAG_field the source layer was dry most of the time (water content at $\theta_{pwp}$), the sensitivity for the group of urine patches was similar to the sensitivity of most of the individual urine patches deposited over the modelling periods.

The results also showed that difference between the sensitivities to $\theta_{fc}$, $\theta_{pwp}$, and $\beta$ over the urine patches in the GAG_field

simulations and the GAG_patch simulation is associated with the different values of $\theta_{fc}$, $\theta_{pwp}$ at the two experimental sites. Furthermore, the different pH of the undisturbed soil at Lincoln and Easter Bush could lead to high differences in the resulted sensitivities to $\beta$ over the individual urine patches at the two sites.

In conclusion, two main reasons can be identified for the large differences in the observed sensitivity of the total net NH$_3$ exchange to $\theta_{fc}$, $\theta_{pwp}$, and $\beta$ between the baseline simulations with GAG_field and GAG_patch. The differences are caused by

firstly, the fact that over the field scale in the net exchange the deposition to the non-urine area is also included, and secondly, the different soil characteristics at the two sites.

### 5.2.2. Parameter-specific findings

Compared with the other parameters, the total $NH_3$ exchange simulated by GAG_field turned out to be the least sensitive to the changes in $\varDelta z$ and $REW$. For $REW$ GAG_patch showed a similar, negligibly weak response to the ±10%, ±20% modifications (Móring et al., 2016). In the case of $\varDelta z$, $NH_3$ exchange was found to be sensitive to the perturbations of this parameter in both the patch-scale and the field-scale experiments, the latter especially in the P2002 simulation.

Móring et al. (2016) carried out a model analysis in which the possible extreme values of $\varDelta z$ (calculated as the penetration depth of urine and applied from Laubach et al. 2012), which showed a strong response in the simulated $NH_3$ fluxes. However, since the modelled $NH_3$ fluxes were in a broad agreement with the measurements in three different model simulations using the same value of $\varDelta z$, these results suggest that the main governing processes of $NH_3$ emission from urine patches might occur in this thin top soil layer ($\varDelta z = 4$ mm) as assumed by Móring et al. (2016). Nevertheless, future work is needed to confirm this hypothesis, considering how further datasets can help characterize the appropriate thickness of the effective soil emission layer. In GAG_field, the horizontal dispersion of $NH_3$ on the field was neglected, and as such, the homogeneity of $\chi_a$ was assumed. However, the perturbation experiments showed that $\chi_a$ can considerably affect the total $NH_3$ exchange over the non-urine area. This suggests that including the effect of horizontal advection to the model could possibly improve the simulation of $NH_3$ exchange over a grazed field. This effect is treated directly when such a bi-directional model as GAG_field is incorporated into a regional atmospheric chemistry transport model, through the influence of surface emission/deposition on the simulated value of near-surface $\chi_a$.

The constant $\varGamma_{sto}$ and $\varGamma_{soil}$ affect $NH_3$ exchange over the whole field exclusively through its effect on the $NH_3$ exchange over the non-urine area. The results suggested that the model is only slightly sensitive to $\varGamma_{sto}$, whilst $\varGamma_g$ can have a considerable effect on $NH_3$ exchange.

Móring et al. (2016) found only a weak sensitivity of the total $NH_3$ emission to $\beta$. Although the exact same value of $\beta$ was used in GAG_field for both modelling periods as by Móring et al. (2016) in GAG_patch, at the field scale, $NH_3$ exchange was found to be highly sensitive to the same changes in $\beta$. It was shown that the dependence of $NH_3$ exchange on $\beta$ is influenced by the soil pH before urine deposition and also by the maximum amount of urine that can be stored in the source layer. According to the results, in the case of the urine patches with higher initial soil pH and higher initial soil water content, the sensitivity of the total net $NH_3$ exchange to $\beta$ is stronger. However, the good agreement found on the field scale between the modelled and the observed $NH_3$ fluxes in both modelling periods, suggests that the natural variability of $\beta$ might be less than the perturbation applied in the sensitivity analysis. Nevertheless, this requires further experimental investigation.

Móring et al. (2016) showed that the dynamic simulation of soil pH was necessary to represent the first, highest peak in $NH_3$ emission after the deposition of a urine patch. This finding can be refined by the current results, suggesting a strong sensitivity in the $NH_3$ exchange associated with the value of soil pH before the deposition of the urine patch, $pH(t_0)$. In contrast, the results for field scale implied that if the value of soil pH after the intensive urea hydrolysis is chosen as a constant (in the presented

baseline simulations this was pH 7.5) for the whole modelling period, the $NH_3$ fluxes by GAG_field are similar to those derived with the dynamic chemistry approach.

The apparent contradiction between the results for the two scales can be explained by that in the baseline simulations with GAG_field (Fig. 12), in most of the urine patches 1.5-2 days after their deposition the soil pH flattened out at 7.5. This means that in a given hour the total $NH_3$ flux over the whole field was mainly affected by urine patches under which the soil pH was about 7.5. These results from the approach with constant soil pH suggests a way for model simplification when it is applied to larger scales. Nevertheless, further considerations are needed to find a generalized approach that determines the applicable value of a constant soil pH.

The sensitivity analysis for both GAG_patch and GAG_field showed that the highest uncertainties are associated with the water content of soil at $\theta_{fc}$ and $\theta_{pwp}$. The results suggested that the sensitivity of the total $NH_3$ exchange over a urine patch is regulated by the maximum amount of urine that the $NH_3$ source layer can hold, which depends on $\theta_{fc}$ and $\theta_{pwp}$, or if the soil volumetric water content is higher than $\theta_{pwp}$ before a urination event, the initial water content of the soil ($\theta(t_0)$). It was found that in the case of a higher initial soil water content (i.e. less urine in the source layer), $NH_3$ exchange was more sensitive to the changes in $\theta_{fc}$ and $\theta_{pwp}$.

The broad agreement between the simulated and measured $NH_3$ fluxes suggests that the uncertainty of the measurement of $\theta_{fc}$ and $\theta_{pwp}$ might be less than the perturbations applied in the sensitivity analysis ($\pm 10\%$, $\pm 20\%$). However, a regional scale model application would require $\theta_{fc}$ and $\theta_{pwp}$ values over a high-resolution grid, which is likely to be coupled with higher uncertainties. Therefore, at regional scale model application, the uncertainty of the input $\theta_{fc}$ and $\theta_{pwp}$ datasets has to be assessed when the model results are evaluated.

For the presented simulations with GAG_field a hypothetical grazing situation was assumed, in which there is no temporal variation in $UF$, $c_N$ and $A_{patch}$. However, $UF$, $c_N$ and the volume of urine deposited by an animal can have a diurnal cycle (Misselbrook et al., 2016), latter with a potential effect on $A_{patch}$ (Li et al., 2012). In addition to these parameters, $LAI$ and $h$ was handled as constant for the whole modelling period, whilst these parameters are decreasing since due to grazing, as there is less and less grass on the field toward the end of the modelling period. To assess the possible influence of these assumptions on $\Sigma F_{net}$, additional sensitivity experiments were performed with GAG_field.

According to the results, whilst the uncertainty originating from the choice of a constant $A_{patch}$ and $UF$ is considerable, the uncertainty coupled with the value of $c_N$ is extremely large. Nevertheless, model simulations with randomized N concentrations implied that this uncertainty might be considerably smaller in reality than it was suggested by the sensitivity analysis. For $LAI$ and $h$, it was found, that LAI has a negligible effect on $\Sigma F_{net}$, whereas $h$ can substantially affect the $NH_3$ exchange over the field. Therefore, future work should investigate how the modelled $NH_3$ exchange responds when a real grazing situation assumed, including a diurnal cycle of $UF$, $c_N$ and $A_{patch}$ as well as temporal changes of $LAI$ and $h$.

# 6 Conclusions

In this study the GAG model (Móring et al., 2016) for simulating $NH_3$ emission from individual urine patches was extended and applied for field scale. The new, field-scale model (GAG_field) was tested over two modelling periods for a grazed grassland at Easter Bush, UK. Comparison with micrometeorological $NH_3$ flux measurements showed that the model reproduced the main features of the observed fluxes.

The simulations indicated that the temporal evolution of the $NH_3$ exchange flux over a grazed field is dominated by the $NH_3$ emission from the urine patches, which is substantially decreased by the simultaneous $NH_3$ deposition the non-urine area. The results presented also showed that the evolution of $NH_3$ emission from urine patches deposited in different time steps can be substantially different and that $NH_3$ fluxes in a given day can be considerably affected by urine patches deposited several days earlier.

The sensitivity analysis to the regulating model parameters showed that the total $NH_3$ flux modelled by GAG_field is highly sensitive to the buffering capacity ($\beta$), the field capacity ($\theta_{fc}$) and the permanent wilting point ($\theta_{pwp}$). The observed sensitivities turned out to be much higher than was found in the case of GAG_patch. The reason for these different sensitivities is dual. Firstly, the difference originates from the different scales. When a model parameter, affecting the $NH_3$ emission from the urine patches is perturbed, the resulting change in the total net $NH_3$ exchange over the whole field will be larger compared to that in the total $NH_3$ emission from the urine patches. The reason for this is the negative deposition term in GAG_field over the non-urine area. Secondly, and more importantly, the different sensitivities observed for the two models can be explained by the environmental circumstances at the two sites the model was applied for, i.e the different initial soil pH and the different soil physical characteristics at the two sites which determine the maximum volume of urine that can be stored in the $NH_3$ source layer. It was found that in the case of urine patches with a higher initial soil pH and higher initial soil water content, the sensitivity of $NH_3$ exchange to $\beta$ was stronger. Also, in the case of a higher initial soil water content, $NH_3$ exchange was more sensitive to the changes in $\theta_{fc}$ and $\theta_{pwp}$.

The sensitivity analysis also showed that the nitrogen content of urine ($c_N$) is associated with a high uncertainty. However, model experiments based on $c_N$ values randomized from an estimated statistical distribution, implied that this uncertainty might considerably smaller in practice.

Finally, GAG_field was tested with a constant soil pH of 7.5 to see how well a simpler model structure could perform, such for a regional scale application. The variation of $NH_3$ fluxes simulated in this way showed a broad agreement with those from the baseline simulations with GAG_field that accounts for a dynamically changing soil pH. Although there were differences in the detailed time-course of emissions, the overall patterns and magnitude of $NH_3$ emissions were similar. These results suggest a way for model simplification when GAG_field is applied later for regional scale. However, since the $NH_3$ exchange fluxes showed a large sensitivity to the value of the applied constant soil pH, further examinations are needed, concerning the choice of this constant value in realtion to difference in underlying soil conditions.

**Acknowledgement**

This work was carried out within the framework of the ÉCLAIRE project (Effects of Climate Change on Air Pollution and Response Strategies for European Ecosystems) funded by the EU's Seventh Framework Programme for Research and Technological Development (FP7) and with matching National Capability funds from the UK Natural Environment Research Council through the Centre for Ecology & Hydrology.

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

**Table 1. Ranges of the parameters used in the calculation of the urine–covered proportion of a field with an area of 1 ha (= 10 000 m².).**

| Animal | | Sheep | Cattle | Reference |
|---|---|---|---|---|
| Number of animals on $A_{field}$ | | 1 – 100 | 0.1 – 10 | EC, 2015 |
| Urination frequency (urination animal$^{-1}$ day$^{-1}$) | ($UF$) | 15 – 20 | 8 – 12 | Whitehead, 1995 |
| Patches deposited per day ($N_t$) | | 15 - 2 000 | 0.8 – 120 | - |
| Patch area ($A_{patch}$) (m²) | | 0.043 - 0.055 | 0.38 - 0.42 | Williams and Haynes, 1994 |

**Table 2. Urine, soil and site specific constants used in the evaluation of GAG_field. The source of the values that were not measured at the site are also indicated. P2002 and P2003 stand for the modelling periods in 2002 and 2003, respectively. Constants used in the model, but not mentioned here were kept the same as defined for the baseline simulation with GAG_patch (Móring et al., 2016).**

| Model constants | Value | Source (if not measured) |
|---|---|---|
| **Urine specific constants** | | |
| $A_{patch}$ (area of a urine patch) | 40 dm$^2$ | Williams and Haynes, 1994 (average value) |
| $c_N$ (nitrogen content of urine) | 11 g N dm$^{-3}$ | Whitehead, 1995 (average values) |
| $W_{urine}$ (volume of urine) | 2.5 dm$^3$ | |
| **Soil specific constants** | | |
| $\theta_{fc}$ (field capacity) | 0.37 | |
| $\theta_{pwp}$ (permanent wilting point) | 0.192 | |
| $\theta_{por}$ (porosity) | 0.54 | |
| $pH(t_0)$ (initial soil pH) | 4.95 | |
| $\Gamma_g$ (soil emission potential) | 3000 | Modelled (Section 3.2.3) |
| $\theta(t_0)$ (initial volumetric water content) | 0.356 (P2002) 0.24 (P2003) | |
| **Site specific constants** | | |
| Latitude | 55.87° | |
| Longitude | 3.03° | |
| Height above sea level | 190 m | |
| $A_{field}$ (field area) | 5.424 ha | |
| $\Gamma_{sto}$ (stomatal emission potential) | 500 | Massad et al., 2010 (average value) |
| $UF$ (urination frequency) | 10 animal$^{-1}$ day$^{-1}$ | Whitehead, 1995 (average values) |
| $z_w$ (height of wind measurement) | 1 m | |
| Number of cattle on the field | 40, 17 (P2002)[c] 50, 52 (P2003)[c] | |
| $z$ (heights of NH$_3$ concentration measurements) | 0.44 m, 0.96 m, 2.06 m | |

[a]There was no measurement in P2002, therefore, the average of the measurements for P2003 was used.

[b]The value was measured on 23/06/2003.

[c]The date when the number of animals changed in P2002 and P2003 were 28/08/2002 and 23/06/2003, respectively.

**Table 3. Results of the perturbation experiments with GAG_field. The changes in the total NH₃ flux over the field as a response to a change (±10% and ±20%) in the listed model parameters where expressed as the percentage of the total NH₃ exchange in the baseline simulations with GAG_field and in the brackets as the hourly change in the total net exchange over the whole field (g N hr⁻¹). Results are listed for both modelling periods, P2002 and P2003, separately for the whole field (Sens_net) and the urine patches (Sens_patch). As a comparison, the results of the sensitivity analysis carried out by Móring et al. (2016) for GAG_patch are also indicated (Sens_patch^single). In the column 'Effect' the letters denote how the given parameters affect total NH₃ exchange in GAG_field: through the urine patches (P) or the non-urine area (N) or both.**

| Constants (x) | Effect | Δx | Change in the total net flux in response to the perturbation | | | | |
| --- | --- | --- | --- | --- | --- | --- | --- |
| | | | P2002 | | P2003 | | GAG_patch |
| | | | Sens_net | Sens_patch | Sens_net | Sens_patch | Sens_patch^single |
| *Δz* (thickness of the source layer) | P | -20% | -40% (-0.26) | -7% | -8% (-0.27) | -4% | -12% |
| | | -10% | -18% (-0.11) | -3% | -4% (-0.12) | -2% | -6% |
| | | +10% | +14% (+0.09) | +2% | +2% (+0.08) | +1% | +5% |
| | | +20% | +25% (+0.16) | +4% | -2% (-0.06) | -1% | +11% |
| *REW* (readily evaporable water) | P | -20% | 0% (0.0) | 0% | -3% (-0.08) | -1.3% | -3% |
| | | -10% | 0% (0.0) | 0% | -1% (-0.04) | -0.6% | -2% |
| | | +10% | 0% (0.0) | 0% | +1% (+0.04) | +0.6% | +2% |
| | | +20% | 0% (0.0) | 0% | +2% (+0.08) | +1.2% | +4% |
| *pH(t₀)* (initial soil pH) | P | -20% | -173% (-1.11) | -31% | -79% (-2.53) | -38% | - |
| | | -10% | -90% (-0.58) | -16% | -42% (-1.36) | -20% | - |
| | | +10% | +96% (+0.61) | +17% | +48% (+1.53) | +23% | - |
| | | +20% | +196% (+1.25) | +35% | +100% (+3.21) | +48% | - |
| *Γ_sto* (stomatal emission potential) | N | -20% | -3% (-0.02) | - | -1% (-0.02) | - | - |
| | | -10% | -1% (-0.01) | - | -0.3% (-0.01) | - | - |
| | | +10% | +1% (+0.01) | - | +0.3% (+0.01) | - | - |
| | | +20% | +3% (0.02) | - | +1% (+0.02) | - | - |

| Constants (x) | Effect | $\Delta x$ | P2002 $Sens_{net}$ | P2002 $Sens_{patch}$ | P2003 $Sens_{net}$ | P2003 $Sens_{patch}$ | GAG_patch $Sens_{patch}^{single}$ |
|---|---|---|---|---|---|---|---|
| $\Gamma_g$ (soil emission potential) | N | -20% | -54% (-0.34) | - | -12% (-0.38) | - | - |
| | | -10% | -27% (-0.17) | - | -6% (-0.19) | - | - |
| | | +10% | +27% (+0.17) | - | +6% (+0.19) | - | - |
| | | +20% | +54% (+0.34) | - | +12% (+0.38) | - | - |
| $\beta$ (soil buffering capacity) | P | -20% | +94% (+0.60) | +17% | +50% (+1.61) | +24% | +1% |
| | | -10% | +46% (+0.29) | +8% | +24% (+0.77) | +11% | +1% |
| | | +10% | -43% (-0.28) | -8% | -22% (-0.69) | -10% | -1% |
| | | +20% | -84% (-0.53) | -15% | -41% (-1.31) | -20% | -1% |
| $\theta_{fc}$ (field capacity) | P | -20% | -360% (-2.30) | -64% | -153% (-4.88) | -72% | -18% |
| | | -10% | -190% (-1.22) | -34% | -85% (-2.71) | -40% | -7% |
| | | +10% | +211% (+1.35) | +37% | +96% (+3.07) | +46% | +6% |
| | | +20% | +448% (+2.86) | +79% | +191% (+6.09) | +91% | +9% |
| $\theta_{pwp}$ (permanent wilting point) | P | -20% | +364% (+2.32) | +64% | +157% (+5.03) | +75% | +9% |
| | | -10% | +173% (1.11) | +31% | +76% (+2.43) | +36% | +5% |
| | | +10% | -156% (-1.00) | -28% | -65% (-2.07) | -31% | -4% |
| | | +20% | -292% (-1.87) | -52% | -118% (-3.79) | -56% | -9% |
| $\chi_a$ (ambient atmospheric $NH_3$ concentration)* | P, N | -20% | +166% (+1.06) | +0.3% (+0.012) | +36% (+1.16) | +0.3% (+0.02) | - |
| | | -10% | +83% (+0.53) | +0.2% (+0.006) | +18% (+0.58) | +0.2% (+0.01) | - |
| | | +10% | -84% (-0.53) | -0.2% (-0.006) | -19% (-0.61) | -0.2% (-0.01) | - |
| | | +20% | -167% (-1.07) | -0.3% (-0.012) | -38% (-1.22) | -0.3% (-0.02) | - |

| Constants (x) | Effect | Δx | Change in the total net flux in response to the perturbation | | | | GAG_patch |
| | | | P2002 | | P2003 | | |
| | | | $Sens_{net}$ | $Sens_{patch}$ | $Sens_{net}$ | $Sens_{patch}$ | $Sens_{patch}^{single}$ |
| LAI (leaf area index) | P, N | -20% | -1.1% (-0.007) | +0.11% (+0.004) | +0.10% (+0.003) | +0.15% (+0.010) | - |
| | | -10% | -0.5% (-0.003) | +0.05% (+0.002) | +0.05% (+0.002) | +0.07% (+0.005) | - |
| | | +10% | +0.5% (+0.003) | -0.05% (-0.02) | -0.05% (-0.002) | -0.07% (-0.005) | - |
| | | +20% | +1.1% (+0.007) | -0.11% (-0.004) | -0.10% (-0.003) | -0.14% (-0.010) | - |
| h (canopy height) | P, N | -20% | -12% (-0.08) | -8% (-0.28) | -9% (-0.28) | -8% (-0.51) | - |
| | | -10% | -6% (-0.04) | -4% (-0.14) | -4% (-0.13) | -4% (-0.25) | - |
| | | +10% | +4% (+0.03) | +4% (+0.13) | +4% (+0.14) | +4% (+0.26) | - |
| | | +20% | +6% (+0.04) | +7% (+0.26) | +8% (+0.26) | +7% (+0.49) | - |

*In both P2002 and P2003 $\chi_a$ was changed by ±10% and ±20% of the average $\chi_{air}$ over each period as explained in Section 3.3.1.

**Table 4. Results from simulations with GAG_patch, testing the effect of pH(t₀) (initial soil pH), $\theta_{fc}$ (field capacity) and $\theta_{pwp}$ (permanent wilting point) on the sensitivity of the total NH₃ emission to β (buffering capacity). Input data were applied from the baseline simulation with GAG_patch (Móring et al., 2016), except for the parameters denoted in the table with a different font style. Bold values are taken from the input data for the baseline simulations with GAG_field, and italics denote a situation when the water content was assumed to be halfway between the field-scale values of $\theta_{fc}$ and $\theta_{pwp}$. The sensitivity was expressed as the percentage difference in the original NH₃ emission derived with the given model settings with GAG_patch (listed also in the table for every model experiment).**

| Model experiment | Model settings | | | Original emission (g N) | Response of emission to a change in β by | | | |
|---|---|---|---|---|---|---|---|---|
| | $pH(t_0)$ | $\theta_{fc}$ | $\theta_{pwp}$ | | -20% | -10% | +10% | +20% |
| A | **4.95** | 0.40 | 0.10 | 1.5 g | +5% | +2% | -2% | -5% |
| B | 6.65 | **0.37** | **0.19** | 0.9 g | +3% | +1% | -1% | -2% |
| C | **4.95** | **0.37** | **0.19** | 0.6 g | +11% | +5% | -5% | -10% |
| D | **4.95** | **0.37** | *0.28* | 0.1 g | +42% | +18% | -16% | -30% |

**Table 5. The maximum space in the NH$_3$ source layer that can be filled by the incoming liquid ($\theta_{urine}$) in the baseline experiments with GAG_patch and GAG_field, and the percentage it changes when $\theta_{fc}$ (field capacity) and $\theta_{pwp}$ (permanent wilting point) are modified by ±10% and ±20%.**

| Scale | $\theta_{urine}$ | Percentage difference in $\theta_{urine}$ as a response to a change in | | | |
| | | $\theta_{pwp}$ | | $\theta_{fc}$ | |
| | | ±10% | ±20% | ±10% | ±20% |
|---|---|---|---|---|---|
| **GAG_patch** | 0.3 | ±3% | ±6% | ±13% | ±26% |
| **GAG_field** | 0.18 | ±11% | ±22% | ±21% | ±42% |

**Table 6. Model results from model experiments with GAG_patch, testing the effect of the initial water content of the soil ($\theta(t_0)$) on the model sensitivity to $\theta_{fc}$ (field capacity) and $\theta_{pwp}$ (permanent wilting point). Input data were applied from the baseline simulation with GAG_patch, except for $\theta_{fc}$ and $\theta_{pwp}$, which were applied from the baseline simulation with GAG_field, and $\theta(t_0)$, which was modified in the simulations as stated below. The sensitivity was expressed as a percentage difference in the original NH$_3$ emission (listed also in the table for every model experiment).**

| Parameter tested (x) | Model setting $\theta(t_0)$ | Original emission (g N) | Response of emission to a change in x by | | | |
|---|---|---|---|---|---|---|
| | | | -20% | -10% | +10% | +20% |
| $\theta_{fc}$ | $\theta_{pwp}$ | 0.9 g | -41% | -20% | +18% | +31% |
| | 0.28 | 0.4 g | -90% | -47% | +45% | +81% |
| $\theta_{pwp}$ | $\theta_{pwp}$ | 0.9 g | +33% | +16% | -16% | -31% |
| | 0.28 | 0.4 g | +67% | +33% | -31% | -58% |

**Table 7. Results from the baseline simulations with GAG_field when the maximum and minimum was applied of the investigated parameters. In every simulation the difference in the total NH$_3$ exchange was derived, expressed as the percentage of the total exchange in the baseline simulations with GAG_field.**

| Parameters | Min/Max | Change in the total NH$_3$ exchange | |
|---|---|---|---|
| | | P2002 | P2003 |
| $A_{patch}$ (dm$^2$) | 38 | -9% | +11% |
| | 40 | +9% | -11% |
| $c_N$ (g N dm$^{-3}$) | 2 | -187% | -211% |
| | 20 | +292% | +403% |
| *UF* (urination animal$^{-1}$ day$^{-1}$) | 8 | -38% | -42% |
| | 12 | +38% | +42% |

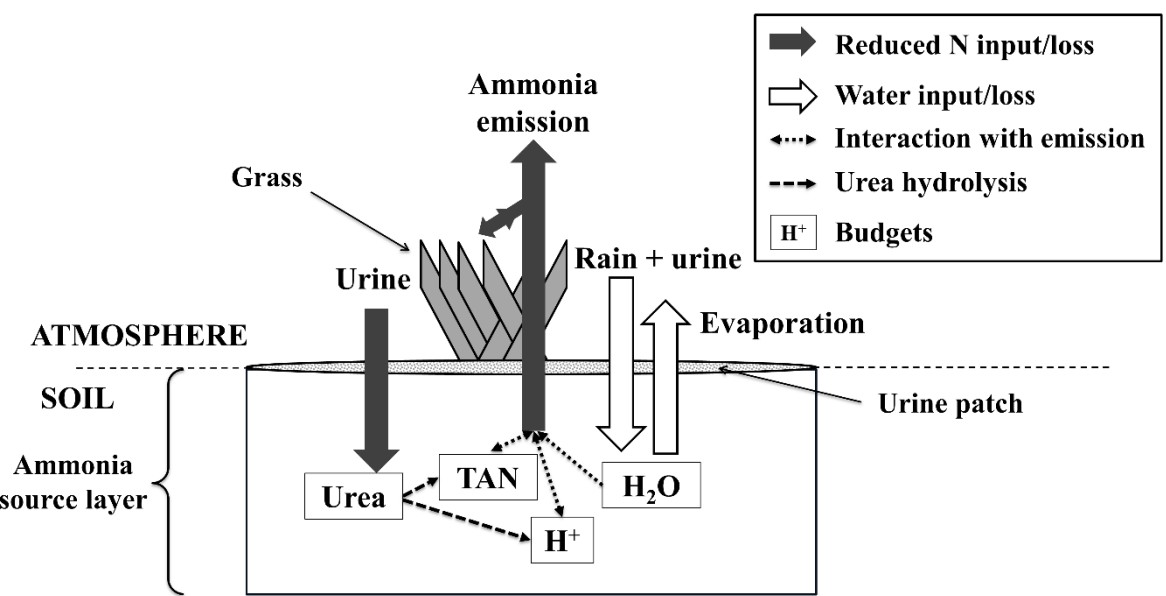

**Figure 1. Simplified schematic of the GAG model by Móring et al. (2016), referred to as GAG_patch in this study.**

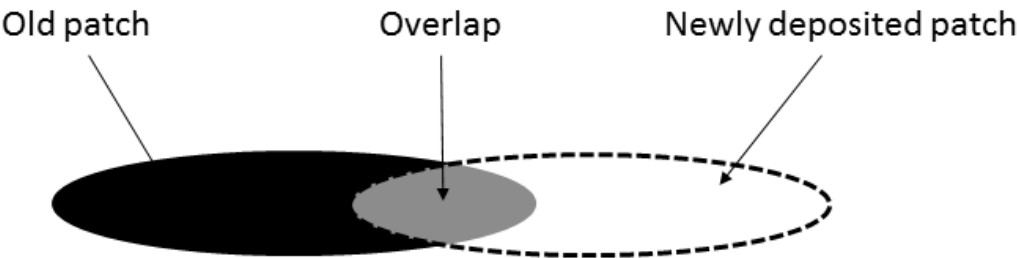

**Figure 2. Difference in the chemical composition of the soil in two urine patches deposited at different times. The different colours of the old and the newly deposited urine patches (black and white, respectively) as well as the overlap between them (grey) show the different soil chemical properties in the different areas.**

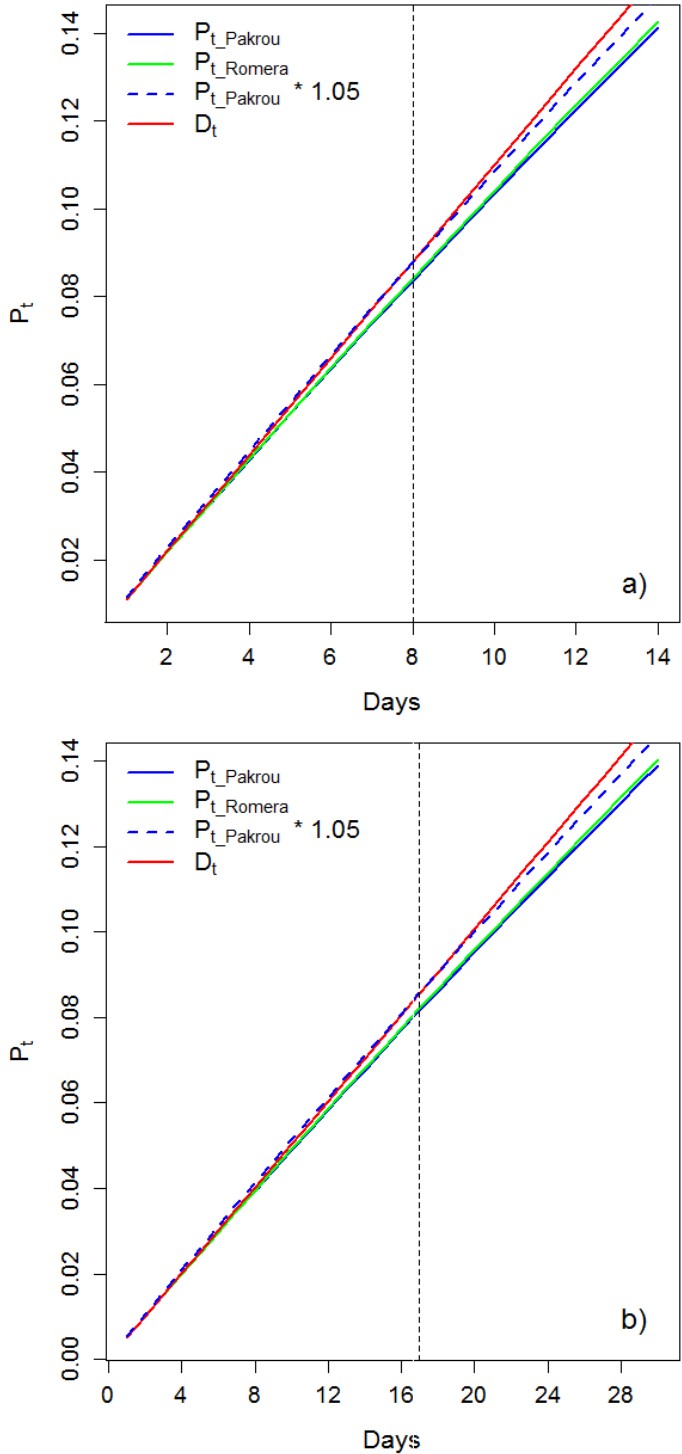

**Figure 3: Proportion of the field covered by urine patches ($P_t$) calculated for sheep (a)) and cattle (b)) as suggested by Pakrou and Dillon (2004) ($P_{t\_Pakrou}$), Romera et al. (2012) ($P_{t\_Romera}$) and when there is no overlap between the patches ($D_t$).**

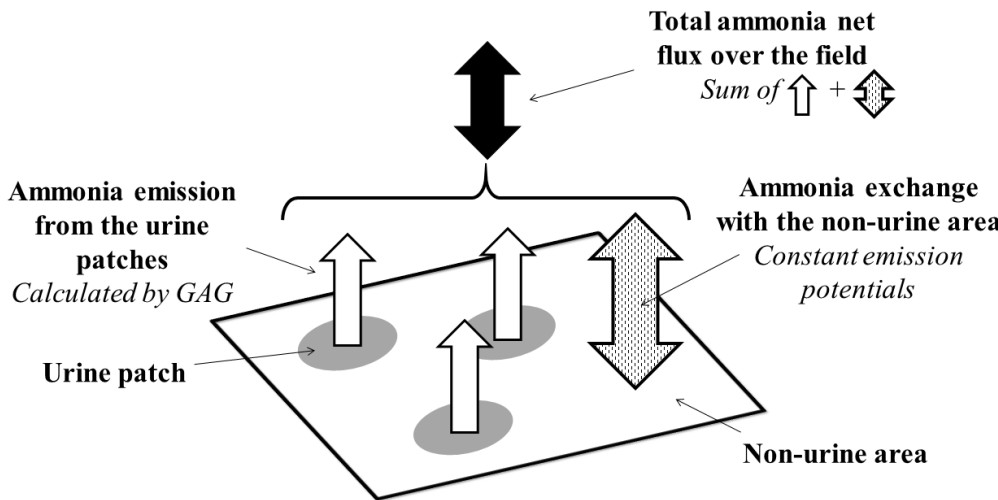

**Figure 4. The schematic of GAG_field. The figure depicts the components of the total net NH₃ flux over the field: NH₃ emission from the urine patches and the NH₃ exchange with the non-urine area.**

| | | NH$_3$ emission from the urine patches | | | | | NH$_3$ exchange with the non-urine area |
|---|---|---|---|---|---|---|---|
| | | Time of the deposition of the urine patches | | | | | |
| | | $t_{j=1}$ | $t_{j=2}$ | $t_{j=3}$ | ... $t_{j=n}$ | | |
| Time since the beginning of the modelling period | $t_{i=1}$ | $F_{patch}^{j=1}(t_{i=1})$ | $F_{non}(t_{i=1})$ | $F_{non}(t_{i=1})$ | ... | $F_{non}(t_{i=1})$ | $F_{non}(t_{i=1})$ |
| | $t_{i=2}$ | $F_{patch}^{j=1}(t_{i=2})$ | $F_{patch}^{j=2}(t_{i=2})$ | $F_{non}(t_{i=2})$ | ... | $F_{non}(t_{i=2})$ | $F_{non}(t_{i=2})$ |
| | $t_{i=3}$ | $F_{patch}^{j=1}(t_{i=3})$ | $F_{patch}^{j=2}(t_{i=3})$ | $F_{patch}^{j=3}(t_{i=3})$ | ... | $F_{non}(t_{i=3})$ | $F_{non}(t_{i=3})$ |
| | ... | ... | ... | ... | ... | ... | ... |
| | $t_{i=m}$ | $F_{patch}^{j=1}(t_{i=m})$ | $F_{patch}^{j=2}(t_{i=m})$ | $F_{patch}^{j=3}(t_{i=m})$ | ... | $F_{patch}^{j=n}(t_{i=m})$ | $F_{non}(t_{i=m})$ |
| Number of patches deposited in $t_j$ | | $n(t_{j=1})$ | $n(t_{j=2})$ | $n(t_{j=3})$ | ... | $n(t_{j=m})$ | 0 |

**Figure 5. Schematic for the temporal development of NH$_3$ fluxes (in every $i^{th}$ time step, $t_i$) as derived by GAG_field. $F_{patch}^{j}(t_i)$ stands for the NH$_3$ flux from the urine patches deposited in the $j^{th}$ time step ($t_j$), and $F_{non}(t_i)$ stands for the NH$_3$ flux from the non-urine area. The bottom row shows how many urine patches were deposited in the given $j^{th}$ time step ($n(t_j)$). Fluxes with striped background are calculated by GAG_patch, and the fluxes with clear background are calculated by a modified version of GAG_patch for non-urine area (explained in the text).**

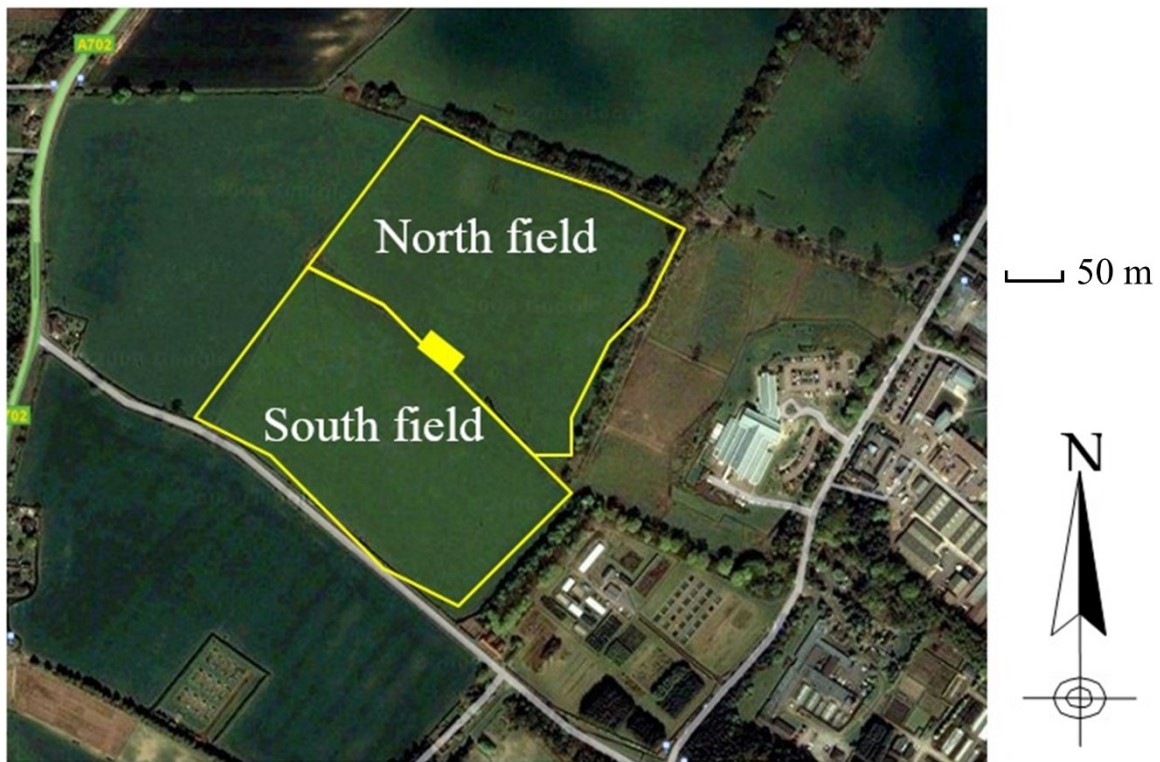

**Figure 6. Satellite photo of the Easter Bush site. The map was generated by Google Maps, indicating the two halves of the field and the place of the instruments on the border of the two denoted by the small yellow rectangle. (The figure is taken from the metadata file by CEH.)**

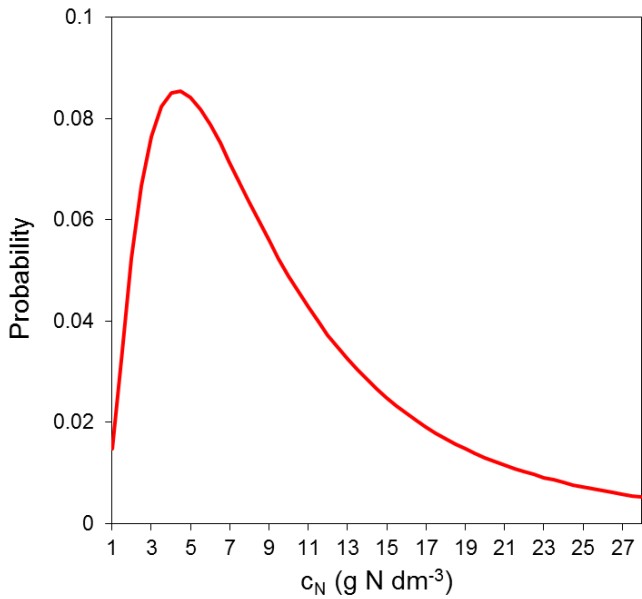

**Figure 7. Probability density function of the log-normal distribution generated for the distribution of the nitrogen content of urine ($c_N$). The scale parameters are $\sigma$ = 0.786 and $\mu$ = 2.089.**

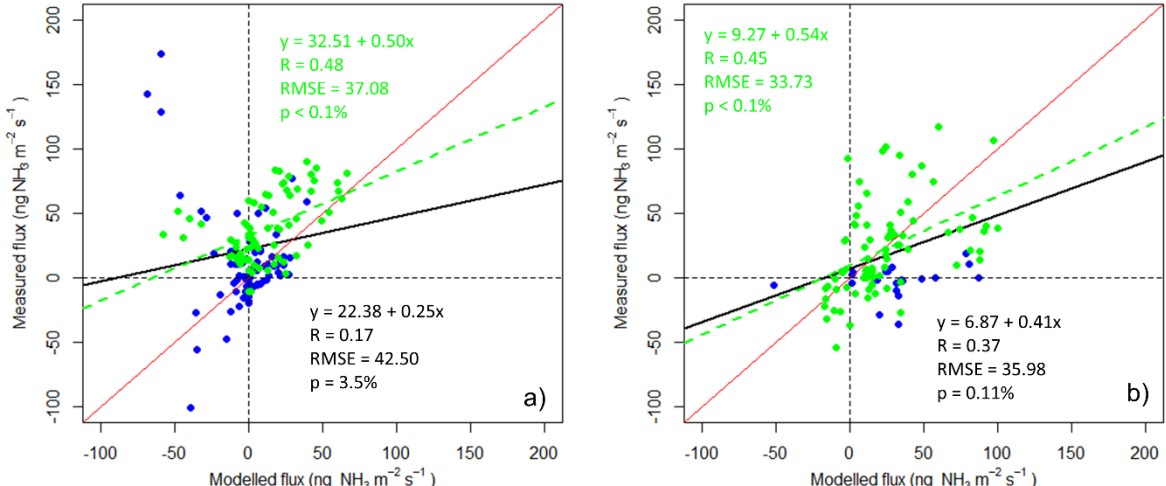

**Figure 8. NH₃ fluxes simulated by GAG_field against the measured NH₃ fluxes in P2002 (a) and P2003 (b). Green and blue dots represent the data for all time steps when measured fluxes were available. The green dots indicate only those time steps in which the measured flux was considered robust as shown in Fig. 9 (on Fig. a, the remaining data points on 27/08/2002 were also excluded as explained in Section 4.1.1). The figures show the fitted lines to the data points (thick black line for all of the data points, green dashed line for the green data points) in comparison with the 1:1 line (red line). The statistics indicated are the equation of the fitted lines (y), the Pearson correlation (R), the relative mean squared error (RMSE) and the level of significance of the relationship between the measured and modelled values (p) in the colour of the corresponding fitted line.**

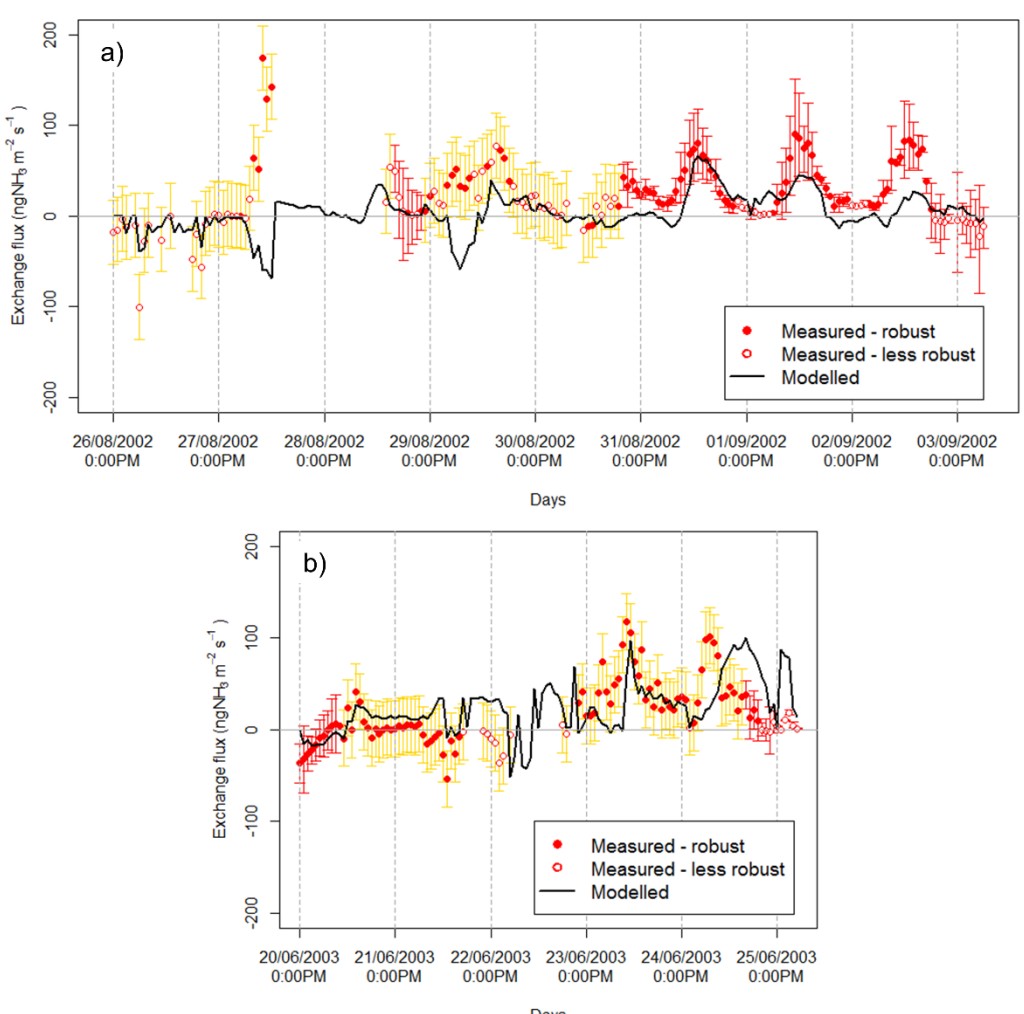

**Figure 9. Comparison of the measured and modelled NH₃ fluxes in the modelling periods P2002 (a) and P2003 (b). The uncertainty of the flux measurements is depicted as error bars. Yellow error bars indicate the cases where one of the three NH₃ concentration denuders were malfunctioning or not registering data at all. For these, the error was estimated as the average of the observed errors (red error bars) multiplied by an arbitrary factor of two. A measured flux was considered to be robust if it met the criteria of the quality control for low wind speed and strong stability as described in Section 3.2.2.**

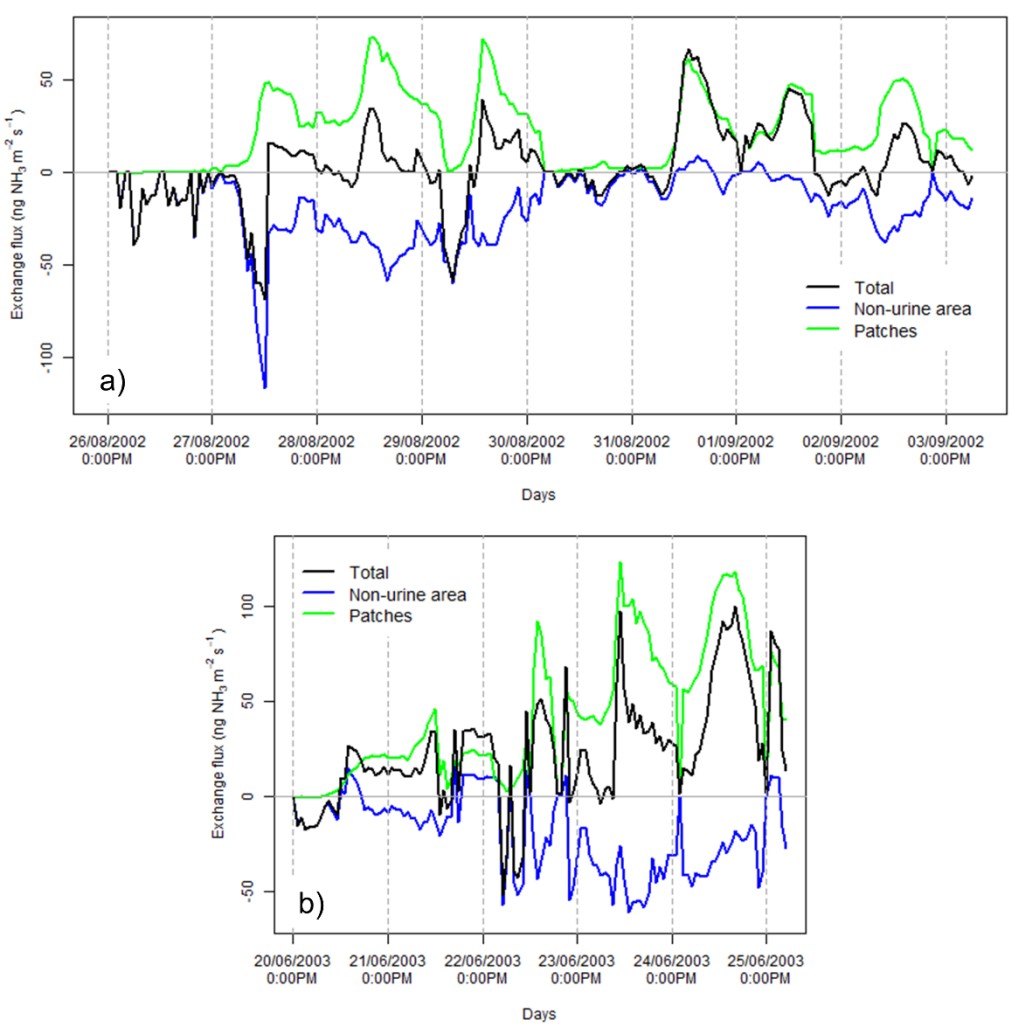

**Figure 10. Simulated NH₃ exchange fluxes over the urine patches, the non-urine area and the whole field in the modelling periods P2002 (a) and P2003 (b).**

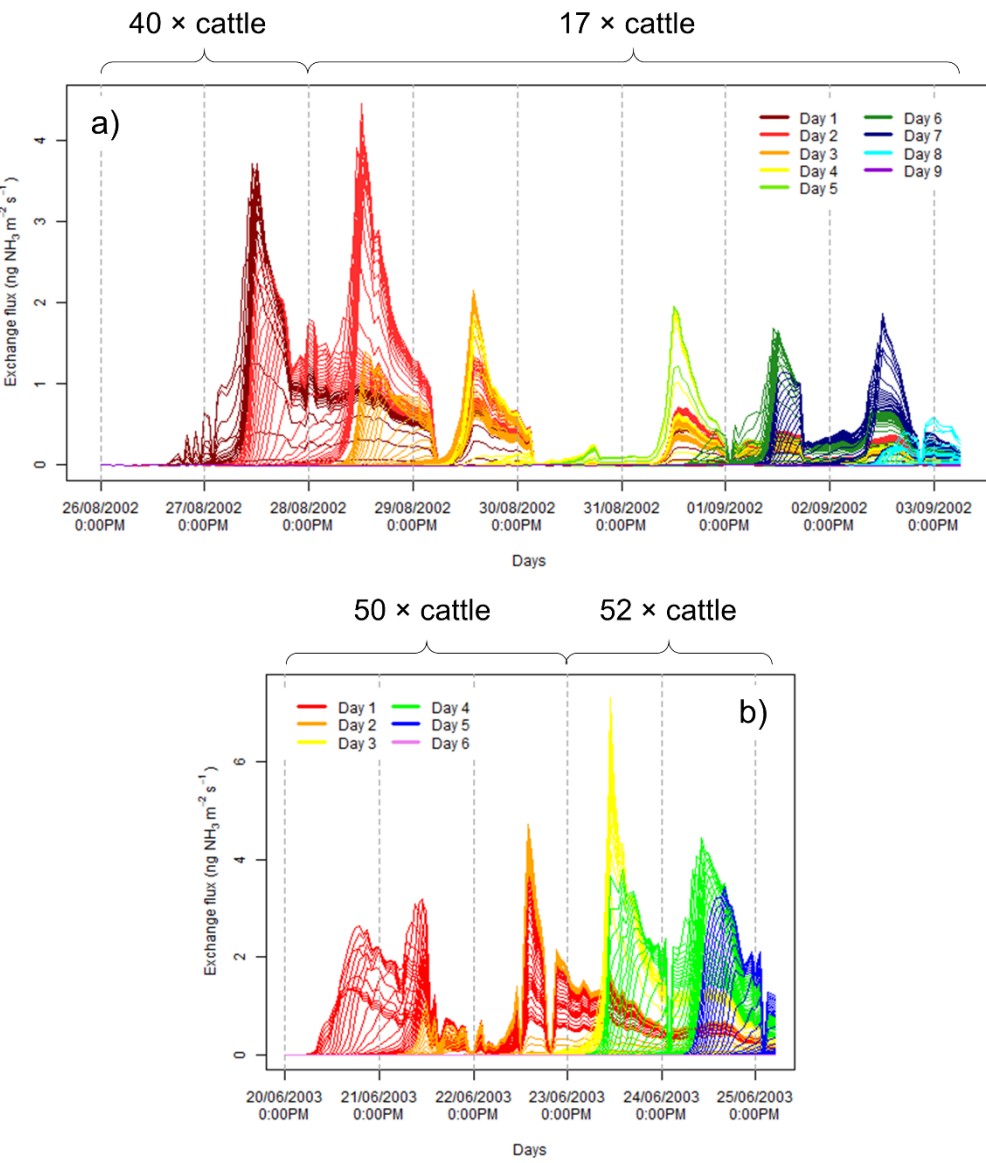

**Figure 11. Simulated NH₃ fluxes from urine patches deposited in the same time step in the modelling periods P2002 (a) and P2003 (b). Each line indicates NH₃ fluxes from urine patches deposited in a given time step (expressed for the whole field), while the different colours indicate the days of the urination events. The number above the plots show how many cattle were grazing in the given time intervals.**

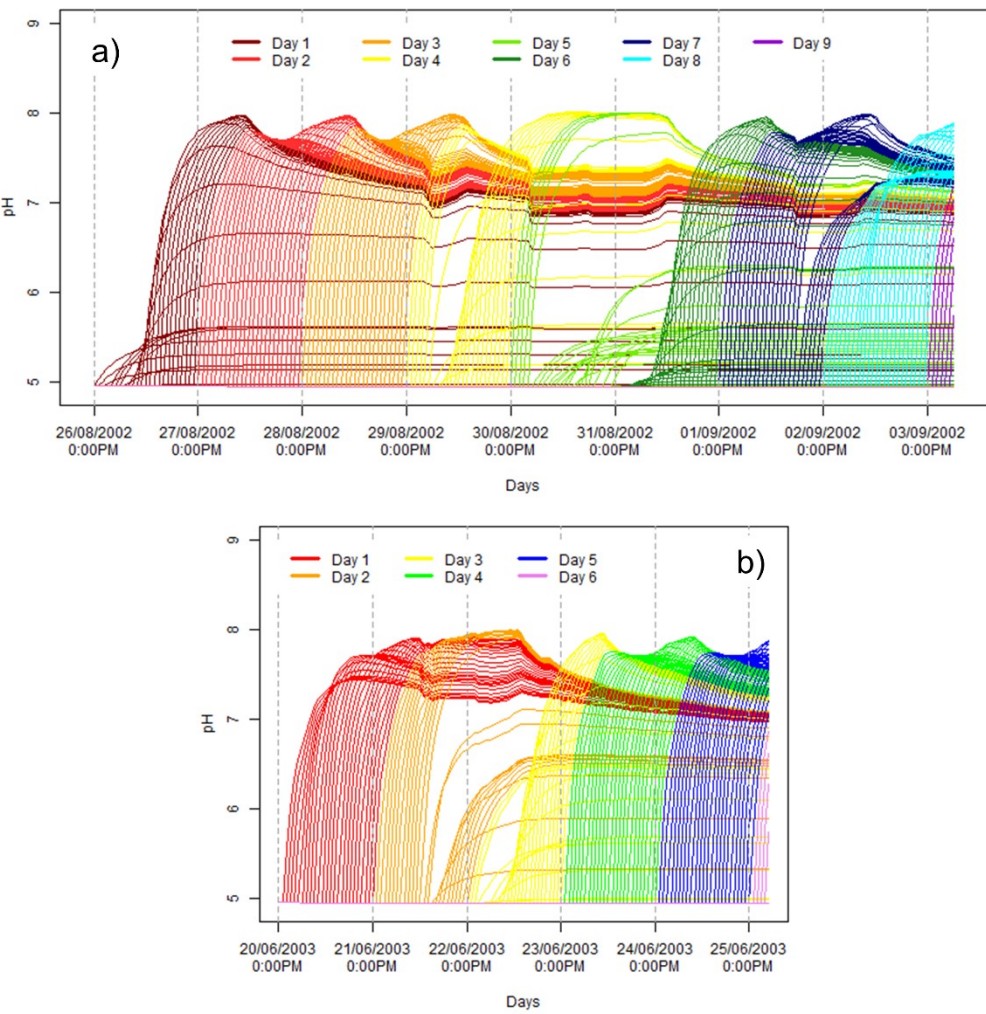

**Figure 12. Simulated soil pH in the NH₃ source layer under urine patches deposited in the same time step in the modelling periods, P2002 (a) and P2003 (b) in the baseline experiments with GAG_field. The different colours indicate the days of the urination events. Each line indicates soil pH under urine patches deposited in a given time step, while the different colours indicate the days of the urination events.**

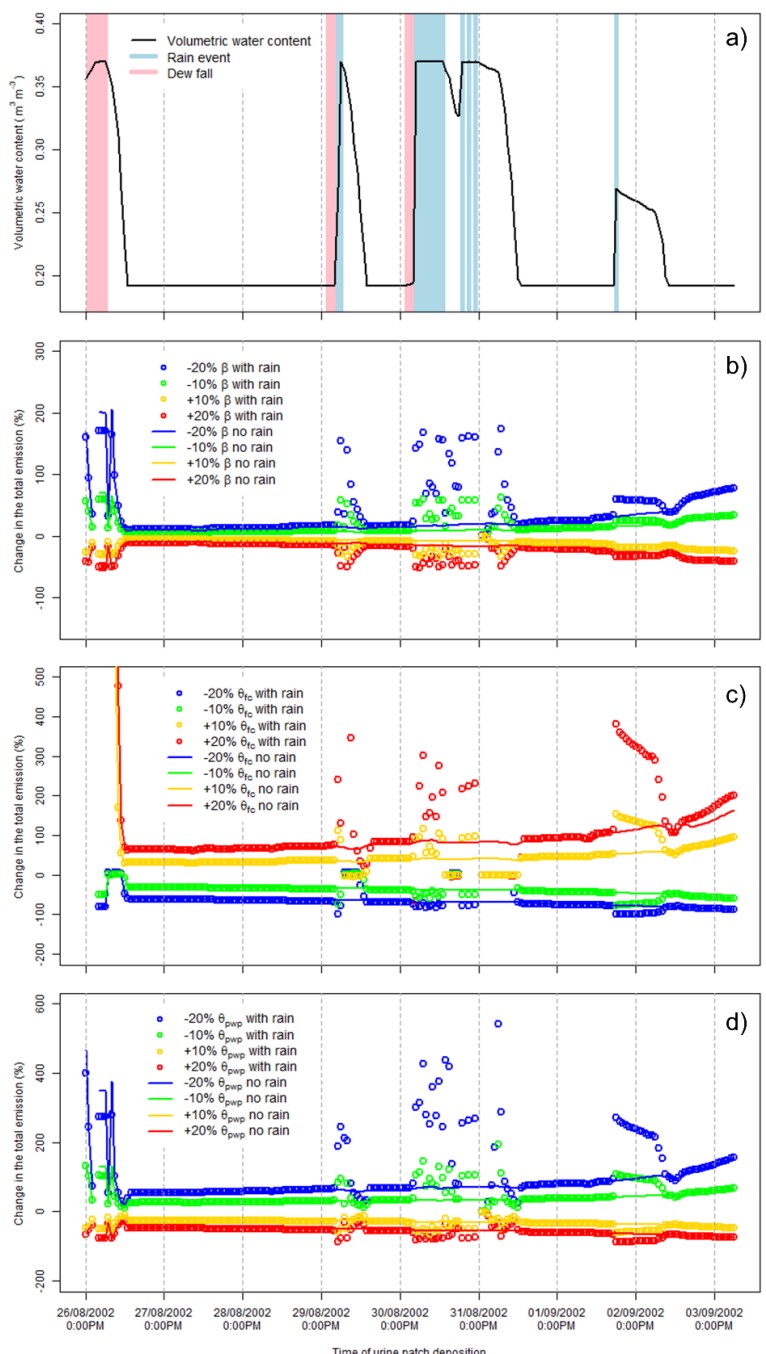

**Figure 13. Results of the perturbation experiments for every single urine patch deposited over P2002. The results are shown in comparison with the volumetric water content of the soil at the time of urine patch deposition, changing in response to the events of precipitation and dewfall (a). The investigated parameters were: the buffering capacity (β, b), the field capacity ($\theta_{fc}$, c) and the permanent wilting point ($\theta_{pwp}$, d). On figures b)-d), a point represents the percentage difference in the total NH₃ emission from the urine patch deposited in the given time step, and lines denotes the same, assuming zero precipitation over the modelling periods.**

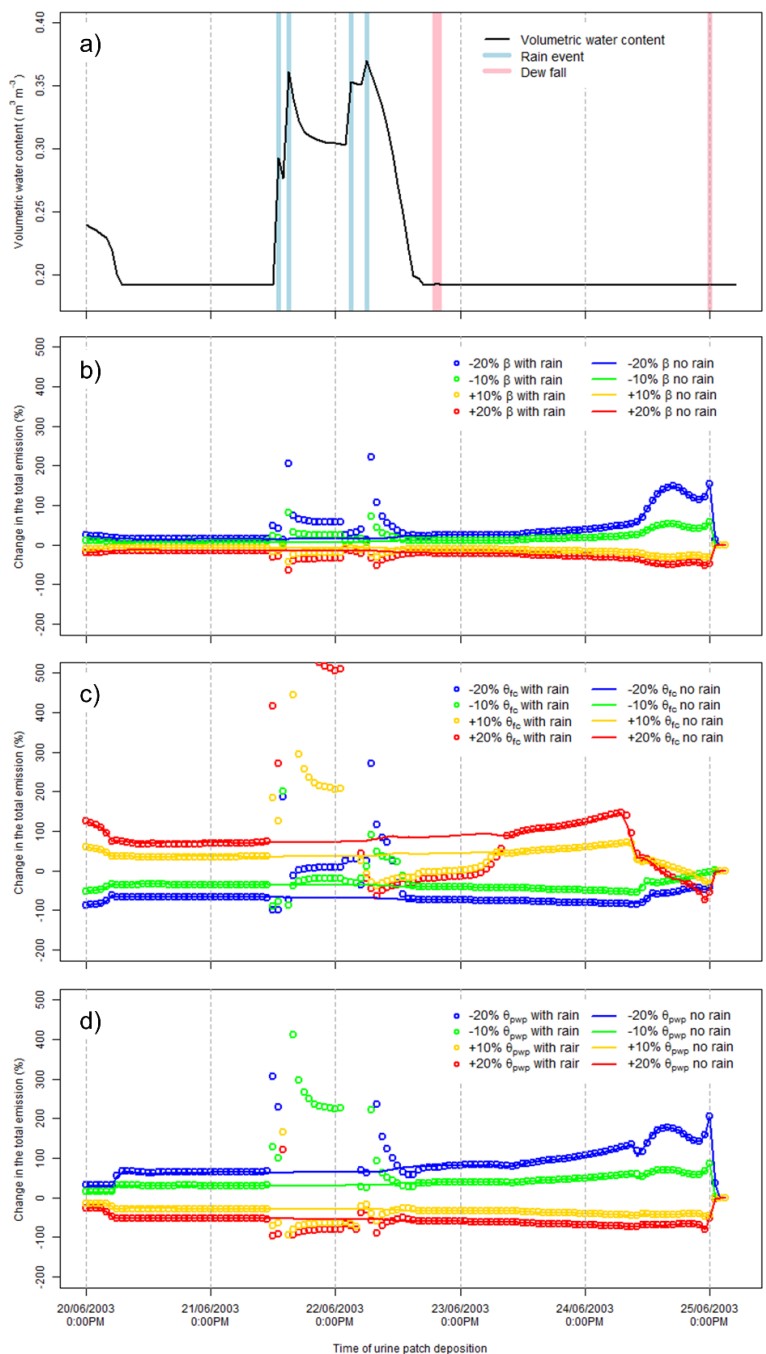

**Figure 14. Results from the same experiments illustrated in Fig. 13, for P2003.**

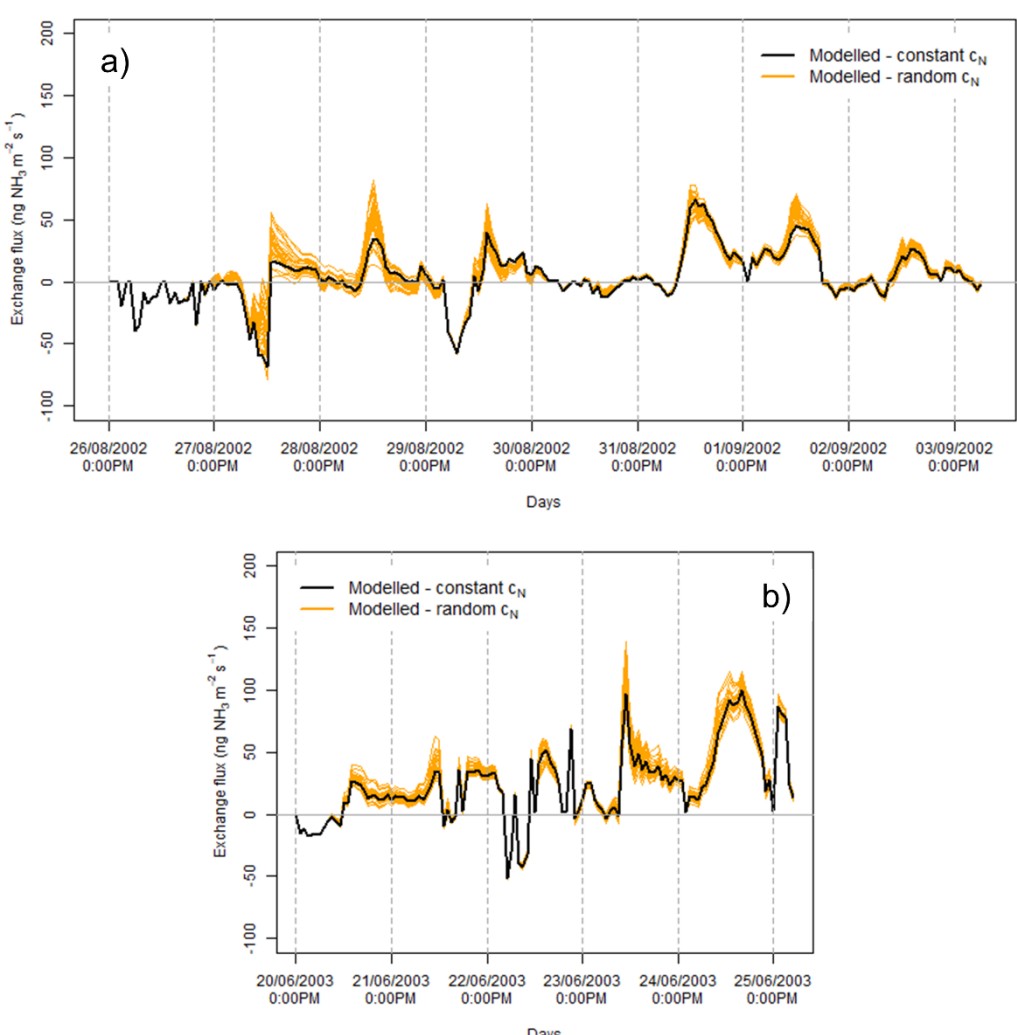

**Figure 15. Simulated NH₃ exchange fluxes from the baseline simulation with GAG_field with a constant c$_N$ (black line), and 30 model experiments in which c$_N$ was randomized for every time step (orange lines) for the modelling periods P2002 (a) and P2003 (b).**

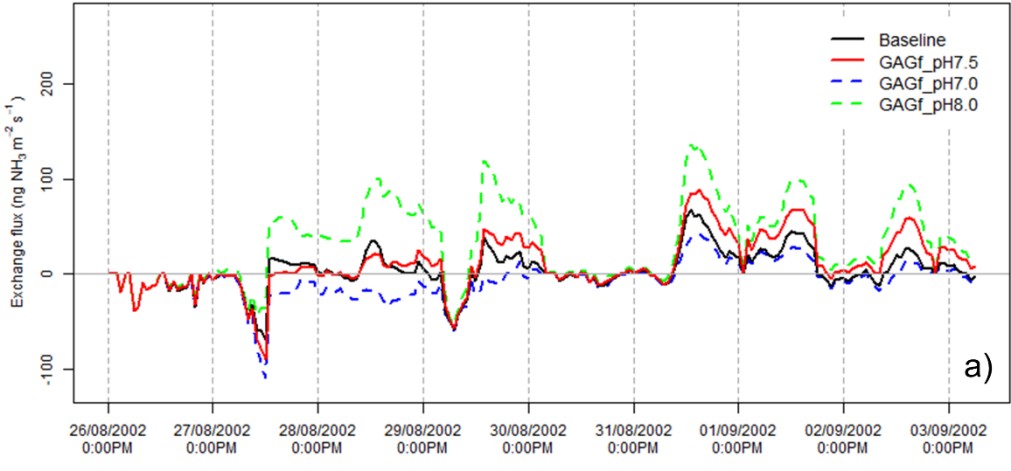

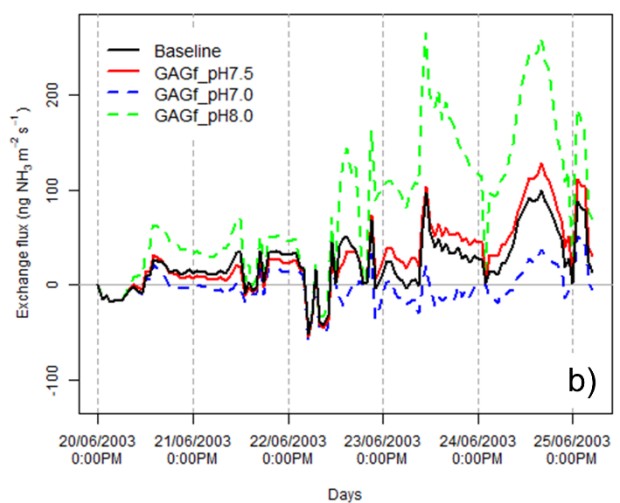

**Figure 16. NH₃ exchange fluxes simulated by GAG_field with the original dynamic approach for soil pH (Baseline), and when constant values of soil pH were assumed: pH 7.5 (GAGf_pH7.5), pH 7.0 (GAGf_pH7.0) and pH 8.0 (GAGf_pH8.0). Simulations were carried out for both modelling periods, P2002 (a) and P2003 (b).**

