# Peer review of "Process-based modelling of NH3 exchange with grazed grasslands"

_Biogeosciences, 2016_

## Referee Comment (RC1) · C.R. Flechard (Referee) · 9 Feb 2017

Reviewer's comments on Biogeosciences manuscript bg-2016-555 "Process-based modelling of NH3 exchange with grazed grasslands" by A. Moring et al.

General comments

The paper describes the development, testing, evaluation and sensitivity analysis of a process-based model of NH3 exchange over grazed grassland, applied at the field scale. The 1-D model (GAG_field) is the extension of the recently-developed model of NH3 exchange (GAG_patch, see BG 13, 1837-1861, 2016) for a single urine patch deposited to soil by a grazing animal; NH3 emissions by multiple urine patches are dynamically simulated at the field scale, and their interactions with the surrounding "clean" areas (unaffected by urine) are accounted for using a bi-directional exchange

scheme to simulate NH3 recapture by the grassland ecosystem in the near-field.

The paper is generally well written, though with some confusing symbols and turns of phrase in model description, and the model is consistently documented, with adequate referencing to the GAG_patch paper and to the wider literature. The model is evaluated versus field data (gradient-flux measurements from 2 short-term campaigns in Scotland, UK), showing broadly consistent features (emission peaks of similar magnitudes, comparable diurnal fluctuations), although considerable discrepancies remain, which may or may not be explained by large uncertainties in the flux measurements. Notwistanding considerable simplifications in model structure and large uncertainty in model parameters, the model is reasonably successful at simulating net NH3 emissions from grazed grassland, at least for the Scottish dataset tested; it remains to be seen how the model would fare in a different environment (different soil, climate, grazing density, etc).

The model is a welcome development and shows potential for deployment in regional-scale chemical transport models, and thus the paper fits the scope - and should be of interest to readers of -Biogeosciences. However, some key issues and minor flaws need to be addressed first, which may well require the authors to recalculate all tables and figures and provide new, additional model runs, without requiring any major changes in structure, contents or conclusions.

This is especially the case for sections 4.2.1-4.2.3 that deal with model sensitivity to soil physical and chemical parameters and particularly the comparison of sensitivity at field scale compared with patch scale. The main flaw here is that the sensitivity analysis for the patch scale refers to Moring et al. (2016), which used input soil, climate and grazing data from New Zealand, while the field-scale sensitivity uses very different Scottish inputs data (different pH, soil texture, etc). Because sensitivity to a model parameter is (as demonstrated by the authors) hugely dependent on initial/boundary conditions, the very different site characteristics between NZ and UK lead to very different sensitivities, regardless of scale (patch or field). The authors thus have to struggle to adjust

input/parameter data in the GAG_patch sensitivity analysis (eg FC, PWP) to 'harmonize' the two datasets, but never actually do so completely because the GAG_patch sensitivity analysis remains a 'hybrid' of NZ and UK data. I recommend a full, new analysis of GAG_patch sensitivity using exclusively Scottish input data, so that the issue of upscaling in sensitivity can be properly addressed.

Specific comments

p2, cN (gN dm-3): is this total N including all N-containing forms, or just urea-N content of urine?

p2, for clarity's sake, please indicate here that Ft is the total net flux over the canopy at patch scale in GAG_patch (while Fnet is the equivalent for field scale in GAG_field)

p5, l10 '...is considered as the only sink term.' Here it would be useful to mention that drainage/leaching of TAN and urea out of the source layer in the case of (heavy) rainfall filling porosity and entrainment of N into deepers soil layers are not considered, and whether, or why, it is reasonable to do so.

p6, l2 '...it would be preferable to neglect the overlap...' : it is not preferable, just easier!

p6, l25, and p7, l12: 10 LSU/ha as 'worst case scenario' is not a valid or representative value for the maximum grazing density in Europe. Intensive and rotational grazing practices can give rise to much higher animal numbers per ha, though for shorter periods of time. See example given in Bell et al., Atmos. Meas. Tech. Discuss., doi:10.5194/amt-2016-350, 2016, with grazing densities above 20-40 LSU/ha.

p7, l8-9, related to the above : 'As a consequence, the total area of the patches grows in the first eight days, then it remains constant while the animals are on the field'. This is true of extensive grazing, but in intensive management, grazing duration may be just 2-3 days.

p8, l25-29: the terminology Ft vs Fnet vs Fnon is slightly confusing, see e.g. the sentence "...Fnon was derived in the same way as the net NH3 flux (Ft)...", is it possible

to use less ambiguous symbols?

p9, l24: related to the above, '...GAG_patch calculates the patch emission (Ft(ti)...': is Ft actually the patch (gross) emission, or the total net flux including exchange with vegetaion? I believe it is the latter, so for clarity's sake please write '...GAG_patch calculates the patch net flux (Ft(ti)...' ?

p9, l1: '...over the non-urine area the dynamic simulation of soil chemistry is not needed...' : it would be needed, to better resolve background exchange fluxes (instead of default /constant Gamma_g values); it's just that we don't have adequate understanding, models and data to do it. Please rephrase.

p9, l17: add '(assuming no overlap)' after '...the area of the field that is not covered by any urine patches.'

p10, l1: 'When calculating Ft(ti) a slight modification is also required...' : a small modification compared with what? with GAG_patch?

p10, l5-6: sentence not clear: why does B=Bmax 'prevent infiltration' ? Do you mean rather that the model formulation cannot account for/simulate infiltration when the B=Bmax situation occurs?

p10, l6: "...prevents infiltration, resulting in no N input to the system and consequently no NH3 emission': surely you don't mean that B=Bmax means no NH3 emission?

p10, l9-10: '...the GAG_patch model modified for the non-urine area...': not very clear what this means?

p10, l11: why was dilution only treated in the first time step in GAG_patch ? And can you please state explicitly on l12 (just before Eq. 11) that dilution is now treated in GAG_field at all time steps and not just at the time of urination ? (if I understand correctly)

p10, l18: which 'second point' , what does this refer to?

p12, l9: the interval 03/09 13-17:00 is not shown anywhere on the figures, thus need not be mentioned here. Please delete.

p12, l11: "...values were assumed to be zero." This is not a reasonable assumption to make, as doing so will necessarily lead, in the model, to the maximum possible net emission (through the maximum possible soil-vegetation-atmsophere gradient). I believe this effect is clearly visible on Figures 8a, 9a, 12a, 13a, where in each case there is a sharp, step-wise, instantaneous increase in modelled flux from large deposition to large emission (step change > +100 ng m-2 s-1) around midday on 27/08, followed by a steep, instantaneous decline one day later around midday on 28/08. The timing of these step changes coincides exactly with the period of missing concentration data, where the authors assume Xa=0, with the strong and immediate effect of boosting net emission b. This is clearly not right. The authors should either: i) start the modelling period at 13:00 on 28/08, or ii) fill this 1-day gap in Xa by assuming Xa equals the mean background concentration in the area at this time of year ($\sim$2-3 $\mu$g m-3 according to C. Milford, PhD thesis, The University of Edinburgh, 2004). In either case, all flux figures should be redrawn, and all subsequent sensitivity analyses should be recalculated because the results of this day will affect the total.

This raises another important issue. The measured Xa values were used as inputs to drive the emission and bi-directional exchange models; however, in most cases, the concentrations were measured downwind from the S. field, since the prevailing wind was south-westerly, i.e. the measured Xa values were enhanced with respect to background through the emissions occurring on the S. field, and thus were themselves partly a result of the emission. The concentration gradient across a grazed field may be several $\mu$g/m3, as shown by Bell et al., Atmos. Meas. Tech. Discuss., doi:10.5194/amt-2016-350, 2016. There is thus clearly a problem of recursive logic in using the downwind concentration as input, in such a situation where there is a strong horizontal gradient. There is no easy way out of this issue since the model does not address advection, but at least i) the issue must ne mentioned in the text, and ii) a sensitivity analysis must be run and added to Section 4, in which the model will be run with a range of other Xa values e.g. Xa'= 0.5*Xa, 0.6*Xa, 0.7*Xa etc, or Xa - 0.1$\mu$g/m3, Xa - 0.2$\mu$g/m3, Xa - 0.3$\mu$g/m3, etc... This will likely have the effect of increasing emissions throughout (as already shown by the Xa=0 bias on 27-28/08), and may thus incidently improve the comparison to flux data late August/early September 2002.

p12, l21: related to the above comment: delete the reference to substitution by zero.

p12, l30-33: Gamma_g for the non-affected grassland is a key parameter for the NH3 recapture within the field, and the authors use a value of 3000 based on a comparison of model and measurements early June 2003. It would be useful to see these data as Supplementary Material, together with alternative runs using e.g. Gamma_g = 500, 1000, 5000.

p14, l2 & l12: why must there be a 'conversion', what does it mean to 'convert' SENSnet to SENSpatch? I don't quite see why SENSnet needs to be made 'compatible' with SENSpatch. The sensitivity of Fnet (GAG_field) to model parameters is the sensitivity of Fnet (GAG_field) to model parameters; there is no need for further transformation?

Perhaps the authors need to start this argument on p13, l31 by writing that they wish to compare the model sensitivities of "...Fpatch in the case of the multiple patches simulated within GAG_field and the single urine patch simulated by GAG_patch...", and that in order to do this a mathematical transformation is needed to extract the sensitivity of Fpatch from the overall sensitivity of Fnet. Thus the argument will become clearer.

p15, l8, presumably these scale parameters are the geometric standard deviation (sigma=0.786) and geometric mean ($\mu$=1.154)? The text should say so. Then the start of the next sentence says "The mean of cN ...", I presume but can't be certain that this signifies the arithmetic mean? Again should be clarified.

Equation 21: geometric or arithmetic mean?

p15, l15-16, the mean cN of 11 g dm-3 is the arithmetic mean, and the 'scale parameter'

of 2.089 is the geometric mean?

p15, l19: why 30 cN time series (why not 50 or 2000) ?

p16, l2: '...broad accordance with the observations.' Please provide the regression R^2, slope for model vs measurements, as well as RMSE and other such statistics classically used for model evaluation. Was there any filtering of the flux data for periods of low wind speed, strong nocturnal stability or any such quality criteria for flux-gradient measurements ? (it looks as though the flux time series is completely uninterrupted apart from aforementioned periods of AMANDA down time)

p16, l27-28: '...could explain part of the difference between the simulation and measurements on this day (Fig. 8), if the model overestimated the deposition component of the net flux.' The difference could just as well be due to an underestimation of the gross emission from urine patches, there is no telling which; possibly a combination of both.

p16, l29-32 and also section 4.2.4, on the diurnal variations of net emission: is it possible/likely that there is a diurnal variation in urination frequency, with animals being e.g. more actively grazing during day than night, or other temporal urination patterns ? Could this be tested by using e.g. UF(day) = 2*UF(night)? The impact of higher urination frequency during daytime would be compounded by the effect of higher temperatures.

p18, l2: change to "...are between 1-2 orders of magnitude larger than..."

Sections 4.2.1, 4.2.2, 4.2.3: these sections describe the sensitivity of GAG_field to soil physical and chemical parameters, which is very well and fine. However the authors also try to compare this sensitivity to the results obtained for GAG_patch in Moring et al (2016), and claim that the observed differences in sensitivity can be assigned to upscaling. I have a major reservation with this approach, because GAG_patch was run on a dataset from New Zealand in Moring et al 2016. The authors are aware of

this limitation because they adjust soil parameters one by one in order to make the two datasets comparable (e.g., very different initial pH values of 6.65 and 4.95, or Theta_urine of 0.18 and 0.3, etc), but as far as I understand they did not go as far as to -re-run GAG_patch and its sensitivity analysis specifically for and using only Scottish data (not just soil inputs, but also weather, stocking density, etc). The comparison of sensitivity for the different scales (patch and field) can only make sense (in terms of the impact of upscaling) if data from the same site are used.

My recommendation therefore is

- either focus on the sensitivity of GAG_field, and leave aside the comparison with GAG_patch results from Moring et al, or at least make it clear that differences cannot be assigned to upscaling

-or re-run the GAG_patch sensitivity analysis using only input data from the Scottish site (forget about GAGpatch results from Moring et al 2016), such that upscaling can be invoked to explain differences for the same soil/weather/grazing conditions In either case the authors would have to re-think/re-draw/re-calculate Tables 3, 4, 5, 6, and rewrite sections 4.2.1 through 4.2.3.

p18, l6-7: '...the main factors that can regulate the governing role of buffering in the evolution of soil pH in the NH3 source layer ... are ... pH(ti) - pH(ti-1) ...' : this turn of phrase is strange, because it is buffering that controls/reglates/modulates the change in pH over a time interval, not the other way around, semantically.

p18, l16, similarly to the above comment, '...These larger changes in soil pH generate a larger buffering effect...' sounds strange; it is the extent of buffering that controls pH change

p20, l20: Fig. 12 does not show a comparison of GAG_field vs measurements

p24, l2-3: 'Over the field scale the response of the NH3 fluxes was extremely strong to the perturbation of these parameters'. This is true, but as pointed out above, it is

not adequately demonstrated that this response is stronger at field than at patch scale, because the NZ and UK sites are different.

p24, l27-28: 'The observed sensitivities [of GAG_field] turned out to be much higher than was found in the case of GAG_patch': again, this is misleading because it gives the impression that the only reason for the difference is scale (patch vs field), which is not the case.

Same paragraph: 'The different sensitivities over the two scales can be explained by the different initial soil pH and the different soil physical characteristics': ergo, the difference has nothing to do with scale, but with soil characteristics.

Tables and figures:

Table 2: it seems the model used constant canopy height and LAI over the whole modelling period, this is surprising since cattle will consume grass, so the values should decrease from start to end, which would impact model results. Also, a leaf area index of 1m2 m-2 is very small, there would be hardly anything to eat for 50 cows for a week! I would venture that these values were measured at the end of the grazing period? It might be reasonable to re-run the model with starting LAI and canopy height values of 3 m2 m-2 and 0.2m, respectively, and assume a linear decrease until the end of the period ?

Tables 3,4,5 to be recalculated to show GAG_patch results using fully Scottish input data (soil parameters + weather data + grazing/field data + NH3 concentration data, etc), instead of using GAG_patch sensitivity values from NZ site of Moring et al 2016

Figure 5, bottom line, second cell from right: presumably this is n(tj=n) ?

Figure 6, add scale

Figure 7, the geometric mean value (mu) of 2.089 seems to be abnormally small for this distribution, I would expect the geomean nearer 5-6, close to the median?

Figures 8 through 13 to be redrawn using a non-zero concentration for the missing Xa data on 27-28/08/2002

Figure 8: 'Where the error bars are missing one of the three NH3 concentration denuders were malfunctioning or not registering data at all.' This is slightly misleading, visually, because it is at times when fluxes are most uncertain (calculated from only 2 concentration heights) that there is no indication of uncertainty on the figure... I would suggest to calculate the mean uncertainty from all fluxes from 3-point gradients (mean of red error bars already present on figure), multiply this value by e.g. a factor of 2, and apply to the rest of the points (in a different color) ?

Technical corrections

p4, l9 change 'atmospheric NH3 concentration right above the surface' to 'atmospheric NH3 concentration at thermodynamic equilibrium with the surface'

p9, l16: '... it was treated as a constant...'

p10, l4: change to '...will dilute the incoming urine...'

p10, l28: '...at height z above ground...'

p10, l29: '... the von Karman constant...'

p11, l26: change "as well as" to "and"

p11, l29: '... previous time steps.'

p11, l31: 'hereby"

p12, l3: '...averaged to an hourly...'

p12, l5: '... in the resulting averaged time series...'

p12, l17-18: suggest change to '...the wind direction was from the N. field for 7% and 15% of the time, respectively.'

p13, l12: suggest change 'kept the same' to 'unchanged'

p13, l16 : '...results... ARE compared...'

p13, l19: '...the sensitivity... differS in the case...'

p13, l31: '...and pH(t0) ARE perturbed...'

p14, l3: delete 'to' after 'equals', or write 'is equal to'

p15, l25: Fig.8, not 4 ?

p16, l18: it must be '... not operating until 24/06 13:00...' (not 23/06). 24/06 early afternoon is when the error bars re-appear in Fig.8b, ie back to 3-point vertical gradient?

p16, l19: change to "...suggesting larger uncertainty in the measured dataset."

p16, l22: "...temporal variationS of the NH3 fluxes..."

p22, l16: 'basis' instead of 'base'

p23, l28: '...can be explained by the fact that...'

p24, l9: do you mean rather a low-resolution grid to match the low resolution of the CTM (of the order of a few kmˆ2) ?

p24, l22: change to '...substantially decreased by the simultaneous NH3 deposition to the non-urine area within the field.'

p45, l4, 'The numberS above the plots...'

---

## Referee Comment (RC2) · Anonymous Referee #2 · 27 Mar 2017

**General comments**

This is a very comprehensive study presenting the development and evaluation of a model for ammonia emissions from urine patches in grazed grasslands. This work is an extension to the field scale of a model of urine patches emissions at the patch scale from Moring et al. (2016).

The model description is extensive and mainly focuses on the method used to represent and parameterise the time and space distribution of urine patches depending on the animal presence.

The model makes a reasonable job at simulating the $NH_3$ emissions measured during two cattle grazing periods of a few days in 2002 and 2003. The sensitivity analysis shows a great sensitivity to soil water field capacity and permanent wilting point which determines the maximum quantity of urine that can be hold.

The paper is well structured, well written and the figures and tables are clear and understandable. The conclusions are well drawn. The manuscript is sometimes hard to read and may be shortened; especially the description and results on the sensitivity analysis may be more synthetic.

My main concern regarding this manuscript is that I failed to understand how the GAG model was integrated in GAG field. From what I understand, the main idea which is clear from Figure 4 and eq. (5) is that in GAG field, fluxes from the field are the sum of the emissions from the urine patches and the exchange between the other surfaces and the atmosphere. However, ammonia fluxes from these two surfaces are both dependent on the ammonia concentration in the atmosphere, as it is well described in the GAG patch model described by Moring et al. (2016). More precisely, in the GAG patch model the concentration at the reference height is necessary, while in the present manuscript the concentration at $z_0$ also called here "canopy compensation point" is used to drive the exchange between the area without urine patch and the atmosphere. What is unclear is how this canopy compensation point is calculated in the field situation. If calculated from the GAG patch model the assumption is that the distance from one patch to another is small enough so that the concentration at $z_0$ can be assumed to remain constant. However, one could argue that this concentration at $z_0$ is an equilibrium point resulting from a given flux and set of resistances above and below that level, in which case it would be fair to consider the total flux Fpatch +Fnon_patch as the flux which drives the canopy compensation point. In reality, though the process is much more complex and involves horizontal advection.

I would therefore suggest the authors to better explain the underlying assumptions made on the driving concentration in the GAG_field and to discuss the potential drawbacks. I would also suggest evaluating the difference when considering Fpatch and Ftotal for driving the z0 concentration in the model.

I also suggest to show the diagram of the resistance model which is assumed in the manuscript. From my understanding, the resistance model would be as shown in Figure 1: an additional "leg" with a resistance $R_{ac} + R_{bg}$ and a potential $\chi_g$. It would also be good to explain the underlying hypotheses.

[Figure]

**Figure 1. Resistance scheme of the Gag-field model as I understood $c_{z0}$ was used.**

**Detailed comments**

- P5L9: I would suggest telling in a few words what limitations may imply the fact that no water infiltration is taken into account.

- P6L5-L6. I suggest writing which parameter is modelled with a negative binomial (area covered by patch?)

- P8 EQ5: From the equation I understand that n (over the sum symbol) and n(tj) are not the same. Please clarify.

- P9 EQ6 and L6-8: Since $\chi_{z0}$ is an equilibrium point between the ground and the atmosphere, I do not understand how it could be parameterised. To me it should depend on the flux and the concentration above. Please clarify and explain clearly the assumptions behind the calculation of the fluxes from non-urine patches area and how these are linked to the urine patches area. May be a resistance scheme in a supplementary material would help the understanding: from what I can understand from the current manuscript, the resistance scheme would be as in the GAG patch model of Moring et al. (2016) with an additional "leg" with a resistance Rac + Rbg and a potential $\chi_g$, starting from $\chi_{z0}$. Is that correct? This would imply in particular that the horizontal distance between urine patches and non-urine patches is supposed null. Once the hypotheses clearly explicated I would also suggest discussing in the discussion section what implication this would have.

- P10L1-20: The second point "ii)" is unclear. Does that mean that the total amount of liquid will be larger than the soil capacity and since no runaway and infiltration is considered this water will "disappear". Could you rephrase in a clearer way?

- P10L18-20. This sentence is unclear. Please rephrase. In particular I do not understand what "the minimum amount of urine that is always allowed to penetrate" is, and how it is linked

with the water budget. I would also suggest justifying why the minimum amount is chosen as 5% and what implication this has.

- P12L29-30: I would have thought that the "unfertilised grassland class" of Massad et al. (2010) would not be adapted here as this grassland does receive nitrogen. Please justify and also discuss the possible implications of choosing a "managed grassland class" in the discussion section.
- P16L2: "of the modelled and measured" : I suggest adding 'NH3 exchange' here.
- P18L10: I suggest changing lower and higher to low and high.
- P18L16-20 and L21-25: I found these two paragraphs unclear. Could you clarify?

**Tables and Figures**

- Table 5: Explain what is $\beta$ in the table legend.
- Figure 4: I would suggest adding a resistance scheme to better explain the model.
- Figure 8:I suggest adding the input variables of the model here or in a supplementary material ($u_*$, $T$a, RH, rain, …) as well as the potentials $\chi(z)$, $\chi_{z0}$, $\chi_g$, $\chi_p$. This will ease the understanding of the flux dynamics.

**References**

- Móring, A. *et al.* A process-based model for ammonia emission from urine patches, GAG (Generation of Ammonia from Grazing): description and sensitivity analysis. *Biogeosciences* **13**, 1837-1861, doi:10.5194/bg-13-1837-2016 (2016).
- Massad, R. S., Nemitz, E. & Sutton, M. A. Review and parameterisation of bi-directional ammonia exchange between vegetation and the atmosphere. *Atmospheric Chemistry and Physics* **10**, 10359-10386, doi:10.5194/acp-10-10359-2010 (2010).

---

## Author Comment (AC1) · 5 Jun 2017

**Authors' response to the review of Referee 1, Christophe Flechard on "Process-based modelling of NH₃ exchange with grazed grasslands"**

We thank the referee for the thorough and insightful comments. We believe that following his suggestions our manuscript will be significantly improved. Our responses and the changes we make to address the referee's comments are provided below point-by-point. The cited literature, as well as the modified and the newly created figures and tables are listed at the end of this document.

**Comment 1**: *p2, cN (gN dm-3): is this total N including all N-containing forms, or just urea-N content of urine?*

> **Our answer:** Since in the model we assume that urine consists of urea and water at the moment of the deposition of the urine patch, this refers to the urea-N content of urine. The following changes clarify that this is the only form of urea N considered in the model.

> **Change to the manuscript:**

> On P2, we modify the descriptions of the symbols as follows (text inserted in bold):

| | |
|---|---|
| $c_N$ (g N dm$^{-3}$) | N content of the urine **(assumed to be in the form of urea)** |
| $c_N^{Ave}$ | Average urinary N concentration **(assumed to be in the form of urea)** in urine patches deposited in the same time step |
| $c_N^{Dil}$ (g N dm$^{-3}$) | Urine N content **(assumed to be in the form of urea)** after dilution in the soil |
| $c_N^{k}$ (g N dm$^{-3}$) | Urinary N concentration **(assumed to be in the form of urea)** in the $k^{th}$ urine patch |

For further clarification, on P5 in L7, after the end of the sentence we add the following sentence:

"Following the considerations of Móring et al. (2016), the model handles urine as a water solution of urea, i.e. the urinary N content is assumed to be in the form of urea."

**Comment 2**: *p2, for clarity's sake, please indicate here that Ft is the total net flux over the canopy at patch scale in GAG_patch (while Fnet is the equivalent for field scale in GAG_field)*

> **Our answer:** Agreed. Please see our modification below.

> **Change to the manuscript:**

> On P2, we modify the description of $F_t$ as follows (text inserted in bold):

| | |
|---|---|
| $F_t$ (µg N m$^{-2}$ s$^{-1}$) | Total NH₃ exchange flux over the canopy **above a single urine patch** |

**Comment 3:** *p5, l10 '...is considered as the only sink term.' Here it would be useful to mention that drainage/leaching of TAN and urea out of the source layer in the case of (heavy) rainfall filling porosity and entrainment of N into deepers soil layers are not considered, and whether, or why, it is reasonable to do so.*

**Our answer:** We agree to add information on the assumption for the GAG model for a single urine patch. However, this part of the manuscript is an overview of the model, which summarizes the approach of Móring et al. (2016). Therefore, we add here a short comment which refers to further details in that paper.

**Change to the manuscript:**

On P5 from L5 we change:

„The GAG model (Móring et al., 2016) is a process-based $NH_3$ emission model for a single urine patch that is capable of…"

as follows:

"The GAG model, applied and extended to the field scale in this study, is a process-based $NH_3$ emission model for a single urine patch. An in-depth description of the model, together with a comprehensive sensitivity analysis can be found in Móring et. al (2016) and Móring (2016). The GAG model is capable of…"

On P5 in L12, after „can hold" we add:

„Since during the development of the GAG model simplicity was a key aspect, the effect of the vertical movement of the liquid within the soil (leaching and capillary rise) as well as the mixing of urea and the products of its hydrolysis within the solution was neglected."

**Comment 4:** *p6, l2 '...it would be preferable to neglect the overlap...' : it is not preferable, just easier!*

**Our answer:** To solve a problem, we consider that an easier way is preferable if it gives a reasonable approach for the solution. We believe that our corrected approach, as described in our answers to Comment 5 and 6, is a reasonable solution for the handling of the issue of the overlap of the urine patches.

**Change to the manuscript:**

We add the following amendment to On P6 in L3 (inserted text in bold):

"Thus, it would be preferable to neglect the overlap of the patches **if the error from this simplification can be shown to be small**".

**Comment 5:** *p6, l25, and p7, l12: 10 LSU/ha as 'worst case scenario' is not a valid or representative value for the maximum grazing density in Europe. Intensive and rotational grazing practices can give rise to much higher animal numbers per ha, though for shorter periods of time. See example given in Bell et al., Atmos. Meas. Tech. Discuss., doi:10.5194/amt-2016-350, 2016, with grazing densities above 20-40 LSU/ha.*

**Our answer:** If we assume that an intensive grazing period lasts for 3 days (as mentioned by the reviewer in Comment 6), it is possible to apply Eq. 1 and 3 with the maximum $A_{patch}$ values from Table 1 to estimate the number of LSU/ha that is associated with the same level of error due to patch overlap. Neglecting patch overlap for such a period would imply that 57 and 113 LSU/ha can be on the field to keep the error under 5% and 10%, respectively. In case of sheep, the same numbers will be 26.1 and 51.7 LSU/ha, respectively. For cows the resulting grazing densities are above the values mentioned by the reviewer, therefore, even in the worst case for short periods of rotational grazing, the error will be under 5%. For sheep, calculating with 44 LSU/ha, the highest grazing density in Bell et al., the error will be 8%, which is still reasonable for the worst case scenario.

While patch overlap can therefore be generally neglected for continuous grazing and short periods of rotational grazing, we acknowledge that there may be extreme cases of intense extended grazing where patch overlap could become relevant.

**Change to the manuscript:**

On P7 in L14, after "no over-lap case", before the last sentence of the paragraph we add:

"In addition, it should be stressed that in the above calculation the case of rotational or intensive grazing was not taken into account when the grazing density can be above 20-40 LSU/ha (e.g. Bell et al., 2016), whilst the animals are typically on the field only for only a few days. If it is assumed that an intensive grazing period typically lasts for a maximum of 3 days, using Eq. 1 and 3, with the maximum $A_{patch}$ values from Table 1, in case of cows, 57 and 113 LSU/ha can be on the field to keep the error – originating from the neglect of the overlap between the urine patches - under 5% and 10%, respectively. In case of sheep the same numbers will be 26.1 and 51.7 LSU/ha, respectively. For cows, the resulting grazing densities are above the 40 LSU/ha, therefore, even in the worst case, the error will be under 5%. For sheep, calculating with 44 LSU/ha, the highest grazing density in Bell et al. (2016), the error will be 8%, which can be still considered reasonable for the worst case scenario. While patch overlap can therefore be generally neglected for continuous grazing and short periods of rotational grazing, we acknowledge that there may be some extreme cases of intense extended grazing where patch overlap could become relevant."

**Comment 6:** *p7, l8-9, related to the above : 'As a consequence, the total area of the patches grows in the first eight days, then it remains constant while the animals are on the field'. This is true of extensive grazing, but in intensive management, grazing duration may be just 2-3 days.*

**Our answer:** Please see our answer and the suggested modifications for Comment 5.

**Comment 7**: *p8, l25-29: the terminology Ft vs Fnet vs Fnon is slightly confusing, see e.g. the sentence "...Fnon was derived in the same way as the net NH3 flux (Ft)...", is it possible to use less ambiguous symbols?*

**Our answer:** We think that the symbols used for the description of GAG_field, $F_{net}$, $F_{non}$ and $F_{patch}$ are clearly distinguishable from each other. However, we agree with the reviewer that referring to the symbol $F_t$ used by Móring et al. (2016) in the description of GAG_patch might cause confusion. To avoid this, we modify the text as showed below.

**Change to the manuscript:**

On P8, from L27 we change the last sentence of the paragraph to:

"Based on this, $F_{non}$ was derived in the same way as $F_t$, the net $NH_3$ flux over a urine patch in GAG_patch, described by Eq. (1)-(7) in Móring et al. (2016), together with the following simplifications:"

**Comment 8:** *p9, l24: related to the above, '...GAG_patch calculates the patch emission (Ft(ti)...': is Ft actually the patch (gross) emission, or the total net flux including exchange with vegetaion? I believe it is the latter, so for clarity's sake please write '...GAG_patch calculates the patch net flux (Ft(ti)...' ?*

**Our answer:** As it was indicated earlier in the text of the manuscript (P8 L28) it is the net $NH_3$ flux over the urine patch, which yes, includes the exchange with the vegetation. We clarify this in the text.

**Change to the manuscript:**

On P9, in L23-24, we change the sentence to:

"Finally, $F_{patch}^{j}(t_i)$ was determined by Eq. (10), which expresses that before the deposition of the urine patch, the area is handled as non-urine area (first condition), and afterwards GAG_patch calculates the net $NH_3$ flux over the urine patch ($F_t(t_i)$, second condition)."

**Comment 9:** *p9, l1: '...over the non-urine area the dynamic simulation of soil chemistry is not needed...' : it would be needed, to better resolve background exchange fluxes (instead of default /constant Gamma_g values); it's just that we don't have adequate understanding, models and data to do it. Please rephrase.*

**Our answer:** We agree with the reviewer and clarify our meaning accordingly. We had meant that this is not required according to our model structure. We had not meant to comment on whether dynamic modelling of background soil chemistry would be useful. Please see the rephrased sentence below.

**Change to the manuscript:**

On P9 in L1-2, we change the sentence as follows:

"Since over the non-urine area undisturbed soil chemistry is assumed, the dynamic simulation of soil chemistry in GAG_field is not needed. Therefore, the original version of the two-layer canopy compensation point model by Nemitz et al. (2001) is used. While dynamic simulation of undisturbed soil chemistry would be a useful avenue for further research, it is not addressed in the present study."

**Comment 10:** *p9, l17: add '(assuming no overlap)' after '...the area of the field that is not covered by any urine patches.'*

**Our answer:** Please our modification below.

**Change to the manuscript:**

On P9 in L17, we change the sentence as follows (inserted text with bold):

"The size of $A_{non}$ in the given $t_i$ time step is the area of the field that is not covered by any urine patches **(assuming no overlap)**:"

**Comment 11:** *p10, l1: 'When calculating Ft(ti) a slight modification is also required...' : a small modification compared with what? with GAG_patch?*

**Our answer:** Yes, with the GAG_patch model. Please see our modification below.

**Change to the manuscript:**

On P10 in L1 we change the sentence as follows (inserted text with bold):

When calculating $F_t(t_i)$ a slight modification is also required **compared with the GAG_patch model**, regarding the urea added with a single urination ($U_{add}$).

**Comment 12:** *p10, l5-6: sentence not clear: why does B=Bmax 'prevent infiltration' ? Do you mean rather that the model formulation cannot account for/simulate infiltration when the B=Bmax situation occurs?*

**Our answer:** In GAG_patch the source layer cannot hold more water than $B_{H2O}(max)$ since for the incoming liquid there is no more soil pore to fill. This means that if urine deposition occurs when $B_{H2O} = B_{H2O}(max)$, there is no infiltration, resulting in no N input to the system. We clarify this in the text.

**Change to the manuscript:**

On P10 from L5, we change ii) as follows:

"may lead to the maximal water content ($B_{H2O}(max)$) in the $NH_3$ source layer. In the formulation of GAG_patch this means that for the incoming liquid there is no more soil pore to fill, i.e. there is no infiltration. Therefore, when a urine patch is deposited while the water content is at $B_{H2O}(max)$, will result in no N input to the system and consequently, no $NH_3$ emission from the soil."

**Comment 13:** *p10, l6: "...prevents infiltration, resulting in no N input to the system and consequently no NH3 emission': surely you don't mean that B=Bmax means no NH3 emission?*

**Our answer:** We meant $NH_3$ emission from the soil. In GAG_patch the $NH_3$ emission from the soil is clearly driven by the breakdown of urea and the subsequent $NH_3$ emission. Therefore, in the GAG_patch model if there is no urea-N input to the source layer, then there is no soil emission.

To distinguish the net $NH_3$ emission and soil $NH_3$ emission in GAG_patch, we modify the text.

**Change to the manuscript:**

On P10 from L5, to the end of ii) we add "from the soil" as shown in our response to Comment 12.

**Comment 14:** *p10, l9-10: '...the GAG_patch model modified for the non-urine area...': not very clear what this means?*

**Our answer:** To simulate the $NH_3$ exchange with the non-urine area, we modified the GAG_patch model, as we indicate on P8 in L26-27. To clarify this, we modify the text.

**Change to the manuscript:**

On P10 in L9-10 we change the following piece of text:

"Therefore, the water budget calculated by the GAG_patch model modified for the non-urine area right before the $j^{th}$ patch deposition"

as follows:

"Therefore, the water budget for the non-urine area (simulated by the modified version of the GAG_patch as described above), right before the $j^{th}$ patch deposition"

**Comment 15**: *p10, l11: why was dilution only treated in the first time step in GAG_patch ? And can you please state explicitly on l12 (just before Eq. 11) that dilution is now treated in GAG_field at all time steps and not just at the time of urination ? (if I understand correctly)*

**Our answer:** In this section, we gave a detailed description of how to handle the diluting effect of rain events *when it happens in the same time with urine application* (P10, point i)). In the description of GAG_patch by Móring et al. (2016), $U_{add}$ was not defined as a function of time, since it was defined only for the first time step of a single urine patch. As over the field urine patches are deposited in every time step, $U_{add}$ will be different for every urine patch depending on when the urine patch was deposited ($t_j$ as defined in the our manuscript on P8 in L18 and used in Eq. 11 for $U_{add}$). We modify the text for better clarity.

In addition to the above, we would like to clarify that after urine deposition the diluting effect is simulated by the model, accounting for the water budget and the TAN budget in every time step (see Móring et al., 2016).  In this way, dilution is treated in GAG_field in each time-step where liquid is added: either rain or urine or both.

**Change to the manuscript:**

On P10 in L12 after "applied to the surface" we modify the text as follows:

"This means that in Móring et al. (2016) $U_{add}$ was not defined as a function of time. Therefore, in the field-scale model, where urine patches are deposited in every time step, $U_{add}$ was calculated for all of urine patches deposited in every $t_j$ as:"

**Comment 16:** *p10, l18: which 'second point', what does this refer to?*

**Our answer:** to ii) on P10. No other numbered lists were used in the text previously. To make this clearer, we modify the text.

**Change to the manuscript:**

On P10 in L7, we change "to address the first point" to "to address point i)".

Similarly, on P10 in L18, we change "resulting from the second point" to "resulting from point ii)".

**Comment 17:** *p12, l9: the interval 03/09 13-17:00 is not shown anywhere on the figures, thus need not be mentioned here. Please delete.*

**Our answer:** Indeed. We delete the cited piece of text and amend the following sentence accordingly.

**Change to the manuscript:**

On P12 in L9 we delete "and over 03/09 13:00 – 17:00".

On P12 in L21-22 until "hourly time step" we change the text to:

"The individual gap was interpolated from the values from the previous and next time step, whilst over the long period of missing data in $\chi_a$ (25 consecutive hourly time steps)"

**Comment 18/1:** *p12, l11: "...values were assumed to be zero." This is not a reasonable assumption to make, as doing so will necessarily lead, in the model, to the maximum possible net emission (through the maximum possible soil-vegetation-atmsophere gradient). I believe this effect is clearly visible on Figures 8a, 9a, 12a, 13a, where in each case there is a sharp, step-wise, instantaneous increase in modelled flux from large deposition to large emission (step change > +100 ng m-2 s-1) around midday on 27/08, followed by a steep, instantaneous decline one day later around midday on 28/08. The timing of these step changes coincides exactly with the period of missing concentration data, where the authors assume Xa=0, with the strong and immediate effect of boosting net emission b. This is clearly not right. The authors should either: i) start the modelling period at 13:00 on 28/08, or ii) fill this 1-day gap in Xa by assuming Xa equals the mean background concentration in the area at this time of year (2-3 g m-3 according to C. Milford, PhD thesis, The University of Edinburgh, 2004). In either case, all flux figures should be redrawn, and all subsequent sensitivity analyses should be recalculated because the results of this day will affect the total.*

**Our answer:** Agreed. Since one of the condition for the model application is that grazing should start on the field at the beginning of the model period (P10, L10), we decided to replace the assumed zero values to the average of the existing measurements over the modelling period P2001. We believe that this assumption is more realistic than assuming the background concentration from the literature. Based on this modification, we recalculate all the model results related to P2002.

**Change to the manuscript:**

On P12 in L11, we change "the values were assumed to be zero" to "the values were replaced by the average of the measured values of $\chi_{air}$ over P2002 (1.71 µg m$^{-3}$)".

In addition, we recalculate all the model results related to P2002 as shown at the end of this document: Figures 8-13 and Table 3.

**Comment 18/2:** *This raises another important issue. The measured Xa values were used as inputs to drive the emission and bi-directional exchange models; however, in most cases, the concentrations were measured downwind from the S. field, since the prevailing wind was south-westerly, i.e. the measured Xa values were enhanced with respect to background through the emissions occurring on the S. field, and thus were themselves partly a result of the emission. The concentration gradient across a grazed field may be several g/m3, as shown by Bell et al., Atmos. Meas. Tech. Discuss., doi:10.5194/amt-2016-350, 2016. There is thus clearly a problem of recursive logic in using the downwind concentration as input, in such a situation where there is a strong horizontal gradient. There is no easy way out of this issue since the model does not address advection, but at least i) the issue must ne mentioned in the text, and ii) a sensitivity analysis must be run and added to Section 4, in which the model will be run with a range of other Xa values e.g. Xa'= 0.5\*Xa, 0.6\*Xa, 0.7\*Xa etc, or Xa - 0.1g/m3, Xa - 0.2g/m3, Xa - 0.3g/m3, etc... This will likely have the effect of increasing emissions throughout (as already shown by the Xa=0 bias on 27-28/08), and may thus incidently improve the comparison to flux data late August/early September 2002.*

**Our answer:** As we highlight in the manuscript, we agree with the reviewer concerning the error originating from the neglect of the heterogeneity of the simulated field due to the urine patches (P8 L5-7). To address the reviewer's comment, we therefore extend this part of the text to clarify the effect of this heterogeneity on the atmospheric NH$_3$ concentration above the field. In addition, we carry out a sensitivity analysis to explore the effect of $\chi_{air}$ on the NH$_3$ exchange over the field.

**Change to the manuscript:**

On P8 from L5 we change "Since a grazed field, due to the urine patches, is not a uniform source of NH$_3$, an error of the estimation of the total NH$_3$ flux can originate from the exclusion of the horizontal advection" as follows:

"One of the challenges of simulating bi-directional exchange at the field scale is that fluxes are both driven by atmospheric concentrations (especially for deposition) and affect atmospheric concentrations (especially for emission) (e.g. Loubet et al., 2009). In addition, due to the urine patches, a grazed field is not a uniform source of NH$_3$. One of the consequences is that the atmospheric concentration of NH$_3$ is not homogenous over the field (see e.g. Bell et al., 2016). Both effects result in a horizontal advection of NH$_3$, neglecting which leads to an error of the estimation of the total NH$_3$ flux. At the field scale, this affect can be explored by explicit consideration of horizontal gradients (Loubet et al., 2009) or by sensitivity analysis to the values of $\chi_{air}$. In application to regional scale models, the overall effect of bi-directional exchange can be incorporated as emission/deposition feeds back to the simulated value of $\chi_{air}$."

On the same page in L8 after the end of the sentence we add:

"To investigate of the effect of $\chi_{air}$ on the simulated $NH_3$ flux a sensitivity analysis for $\chi_{air}$ was carried out (Section 4.2.1)."

On P13 in L10, after the last sentence we add:

"Finally, a perturbation experiment was carried out for $\chi_{air}$ (Section 4.2.1)."

We added the results to Table 3 (see at the end of this document).

On P13 in L14 we change 428 g N to 127 g N. (Because of the modified $\chi_{air}$ dataset, this is how the total net $NH_3$ exchange changed over the field in P2002.)

On P13 in L15, we add after the last sentence:

"In the case of the perturbation experiments for $\chi_{air}$, $\chi_{air}$ was modified by the ±10% and ±20% of its average over both periods. These average concentrations in P2002 and P2003 were 1.73 µg $NH_3$ m$^{-3}$ and 1.51 µg $NH_3$ m$^{-3}$, respectively."

On P17 after L26 we add the following paragraph:

"As for $\chi_{air}$, in Table 3 the percentage differences for P2002 over the whole field suggest a significant effect on $\Sigma F_{net}$. However, comparing the absolute hourly change to that for P2003, it can be concluded that the absolute influence was similar in the two modelling periods. It can be also clearly seen that the absolute hourly changes over the urine patches are negligibly small in both P2002 and P2003 compared to the absolute changes observed for the whole field. This suggests that the value of $\chi_{air}$ affects $\Sigma F_{net}$ mainly through the non-urine area, rather than the urine patches."

On P23 after L10 (to the Discussion) we add the following paragraph:

"In GAG_field, the horizontal dispersion of $NH_3$ on the field was neglected, and as such, the homogeneity of $\chi_{air}$ was assumed. However, the perturbation experiments showed that $\chi_{air}$ can considerably affect the total $NH_3$ exchange over the non-urine area. This suggests that including the effect of horizontal advection to the model could possibly improve the simulation of $NH_3$ exchange over a grazed field. This effect is treated directly when such a bi-directional model as GAG_field is incorporated into a regional atmospheric chemistry transport model, through the influence of surface emission/deposition on the simulated value of near-surface $\chi_{air}$."

**Comment 19:** *p12, l21: related to the above comment: delete the reference to substitution by zero.*

**Our answer:** We change this as explained in our response to Comment 18/1.

**Change to the manuscript:** As stated above.

**Comment 20:** *p12, l30-33: Gamma_g for the non-affected grassland is a key parameter for the NH3 recapture within the field, and the authors use a value of 3000 based on a comparison of model and measurements early June 2003. It would be useful to see these data as Supplementary Material, together with alternative runs using e.g. Gamma_g = 500, 1000, 5000.*

**Our answer:** We create a plot for the calibration experiment, together with the alternative model runs requested by the reviewer, and add it to the supplementary material. Please see Fig. S1 at the end of this document.

In addition, due to the opposite wind speed (from the direction of the North Field instead of the South Field), at the end of the calibration period as mentioned in the manuscript (P12, L32) there were no measured fluxes to compare with the model results. Therefore, we shortened the period so that it ends with the last measured flux that is representative for the South field. We modify the text accordingly.

**Change to the manuscript:**

We add Fig. S1 to the supplementary material (see at the end of this document).

On P12, from L32 we change the following:

"The time period of 01/06/2003 00:00 – 09/06/2003 00:00 fulfilled this criteria. These preliminary model experiments indicated a close agreement between the measured and simulated $NH_3$ fluxes with a $\Gamma_g$ of 3000. Therefore, this value of $\Gamma_g$ was applied in the baseline simulations with GAG_field."

as follows:

"The period of 01/06/2003 00:00 – 08/06/2003 16:00 fulfilled these criteria. These preliminary model experiments indicated a reasonable agreement between the measured and simulated $NH_3$ fluxes with a $\Gamma_g$ of 3000 (see Fig. S1 in the supplementary material). Therefore, this value of $\Gamma_g$ was applied in the baseline simulations with GAG_field. To investigate the model sensitivity to this choice of $\Gamma_g$, a sensitivity analysis was carried out in Section 4.2.2."

**Comment 21:** *p14, l2 & l12: why must there be a 'conversion', what does it mean to 'convert' SENSnet to SENSpatch? I don't quite see why SENSnet needs to be made 'compatible' with SENSpatch. The sensitivity of Fnet (GAG_field) to model parameters is the sensitivity of Fnet (GAG_field) to model parameters; there is no need for further transformation? Perhaps the authors need to start this argument on p13, l31 by writing that they wish to compare the model sensitivities of "...Fpatch in the case of the multiple patches simulated within GAG_field and the single urine patch simulated by GAG_patch...", and that in order to do this a mathematical transformation is needed to extract the sensitivity of Fpatch from the overall sensitivity of Fnet. Thus the argument will become clearer.*

**Our answer:** We believe that from the comparison of the results for GAG_patch from Móring et. al (2016) and the results from our current study, valuable conclusions can be drawn for the model behaviour of GAG_field in response to the perturbation of the patch-related model parameters. In this part of the manuscript we attempted to highlight that the direct comparison of the two is not possible, since over the field the total net exchange is also affected by the $NH_3$ exchange over the non-urine area.

At this point of the manuscript our purpose was to show the relationship between $Sens_{net}$ and $Sens_{patch}$, avoiding to create an overly-complex table with data separately for the whole field and the urine patches. However, we also agree that our approach may cause

confusion for the readers. Therefore, we added the $Sens_{patch}$ values to Table 3, as shown at the end of this document. In addition, we think that it is an important conclusion, that for the patch-related parameters the ratio of $Sens_{net}$ and $Sens_{patch}$ is a close-to-constant value, therefore, we modify Section 4.2. accordingly.

**Change to the manuscript:**

We change Table 3 as showed at the end of this document.

We extend 4.2 with a subsection titled „General remarks" as shown in our answer to Comment 30.

**Comment 22:** *p15, l8, presumably these scale parameters are the geometric standard deviation (sigma=0.786) and geometric mean (=1.154)? The text should say so. Then the start of the next sentence says "The mean of cN ...", I presume but can't be certain that this signifies the arithmetic mean? Again should be clarified.*

**Our answer:** The "scale parameters" (as referred to in statistics) σ and μ are the arithmetic standard deviation and the arithmetic mean of the normal distribution of $\log(c_N)$. We think that this information might confuse the readers with a less advanced mathematical knowledge, and is not necessary to understand the purpose of this calculation. Therefore, we clarify this in the list of symbols at the beginning of the manuscript.

We agree that the definition of mean($c_N$) should be clarified in the text, which is indeed the arithmetic mean.

**Change to the manuscript:**

On P3 after the description of σ and μ we add:

"(the arithmetic standard deviation and the arithmetic mean of the normal distribution of $\log(c_N)$, respectively)"

On P15 in L8 we add "arithmetic" before "mean of $c_N$".

**Comment 23:** *Equation 21: geometric or arithmetic mean?*

**Our answer:** Please see our response to Comment 22.

**Comment 24:** *p15, l15-16, the mean cN of 11 g dm-3 is the arithmetic mean, and the 'scale parameter' of 2.089 is the geometric mean?*

**Our answer:** As noted in our response to Comment 22, $c_N = 11 \text{g dm}^{-3}$ refers to the arithmetic mean, while μ=2.089 refers to the arithmetic mean of $\log(c_N)$.

**Comment 25:** *p15, l19: why 30 cN time series (why not 50 or 2000)?*

**Our answer:** 30 is an arbitrary choice as a sample size but widely used to calculate statistics.

**Comment 26/1:** *p16, l2: '...broad accordance with the observations.' Please provide the regression R^2, slope for model vs measurements, as well as RMSE and other such statistics classically used for model evaluation.*

**Our answer:** As suggested, we calculate the model statistics and create a scatter plot to summarise the measurements against the model results. Also, we extend the text of the manuscript accordingly.

**Change to the manuscript:**

We add a scatter plot together with the model statistics. Please see Fig. N3. at the end of this document. Please see our further modifications in our response to Comment 26/2.

**Comment 26/2:** *Was there any filtering of the flux data for periods of low wind speed, strong nocturnal stability or any such quality criteria for flux-gradient measurements? (it looks as though the flux time series is completely uninterrupted apart from aforementioned periods of AMANDA down time)*

**Our answer:** No, there was not any filtering, which we implement now according the following criteria:

- according to the footprint analysis, the field contributed at least 67% to the measured flux,
- $u_* > 0.15$ m s$^{-1}$ for at least 45 minutes,
- $L^{-1} < 0.2$ m$^{-1}$, and
- $u > 1$ m s$^{-1}$.

**Change to the manuscript:**

On Fig 8. we denote the flux measurements that - based on the quality check detailed above - turned out to be robust (meeting all the above criteria) and less robust (failing one or more of the above criteria). Please see the modified version of Fig. 8 at the end of this document.

We calculate the model statistics for all of the measured data and separately for only the robust data. Please see Fig. N3 at the end of this document.

We also modify the text accordingly. On P12 after L18 we add the following paragraph:

"In addition, a quality check was carried out on the measured flux dataset, distinguishing the time periods with low wind and strong stability. A flux measurement was considered robust if it met all of the following criteria:

- according to the footprint analysis, the field contributed at least 67% to the measured flux,
- u* > 0.15 m s$^{-1}$ for at least 45 minutes,
- L$^{-1}$ < 0.2 m$^{-1}$, and
- u > 1 m s$^{-1}$.

The fluxes failing to meet one or more of the above criteria were considered as less robust. The robust and less robust data determined in this way, can be seen in Section 4.1.1 on Fig. 8."

On P16 in from L2 we change:

"In the case of P2002 (Fig. 8a) the model was in a broad accordance with the observations. It captures…"

as follows:

"In the case of P2002, although the model statistics imply a weak model performance (Fig. N3a), the visual comparison of the modelled and measured $NH_3$ exchange (Fig 8a) suggests a broad accordance between the two datasets. The model captures…"

On the same page after the last sentence of the first paragraph we add:

"When the last 6 values before this event as well as the less robust data were removed from the dataset, the calculated statistics reflected a much promising model performance."

On the same page, we change the first sentence of the second paragraph as follows:

"Similarly to P2002, the model statistics implies a relatively low model performance (Fig. N3b) for P2003 as well, however, according to Fig. 8b, the simulation generally agreed with the observations within 50 ng m$^{-2}$ s$^{-1}$. The removal of the less robust data from the dataset, resulted in improved model statistics (Fig. N3b), suggesting a better agreement between the model and the measurements."

**Comment 27:** *p16, l27-28: '…could explain part of the difference between the simulation and measurements on this day (Fig. 8), if the model overestimated the deposition component of the net flux.' The difference could just as well be due to an underestimation of the gross emission from urine patches, there is no telling which; possibly a combination of both.*

**Our answer:** Agreed.

**Change to the manuscript:** On P16, we remove L26-28.

**Comment 28:** *p16, l29-32 and also section 4.2.4, on the diurnal variations of net emission: is it possible/likely that there is a diurnal variation in urination frequency, with animals being e.g. more actively grazing during day than night, or other temporal urination patterns? Could this be tested by using e.g. UF(day) = 2\*UF(night)? The impact of higher urination frequency during daytime would be compounded by the effect of higher temperatures.*

**Our answer:** We agree, that it would be interesting to test how the modelled $NH_3$ fluxes would respond to an assumed diurnal pattern of urination frequency. However, we believe that this is out of the scope of the present study. However, in Section 4.2.4. we investigated the sensitivity of the total net $NH_3$ exchange to the changes in the applied average hourly urination frequencies.

**Change to the manuscript:**

Please see the suggested changes in our response to Comment 37.

**Comment 29:** *p18, l2: change to "…are between 1-2 orders of magnitude larger than…"*

**Our response:** We do the correction as suggested by the reviewer.

**Change to the manuscript:**

On P18 in L2 we change "significantly larger than" to "1-2 orders of magnitude larger than".

**Comment 30:** *Sections 4.2.1, 4.2.2, 4.2.3: these sections describe the sensitivity of GAG_field to soil physical and chemical parameters, which is very well and fine. However the authors also try to compare this sensitivity to the results obtained for GAG_patch in Moring et al (2016), and claim that the observed differences in sensitivity can be assigned to upscaling. I have a major reservation with this approach, because GAG_patch was run on a dataset from New Zealand in Moring et al 2016. The authors are aware of this limitation because they adjust soil parameters one by one in order to make the two datasets comparable (e.g., very different initial pH values of 6.65 and 4.95, or Theta_urine of 0.18 and 0.3, etc), but as far as I understand they did not go as far as to -re-run GAG_patch and its sensitivity analysis specifically for and using only Scottish data (not just soil inputs, but also weather, stocking density, etc). The comparison of sensitivity for the different scales (patch and field) can only make sense (in terms of the impact of upscaling) if data from the same site are used.*

*My recommendation therefore is*

*- either focus on the sensitivity of GAG_field, and leave aside the comparison with GAG_patch results from Moring et al, or at least make it clear that differences cannot be assigned to upscaling*

*-or re-run the GAG_patch sensitivity analysis using only input data from the Scottish site (forget about GAGpatch results from Moring et al 2016), such that upscaling can be invoked to explain differences for the same soil/weather/grazing conditions In either case the authors would have to re-think/re-draw/re-calculate Tables 3, 4, 5, 6, and rewrite sections 4.2.1 through 4.2.3.*

**Our answer:** We agree that we did not describe the idea well-enough behind our model comparison in the manuscript, therefore, we rewrite the Sections 4.2.1-4.2.3 and add further, explanatory figures to the manuscript. In our modifications, the logic we follow is:

- The comparison of the result from the sensitivity analysis for the patch-related parameters is important, since we believe that in this way important lessons can be learned on the behaviour of the model.

- There are three main differences in the GAG_field simulations and the GAG_patch model experiments:

    1) in GAG_field the total net $NH_3$ exchange consists of not only the total $NH_3$ emission over the urine patches, but also the total net $NH_3$ exchange over the non-urine area,

    2) in GAG_field multiple urine patches are deposited in every time step, whilst in GAG_patch a single urine patch is simulated,

3) and the two models were applied for two different sites with different circumstances: GAG_field was applied for a grazed grassland at Easter Bush, Scotland and GAG_patch was evaluated for a grassland at Lincoln, New-Zealand.

- To address these differences point-by-point:

    1) We derive the Sens$_{patch}$ values for the GAG_field simulations as explained in our response to Comment 21.

    2) For the parameters $\beta$, $\theta_{fc}$, and $\theta_{pwp}$ we calculate the corresponding sensitivities (Sens$_{patch}^{single}$). See the new figures Fig. N1 and Fig. N2 at the end of this document. Apart from showing the difference between the sensitivity of the area covered by urine patches (multiple patches) and the single urine patches, this will give an insight on how the model sensitivity responds to upscaling.

    3) The Sens$_{patch}^{single}$ values calculated in the previous point are now comparable with the Sens$_{patch}^{single}$ values for GAG_patch (as published by Móring et al., 2016), and the differences will reflect on the different circumstances at the two experimental site, Easter Bush, UK (GAG_field) and Lincoln, NZ (GAG_patch).

- To explore what could cause the differences in the *Sens$_{patch}^{single}$* for the urine patches in the GAG_field and GAG_patch simulations, the general model behaviour should be investigated. As such, it is equivalent to rerun the original GAG_patch with parameters from Easter Bush, or pick a urine patch deposited in the GAG_field simulation and rerun the patch scale model for it with parameters from Lincoln. Therefore, we keep Table 4 and 6 in the manuscript in their original form.

**Change to the manuscript:**

We rewrite Section 3.3.1 from the second paragraph as follows (the parts kept from the original manuscript is indicated with grey):

[revised manuscript text omitted]

Finally, we divide the discussion to two subsections and extend it.
After the first paragraph of Section 5.2 we add:

"**5.2.1. General conclusions**

The results of the perturbation experiments were compared with those from Móring et al. (2016) for GAG_patch. In general, it can be concluded that the differences in the sensitivity of the two models can originate from three sources: 1) the effect of the non-urine area on the total net $NH_3$ exchange over the whole field, 2) the different response in the total $NH_3$ exchange of the urine patches as a group, and as individual urine patches, and 3) the different soil characteristics at the two experimental sites, Easter Bush, UK (GAG_field) and Lincoln, NZ (GAG_patch).

For point 1) it was shown in general that if a patch-related parameter ($\Delta z$, $REW$, $\beta$, $pH(t_0)$, $\theta_{fc}$, $\theta_{pwp}$) is perturbed, even if the resulting change in the total $NH_3$ emission over the urine patches is the same, the percentage difference over the whole field will be larger if the deposition to the non-urine area is stronger. This is because a larger deposition term results in a smaller total net $NH_3$ exchange over the whole field, suggesting a proportionally larger change in the total over the whole field in response to the perturbation of the given parameter.

Regarding point 2) a 3) additional perturbation experiements were carried out for $\theta_{fc}$, $\theta_{pwp}$, and $\beta$. Overall, these suggest that the sensitivity of the total $NH_3$ exchange of an individual urine patch is similar to the sensitivity of the urine patches as a group if the investigated urine patch is deposited when the water content of the source layer is minimal ($\theta_{pwp}$). However, over a urine patch, the total $NH_3$ exchange can be extremely sensitive to the perturbations of $\theta_{fc}$, $\theta_{pwp}$, $\beta$, if it is deposited shortly after an event of rain fall (or dew fall), which increases the water content of the source layer at the time of urine deposition. Since in the baseline simulations with GAG_field the source layer was

dry most of the time (water content at $\theta_{pwp}$), the sensitivity for the group of urine patches was similar to the sensitivity of most of the individual urine patches deposited over the modelling periods.

The results also showed that difference between the sensitivities to $\theta_{fc}$, $\theta_{pwp}$, and $\beta$ over the urine patches in the GAG_field simulations and the GAG_patch simulation is associated with the different values of $\theta_{fc}$, $\theta_{pwp}$ at the two experimental sites. Furthermore, the different pH of the undisturbed soil at Lincoln and Easter Bush could lead to high differences in the resulted sensitivities to $\beta$ over the individual urine patches at the two sites.

In conclusion, two main reasons can be identified for the large differences in the observed sensitivity of the total net $NH_3$ exchange to $\theta_{fc}$, $\theta_{pwp}$, and $\beta$ between the baseline simulations with GAG_field and GAG_patch. The differences are caused by firstly, the fact that over the field scale in the net exchange the deposition to the non-urine area is also included, and secondly, the different soil characteristics at the two sites."

After this we insert the title of the second subsection:

"**5.2.2. Parameter-specific findings**"

Finally, on P24 from L2 we change the following two sentences:

"Over the field scale the response of the $NH_3$ fluxes was extremely strong to the perturbation of these parameters. This high sensitivity was attributed to the maximum amount of urine that the $NH_3$ source layer can hold, which depends on $\theta_{fc}$ and $\theta_{pwp}$, or if the soil volumetric water content is higher than $\theta_{pwp}$ before a urination event, the initial water content of the soil ($\theta(t_0)$)."

as follows:

"The results suggested that the sensitivity of the total $NH_3$ exchange over a urine patch is regulated by the maximum amount of urine that the $NH_3$ source layer can hold, which depends on $\theta_{fc}$ and $\theta_{pwp}$, or if the soil volumetric water content is higher than $\theta_{pwp}$ before a urination event, the initial water content of the soil ($\theta(t_0)$)."

**Comment 31:** *p18, l6-7: '...the main factors that can regulate the governing role of buffering in the evolution of soil pH in the NH3 source layer ... are ... pH(ti) - pH(ti-1) ...' : this turn of phrase is strange, because it is buffering that controls/reglates/modulates the change in pH over a time interval, not the other way around, semantically.*

**Our response:** Following the logic of the text on P18 from L5 this is logical: the effect of buffering in the model is described by the expression mentioned also in the manuscript (P18 L6):

$(pH(t_i)-pH(t_{i-1})) \times \beta_{patch.}$

The part $(pH(t_i) - pH(t_{i-1}))$ describes the change of pH. This expression is based on the definition of buffering capacity, which is the released/consumed $H^+$ (mol) by the buffers in the solution as a response to 1 unit change of pH in a $dm^3$ of solution, which implies

that the stronger are the $H^+$ consuming and producing terms in the system, the stronger will be the "buffering effect". This was used in Móring et al. (2016), where the "buffering effect" is considered as an additional term in the $H^+$ budget on top of the other $H^+$ consuming and producing terms, as shown in the referred study in Eq. 47:

$$B_{H^+}(t_i) = B_{H^+}(t_{i-1}) - i_C(t_i) + (-r_{R3} + r_{R2} + r_{R1}) + \beta_{patch}(pH(t_i) - pH(t_{i-1}))$$

**Comment 32:** *p18, l16, similarly to the above comment, '...These larger changes in soil pH generate a larger buffering effect...' sounds strange; it is the extent of buffering that controls pH change*

**Our answer:** As pointed in our answer to the pervious comment, the stronger are the $H^+$ consuming and producing terms in the system, the stronger will be the "buffering effect". In this case, the cited part of the manuscript describes that due to the more active urea hydrolysis in the GAG_field simulations than in the GAG_patch experiment, more $H^+$ ion is consumed in a time step, which means that the buffers in the system will produce more $H^+$, i.e. the buffering effect will be larger. Nevertheless, to avoid confusion by readers we modify the text.

**Change to the manuscript:** Please see the modifications we propose following the suggestion of Reviewer 2, in our response to Comment 11 by Reviewer 2.

**Comment 33:** *p20, l20: Fig. 12 does not show a comparison of GAG_field vs measurements*

**Our answer:** Agreed, we meant to refer to Fig. 8 here. We correct this in the text.

**Change to the manuscript:** On P20 in L20 we change "Fig 12." to "Fig 8."

**Comment 34:** *p24, l2-3: 'Over the field scale the response of the NH3 fluxes was extremely strong to the perturbation of these parameters'. This is true, but as pointed out above, it is not adequately demonstrated that this response is stronger at field than at patch scale, because the NZ and UK sites are different.*

**Our answer:** We change this sentence as also showed in our response to Comment 30.

**Change to the manuscript:** "Over the field scale the response of the $NH_3$ fluxes was extremely strong to the perturbation of these parameters. This high sensitivity was attributed to the maximum amount of urine that the $NH_3$ source layer can hold, which depends on $\theta_{fc}$ and $\theta_{pwp}$, or if the soil volumetric water content is higher than $\theta_{pwp}$ before a urination event, the initial water content of the soil ($\theta(t_0)$)."

as follows:

"The results suggested that the sensitivity of the total $NH_3$ exchange over a urine patch is regulated by the maximum amount of urine that the $NH_3$ source layer can hold, which depends on $\theta_{fc}$ and $\theta_{pwp}$, or if the soil volumetric water content is higher than $\theta_{pwp}$ before a urination event, the initial water content of the soil ($\theta(t_0)$)."

**Comment 35:** *p24, l27-28: 'The observed sensitivities [of GAG_field] turned out to be much higher than was found in the case of GAG_patch': again, this is misleading because it gives the impression that the only reason for the difference is scale (patch vs field), which is not the case.*

**Our answer:** In the cited sentence, we simply use the name of the two models, GAG_patch and GAG_field. The difference mentioned in this sentence is explained in the next sentence.

**Comment 36:** *Same paragraph: 'The different sensitivities over the two scales can be explained by the different initial soil pH and the different soil physical characteristics': ergo, the difference has nothing to do with scale, but with soil characteristics.*

**Our answer:** Please see our suggestion below.

**Change to the manuscript:**

On P24 in L28 after the sentence ending with "in the case of GAG_patch" we add:

"The reason for these different sensitivities is dual. Firstly, the difference originates from the different scales. When a model parameter, affecting the $NH_3$ emission from the urine patches is perturbed, the resulting change in the total net $NH_3$ exchange over the whole field will be larger compared to that in the total $NH_3$ emission from the urine patches. The reason for this is the negative deposition term in GAG_field over the non-urine area."

And in the same paragraph we change the sentence cited by the reviewer as follows:

"Secondly, and more importantly, the different sensitivities observed for the two models can be explained by the environmental circumstances at the two sites the model were applied for, i.e the different initial soil pH and the different soil physical characteristics at the two sites which determine the maximum volume of urine that can be stored in the $NH_3$ source layer."

**Comment 37:** *Table 2: it seems the model used constant canopy height and LAI over the whole modelling period, this is surprising since cattle will consume grass, so the values should decrease from start to end, which would impact model results. Also, a leaf area index of 1m2 m-2 is very small, there would be hardly anything to eat for 50 cows for a week! I would venture that these values were measured at the end of the grazing period? It might be reasonable to re-run the model with starting LAI and canopy height values of 3 m2 m-2 and 0.2m, respectively, and assume a linear decrease until the end of the period?*

**Our answer:** We thank the reviewer for this comment, especially because it pointed out a mistake in the Table 2, in which an earlier version of the input data was included. Indeed, we also realised that the measured leaf area index (*LAI*) and canopy height values (*h*) indicate too little grass on the field, therefore, instead of these, in our simulations in both modelling periods we used data as suggested by Massad et al. (2010) for summer grasslands (*LAI* = 3.5 m$^2$ m$^{-2}$, *h*=0.3 m, as also in the simulation with GAG_patch in Móring et al., 2016). This choice is also supported by the fact that when the model will be applied to regional scale within an atmospheric chemistry transport model, it is highly unlikely that measured *LAI* and *h* values will be available over a large region. As such

over regional scale similar constants from the literature will be used. We modify Table 2 accordingly.

We agree with the reviewer that the effect of grazing, and its effect through the decreasing canopy height and leaf area index could have a considerable effect on the $NH_3$ flux over the field. Similarly to our answer to Comment 28, it would be interesting to investigate the effects of real grazing situations (together with daily pattern in urination frequencies) but this is out of the scope of our current study. We agree though that such investigations could be good material for a further study. Nevertheless, to illustrate the general effect of $h$ and $LAI$ on the net $NH_3$ exchange over the field, we carry out additional perturbation experiments for these two parameters.

**Change to the manuscript:**

From Table 2 we remove the values for $LAI$ and $h$. (The caption says that the parameters that are not defined in the table are the same as defined for GAG_patch in Móring et al., 2016).

On P13, in L8 we extend the sentence as follows (the added text with bold):

"***LAI* (leaf area index) and *h* (canopy height)** were also examined"

We add to the end of Section 4.2.1 the following paragraph:

"The effect of LAI on $\Sigma F_{net}$ turned out to be weak, the resulting percentage differences are negligibly small compared to the extent of the perturbations applied. Similarly, a relatively weak sensitivity was found for $h$. However, in this case, it has to be noted that the resulting percentage differences are about half of the perturbations. This means that in the case of e.g., a canopy height of 5 cm, which is -83% shorter than the $h$ used in the baseline simulations, could lead to considerable changes in the $NH_3$ exchange flux, especially toward the end of the period when due to the continuous grazing the grass is shorter on the field."

In Section 5.2, we add the following paragraph before the last one:

"For the presented simulations with GAG_field a hypothetical grazing situation was assumed, in which there is no temporal variation in $UF$, $c_N$ and $A_{patch}$. However, $UF$, $c_N$ and the volume of urine deposited by an animal can have a diurnal cycle (Misselbrook et al., 2016), latter with a potential effect on $A_{patch}$ (Li et al., 2012). In addition to these parameters, $LAI$ and $h$ was handled as constant for the whole modelling period, whilst these parameters are decreasing since due to grazing, there is less and less grass on the field toward the end of the modelling period. To assess the possible influence of these assumptions on $\Sigma F_{net}$, additional sensitivity experiments were performed with GAG_field."

Then we rewrite the last paragraph as follows:

"According to the results, whilst the uncertainty originating from the choice of a constant $A_{patch}$ and $UF$ is considerable, the uncertainty coupled with the value of $c_N$ is extremely large. Nevertheless, model simulations with randomized N concentrations implied that this uncertainty might be considerably smaller in reality than it was suggested by the sensitivity analysis. For $LAI$ and $h$, it was found, that LAI has a negligible effect on $\Sigma F_{net}$,

whereas $h$ can substantially affect the $NH_3$ exchange over the field. Therefore, future work should investigate how the modelled $NH_3$ exchange responds when a real grazing situation assumed, including a diurnal cycle of $UF$, $c_N$ and $A_{patch}$ as well as temporal changes of $LAI$ and $h$."

**Comment 38:** *Tables 3,4,5 to be recalculated to show GAG_patch results using fully Scottish input data (soil parameters + weather data + grazing/field data + NH3 concentration data, etc), instead of using GAG_patch sensitivity values from NZ site of Moring et al 2016*

**Our answer:** Please see the modified version of Table 3 at the end of this document. We keep Tables 4-6 in the manuscript in their original form as explained in our answer to Comment 30.

**Change to the manuscript:** As stated above.

**Comment 39:** *Figure 5, bottom line, second cell from right: presumably this is n(tj=n) ?*

**Our answer:** Agreed. Reviewer 2 pointed out that we used $n$ for the number of the urine patches as well as for the maximum number of the time steps within the modelling period. Therefore, we change the latter to $m$. Following this, please see our modification below.

**Change to the manuscript:**

In Fig. 5, in the bottom row, we change "$n(t_{j=4})$" to "$n(t_{j=m})$" and every $n$ in brackets to $m$. See the new figure at the end of this document.

**Comment 40:** *Figure 6, add scale*

**Our answer:** We change Fig. 6 as shown at the end of this document.

**Change to the manuscript:** As stated above.

**Comment 41:** *Figure 7, the geometric mean value (mu) of 2.089 seems to be abnormally small for this distribution, I would expect the geomean nearer 5-6, close to the median?*

**Our response:** As pointed out in our response to Comment 22, $\mu$ does not denote the geometric mean, but the arithmetic mean of $\log(c_N)$.

**Comment 42:** *Figures 8 through 13 to be redrawn using a non-zero concentration for the missing Xa data on 27-28/08/2002*

**Our answer:** Please see the modified versions of Figs. 8-13 at the end of this document.

**Change to the manuscript:** As stated above.

**Comment 43:** *Figure 8: 'Where the error bars are missing one of the three NH3 concentration denuders were malfunctioning or not registering data at all.' This is slightly misleading, visually, because it is at times when fluxes are most uncertain (calculated from only 2*

*concentration heights) that there is no indication of uncertainty on the figure... I would suggest to calculate the mean uncertainty from all fluxes from 3-point gradients (mean of red error bars already present on figure), multiply this value by e.g. a factor of 2, and apply to the rest of the points (in a different color)?*

**Our answer:** We calculate the error bars as suggested and add them to Fig. 8. See the modified figure together with the belonging caption at the end of this document.

**Change to the manuscript:** As stated above.

[revised manuscript text omitted]

---

## Author Comment (AC2) · 5 Jun 2017

**Authors' response to the review of Referee 2 on "Process-based modelling of NH₃ exchange with grazed grasslands"**

We thank the referee for the valuable comments. Our responses and the changes we make to address the referee's comments are provided below point-by-point. The cited literature as well as the modified and the newly-created figures are listed at the end of this document.

**Comment 1:** "*The manuscript is sometimes hard to read and may be shortened; especially the description and results on the sensitivity analysis may be more synthetic.*" (Cited from the section "General comments").

**Our answer:** Apart from Comment 11, the reviewer did not specify the exact parts of the manuscript that should be shortened. However, we are open for the reviewer's additional suggestions.

**Comment 2:** *P5L9: I would suggest telling in a few words what limitations may imply the fact that no water infiltration is taken into account.*

**Our answer:** The GAG model applied and extended to the field scale is described in Móring et al., (2016) as well as in the PhD thesis by Móring (2016). In these studies the possible consequences of this – and the other model assumptions – are investigated in detail. In this part of the manuscript only a general description of the model is provided to set the context for the work described in the following part of the manuscript.

**Change to the manuscript:**

On P5 from L5 we change:

„The GAG model (Móring et al., 2016) is a process-based NH₃ emission model for a single urine patch that is capable of…"

as follows:

"The GAG model, applied and extended to the field scale in this study, is a process-based NH₃ emission model for a single urine patch. An in-depth description of the model, together with a comprehensive sensitivity analysis can be found in Móring et. al (2016) and Móring (2016). The GAG model is capable of…"

**Comment 3:** *P6L5-L6. I suggest writing which parameter is modelled with a negative binomial (area covered by patch?)*

**Our answer:** Please see our modification below.

**Change to the manuscript:**

On P6 we insert text to L5-6 (inserted text in bold):

"A way to estimate the temporal evolution of the urine-covered proportion of the field is to use a negative binomial distribution function **for the time-space distribution of the urine patches** as suggested by…"

**Comment 4:** *P8 EQ5: From the equation I understand that n (over the sum symbol) and n(tj) are not the same. Please clarify.*

**Our answer:** Indeed, this could be confusing for the readers. Please see our modifications below.

**Change to the manuscript:**

In Eq. 5 we change *n* (over Σ) to *m*:

$$F_{net}(t_i) = \frac{F_{non}(t_i)A_{non}(t_i) + \sum_{j=1}^{m} F_{patch}^{j}(t_i)n(t_j)A_{patch}}{A_{field}}$$

In accordance, the size of the matrix considered for the calculation will be m × m, so on P8 in L16 we change "*n × n*" to "*m × m*", as well as we modify Fig. 5 as shown at the end of this document.

**Comment 5:** *P9 EQ6 and L6-8: Since $\chi_{z0}$ is an equilibrium point between the ground and the atmosphere, I do not understand how it could be parameterised. To me it should depend on the flux and the concentration above. Please clarify and explain clearly the assumptions behind the calculation of the fluxes from non-urine patches area and how these are linked to the urine patches area. May be a resistance scheme in a supplementary material would help the understanding: from what I can understand from the current manuscript, the resistance scheme would be as in the GAG patch model of Moring et al. (2016) with an additional "leg" with a resistance $R_{ac} + R_{bg}$ and a potential $\chi_g$, starting from $\chi_{z0}$. Is that correct? This would imply in particular that the horizontal distance between urine patches and non-urine patches is supposed null. Once the hypotheses clearly explicated I would also suggest discussing in the discussion section what implication this would have.*

**Our answer:** As pointed out in the manuscript on P8 from L2: "it was assumed that the total flux over the field is the sum of the emission from the urine affected area (calculated by GAG) and the exchange with the non-urine area (derived by GAG, assuming constant emission potentials, as explained later, in Section 3.1)", which is in accordance with Eq. 5. This means that the $NH_3$ exchange flux is calculated separately for the non-urine area and for every single urine patch, so a different $\chi_{z0}$ is derived separately for the non-urine area and for every single urine patch deposited on the field. For the urine patches GAG_patch is used (as described by Móring et al. 2016) and for the clean area its modified version is applied.

This assumption also means that all the $\chi_{z0}$ values are driven by the compensation points at the given point of the field ($\chi_p, \chi_g, \chi_{sto}$) and the air concentration of $NH_3$, $\chi_a$. The effect of the neighbouring patches or non-urine area via horizontal dispersion on $\chi_{z0}$ in a given point is neglected. As we argue in the manuscript, to account for this effect is not straightforward, and would involve the application of a dispersion model (P8 L7). However, following the suggestion of Reviewer 1, we carried out a sensitivity analysis for $\chi_a$ that could give an approximate picture on how the $NH_3$ exchange would change with a higher $\chi_{air}$ (the effect of the urine patches over the non-urine area via horizontal dispersion) or a lower $\chi_{air}$ (the same effect of the non-urine area over the urine patches).

**Change to the manuscript:**

On P8 in L3-5 we change the following sentence:

"Therefore, it was assumed that the total flux over the field is the sum of the emission from the urine affected area (calculated by GAG) and the exchange with the non-urine area (derived by GAG, assuming constant emission potentials, as explained later, in Section 3.1)."

as follows:

"Therefore, it was assumed that the total flux over the field is the sum of the emission from the urine affected area and the exchange with the non-urine area. Over the urine affected area the GAG model was applied to every single urine patch and for the non-urine area a modified version of the GAG model was used, assuming constant emission potentials, as explained later, in Section 3.1."

On P8, from L27 we change the last sentence of the paragraph to:

"Based on this, $F_{non}$ was derived in the same way as $F_t$, the net $NH_3$ flux over a urine patch in GAG_patch, described by Eq. (1)-(7) in Móring et al. (2016), together with the following simplifications:"

We add the applied resistance models to the supplementary material (see Fig. S1 at the end of this document), and on P9 in L9, we add to the end of the sentence:

"(see the applied resistance model in the Supplementary Material on Fig. S1)"

For the modifications related to the sensitivity analysis for $\chi_a$, please see our response to Comment 18/2 by Reviewer 1.

**Comment 6:** *P10L1-20: The second point "ii)" is unclear. Does that mean that the total amount of liquid will be larger than the soil capacity and since no runaway and infiltration is considered this water will "disappear". Could you rephrase in a clearer way?*

**Our answer:** Point ii) means that in GAG_patch the source layer cannot hold more water than $B_{H2O}(max)$ since for the incoming liquid there is no more soil pore to fill. This means that if urine deposition occurs when $B_{H2O} = B_{H2O}(max)$, there is no infiltration, resulting in no N input to the system. We clarify this in the text.

**Change to the manuscript:**

On P10 from L5, we change ii) as follows:

"may lead to the maximal water content ($B_{H2O}(max)$) in the $NH_3$ source layer. In the formulation of GAG_patch this means that for the incoming liquid there is no more soil pore to fill, i.e. there is no infiltration. Therefore, when a urine patch is deposited while the water content is at $B_{H2O}(max)$, will result in no N input to the system and consequently, no $NH_3$ emission from the soil."

**Comment 7:** *P10L18-20. This sentence is unclear. Please rephrase. In particular I do not understand what "the minimum amount of urine that is always allowed to penetrate" is, and how it is linked with the water budget. I would also suggest justifying why the minimum amount is chosen as 5% and what implication this has.*

**Our answer:** We modify the cited part of the text for better clarity. As for the 5%, it was an arbitrary choice. As explained above, in the original form of the GAG model if the $NH_3$ source layer's water content is at $B_{H2O}(max)$, no urine can infiltrate, and consequently, the model will derive zero urea-driven soil emission. It would be unrealistic to assume that in reality infiltration is prevented to the soil after every rain event (might happen after heavy rain or an elongated rain event), i.e. in most of the cases urine can infiltrate to the soil. However, if urine penetrates to a wet soil, the $NH_3$ emission flux might be weaker for two reasons: 1) due to the soil wetness, the urine might dilute after its deposition, and 2) the high water content is associated with large soil resistance, leading to a weaker $NH_3$ emission flux. Therefore, we think that the choice of 5% of $B_{H2O}(max)$ is large enough to avoid zero soil emission, but small enough to represent the described effects.

On P10 we change extend L18-19 as follows:

"To avoid the possible error resulting from the second point, it was assumed that instead of no infiltration, a small amount of water is always allowed to penetrate to the soil. This amount was chosen to be the 5% of $B_{H2O}(max)$, as shown in Eq. 13. This assumption is necessary since in reality in most of the cases there is infiltration to the soil (except after heavy rain or an elongated rain event), therefore, there is $NH_3$ emission from the soil even if the urine patch deposited to a very wet soil. However, in this case, the $NH_3$ emission flux from the soil might be weaker for two reasons: 1) due to the soil wetness, the urine might dilute after its deposition, leading to a lower $\chi_p$ and 2) the high water content is associated with large soil resistance, leading to a weaker $NH_3$ emission flux. Therefore, the choice of 5% of $B_{H2O}(max)$ could be reasonably large to avoid zero soil emission, but reasonably small to represent the described effects."

**Comment 8:** *P12L29-30: I would have thought that the "unfertilised grassland class" of Massad et al. (2010) would not be adapted here as this grassland does receive nitrogen. Please justify and also discuss the possible implications of choosing a "managed grassland class" in the discussion section.*

**Our answer:** This value of $\Gamma_{sto}$ is used exclusively for the non-urine area (P9 L15-16). The choice of using $\Gamma_{sto}$ values for unfertilised grassland, is in accordance with the assumption we made for the formulation of GAG_field, that is, over the non-urine area there is no considerable nitrogen input (P8 L26). In addition, the sensitivity analysis showed that the total net $NH_3$ exchange is not particularly sensitive to the changes of $\Gamma_{sto}$ applied in the perturbation experiments.

**Change to the manuscript:**

On P12 from L29 we extend the sentence (see the inserted text in bold):

"For the constant $\Gamma_{sto}$ for the non-urine area of the field, **where no considerable N input is assumed**, the values from the emission potential inventory by Massad et al. (2010) for unfertilized grasslands were averaged."

Please note that Table 3 was modified following the suggestion by Reviewer 1. The new table can be found at the end of our response to Reviewer 1. Based on this, on P17 after the last sentence of Section 4.2.1 we add the following piece of text:

"As it can be seen, for $\Gamma_{sto}$ the resulted changes in $\Sigma F_{net}$, depending on the modelling period, are about 5-15% of the perturbations applied to $\Gamma_{sto}$. This means that using a 5 times larger $\Gamma_{sto}$ (+400% perturbation, assuming a soil richer in N) was used in the model

runs, the resulted $\Sigma F_{net}$ were about 20-60% larger, with an overall hourly difference of 0.4 g N."

**Comment 9:** *P16L2: "of the modelled and measured": I suggest adding 'NH₃ exchange' here.*

**Our answer:** Following the suggestion of Reviewer 1 (Comment 26/1), we calculated model statistics and extended the text accordingly. Based on this, please see our modification below.

**Change to the manuscript:**

On P16 in from L2 we change:

"In the case of P2002 (Fig. 8a) the model was in a broad accordance with the observations. It captures…"

as follows:

"In the case of P2002, although the model statistics imply a weak model performance (Fig. N3a), the visual comparison of the modelled and measured $NH_3$ exchange (Fig 8a) suggests a broad accordance between the two datasets. The model captures…"

**Comment 10:** *P18L10: I suggest changing lower and higher to low and high.*

**Our answer:** Please see our modification below.

**Change to the manuscript:**

On P18 in L10 in the first sentence we change "lower" to "low" and "higher" to "high":

"Considering point 1), if pH(*t0*) is **low**, i.e. [H+] is **high**, during urea hydrolysis more $H^+$ ion can be consumed."

**Comment 11:** *P18L16-20 and L21-25: I found these two paragraphs unclear. Could you clarify?*

**Our answer:** Since the reviewer did not point out which parts of the cited two paragraphs need clarification exactly, we modify the text so that it gives more insight on how the buffering effect is taken into account in the GAG model, and we attempt to give our explanations in more details.

**Change to the manuscript:**

On P18 in L5 we change the following sentence:

"Following Móring et al. (2016), the effect of buffering on the $H^+$ ion budget in the $NH_3$ source layer can be expressed with the term $(pH(t_i)-pH(t_{i-1})) \times \beta_{patch}$, where $\beta_{patch} = \beta \times A_{patch} \times \Delta z$."

As follows:

"In Móring et al. (2016), the $H^+$ ion budget depends on the $H^+$ ion consuming and producing processes related to the products of urea breakdown. On top of these, the effect of the buffers in the soil is expressed with an additional term: $(pH(t_i)-pH(t_{i-1})) \times \beta_{patch}$, where $\beta_{patch} = \beta \times A_{patch} \times \Delta z$."

On P18 from L16 we change the paragraph:

"These larger changes in soil pH generate a larger buffering effect (($pH(t_i)$-$pH(t_{i-1})$) × $\beta_{patch}$), i.e. a larger term in the $H^+$ budget, which makes the system more sensitive to a modification of $\beta$ trough $\beta_{patch}$. This was confirmed in the model experiment A (Table 4). In this simulation GAG_patch was run with the initial pH of 4.95 used in the baseline simulation with GAG_field. Although the response of $NH_3$ exchange was relatively weak to the modifications of $\beta$, it was stronger than in the original perturbation experiment for GAG_patch (Table 3)."

as follows:

"These larger changes in soil pH generate a larger buffering effect (($pH(t_i)$-$pH(t_{i-1})$) × $\beta_{patch}$), i.e. a larger term in the $H^+$ budget. This means that in the GAG_field simulations, this term has a stronger effect in the $H^+$ budget, consequently, when $\beta$ is modified (through $\beta_{patch}$), the system gives a stronger response, which means that the model is more sensitive to the perturbation of $\beta$. This was confirmed in the model experiment A (Table 4). In this simulation GAG_patch was run with the initial pH of 4.95 used in the baseline simulation with GAG_field. Although the response of $NH_3$ exchange was relatively weak to the modifications of $\beta$, it was stronger than in the original perturbation experiment for GAG_patch (Table 3)."

On P18 from L21 we change the paragraph:

"Regarding point 2), the definition of $\beta_{patch}$ expresses the buffering effect of the solid material of the soil on the liquid content. Since in the model $\beta_{patch}$ is independent of the liquid content of the soil, within the source layer the same buffering effect takes place even if less urine stored in it. In a smaller amount of urine, the H+ ion budget (expressed in mol $H^+$) and the variations in it are proportionally smaller too. Therefore, the governing role of the same buffering capacity in the case of a smaller amount of urine becomes stronger, resulting in a stronger model sensitivity to $\beta$."

as follows:

"Regarding point 2): the definition of $\beta_{patch}$ expresses the buffering effect of the solid material of the soil on the liquid content. As it can be seen from the formula $\beta_{patch} = \beta \times A_{patch} \times \Delta z$, $\beta_{patch}$ depends clearly on $\Delta z$, but it does not depend on the liquid content of the soil. This means that in the model, in a source layer with the same $\Delta z$, the same buffering effect takes place even if less urine stored in it. In a smaller amount of urine, the $H^+$ ion budget (expressed in mol $H^+$) and the variations in it are proportionally smaller too. Therefore, the governing role of the same buffering capacity in the case of a smaller amount of urine becomes stronger, resulting in a stronger model sensitivity to $\beta$."

**Comment 12:** *Table 5: Explain what is $\beta$ in the table legend.*

**Our answer:** The reviewer must have meant here Table 4. Please see our modification below.

**Change to the manuscript:**

In the legend of Table 4 in the second row after $\beta$ we add: "(buffering capacity)".

**Comment 13:** *Figure 4: I would suggest adding a resistance scheme to better explain the model.*

**Our answer:** Following the suggestion of the reviewer in Comment 5, we add the resistance schemes to the supplementary material. Please see these in Fig. S1, at the end of this document.

**Change to the manuscript:** As stated above.

**Comment 14:** *Figure 8: I suggest adding the input variables of the model here or in a supplementary material ($u_*$, $T_a$, RH, rain, ...) as well as the potentials $\chi(z)$, $\chi_{z0}$, $\chi_g$, $\chi_p$. This will ease the understanding of the flux dynamics.*

**Our answer:** The model input data are the meteorological variables identified for GAG_patch in Móring et. al (2016). The value of $u_*$ is simulated by the model. We create a plot for the meteorological input variables together with the fluxes, and we add these figures to the supplemetary material.

As for the potentials: as pointed out in our answer to Comment 5, in a given time step, $\chi_{z0}$ is different above every urine patch depositied in the different time steps, as well as above the non-urine area (see Comment 5). Similarly, $\chi_p$ varies among the urine patches in a given time step. Therefore, $\chi_{z0}$ and $\chi_p$ cannot be plotted on a single figure, and to create plot only for the non-urine area (for $\chi_{z0}$, and $\chi_g$) would not make much sense on its own. However, we agree that a figure, showing the measured ambient atmospheric $NH_3$ concentration ($\chi_a$), which was an input variable as well, could provide useful information for the readers.

**Change to the manuscript:**

Please see Figure S2 and S3 at the end of this document. We add these figures to the supplementary material. (Please note that in the caption of these figures Fig. 8a and Fig. 8b refer to the improved version of these figures as showed in our response to Reviewer 1.)

On P11 in L13, we add after the last sentence:

"The measured input data is illustrated in the supplementary material, in Fig. S1 and S2."

**References**

Móring, A. 2016. Process-based modelling of ammonia emission from grazing. PhD, University of Edinburgh.

Móring, A., Vieno, M., Doherty, R. M., Laubach, J., Taghizadeh-Toosi, A. & Sutton, M. A. 2016. A process-based model for ammonia emission from urine patches, GAG (Generation of Ammonia from Grazing): description and sensitivity analysis. Biogeosciences, 13, 1837-1861.

| | | NH$_3$ emission from the urine patches | | | | | NH$_3$ exchange with the non-urine area |
|---|---|---|---|---|---|---|---|
| | | Time of the deposition of the urine patches | | | | | |
| | | $t_{j=1}$ | $t_{j=2}$ | $t_{j=3}$ | ... | $t_{j=n}$ | |
| Time since the beginning of the modelling period | $t_{i=1}$ | $F_{patch}^{j=1}(t_{i=1})$ | $F_{non}(t_{i=1})$ | $F_{non}(t_{i=1})$ | ... | $F_{non}(t_{i=1})$ | $F_{non}(t_{i=1})$ |
| | $t_{i=2}$ | $F_{patch}^{j=1}(t_{i=2})$ | $F_{patch}^{j=2}(t_{i=2})$ | $F_{non}(t_{i=2})$ | ... | $F_{non}(t_{i=2})$ | $F_{non}(t_{i=2})$ |
| | $t_{i=3}$ | $F_{patch}^{j=1}(t_{i=3})$ | $F_{patch}^{j=2}(t_{i=3})$ | $F_{patch}^{j=3}(t_{i=3})$ | ... | $F_{non}(t_{i=3})$ | $F_{non}(t_{i=3})$ |
| | ... | ... | ... | ... | ... | ... | ... |
| | $t_{i=m}$ | $F_{patch}^{j=1}(t_{i=m})$ | $F_{patch}^{j=2}(t_{i=m})$ | $F_{patch}^{j=3}(t_{i=m})$ | ... | $F_{patch}^{j=n}(t_{i=m})$ | $F_{non}(t_{i=m})$ |
| Number of patches deposited in $t_j$ | | $n(t_{j=1})$ | $n(t_{j=2})$ | $n(t_{j=3})$ | ... | $n(t_{j=m})$ | 0 |

**Figure 5. Schematic for the temporal development of NH$_3$ fluxes (in every i[th] time step, $t_i$) as derived by GAG_field. $F_{patch}^{j}(t_i)$ stands for the NH$_3$ flux from the urine patches deposited in the j[th] time step ($t_j$), and $F_{non}(t_i)$ stands for the NH$_3$ flux from the non-urine area. The bottom row shows how many urine patches were deposited in the given j[th] time step ($n(t_j)$). Fluxes with striped background are calculated by GAG_patch, and the fluxes with clear background are calculated by a modified version of GAG_patch for non-urine area (explained in the text).**

[Figure]

**Figure S1. Resistance models applied for the simulation of the NH₃ exchange flux a) over the urine patches (as used in GAG_patch in Móring et al., 2016) and b) over the non-urine area (as suggested by Nemitz et al. 2001). The indicated resistances are: the aerodynamic resistance ($R_a$), the quasi-laminar resistance ($R_b$) over the canopy, aerodynamic resistance within the canopy ($R_{ac}$), quasi-laminar resistance at the ground ($R_{bg}$), soil resistance ($R_{soil}$), resistance to water and wax on the leaf surface ($R_w$) and stomatal resistance ($R_{sto}$). The gaseous NH₃ concentrations illustrated are: the ambient air concentration ($\chi_a$), the canopy compensation point ($\chi_{z0}$), the compensation point above the vegetation ($\chi_c$), the compensation point in the model soil pore under a urine patch ($\chi_p$), the stomatal compensation point ($\chi_{sto}$) and the compensation point on the ground in the non-urine area ($\chi_g$). The fluxes shown are: the total net exchange above the given canopy ($F_t$), the emission flux from soil ($F_g$), the exchange flux above the vegetation ($F_f$), the deposition flux to the leaf surface ($F_w$) and the stomatal flux ($F_{sto}$). For the definition of the resistances, fluxes and concentrations on Fig. a) and b, see Móring et al. (2016) and Section 3.1 in the present study, respectively.**

[Figure]

**Figure S2. Measured meteorological variables (relative humidity, soil and air temperature (a), wind speed and global radiation (b), precipitation and surface pressure (c)), the measured ambient atmospheric concentration of NH₃ (d) and the measured and simulated hourly NH₃ fluxes (e) in P2002 in Easter Bush as plotted in Fig. 8a.**

[Figure]

**Figure S3. Measured meteorological variables (relative humidity, soil and air temperature (a), wind speed and global radiation (b), precipitation and surface pressure (c)), the measured ambient atmospheric concentration of NH₃ (d) and the measured and simulated hourly NH₃ fluxes (e) in P2003 in Easter Bush as plotted in Fig. 8b.**

---

## Author Response (AR1)

**Authors' response to the editor's comments on "Process-based modelling of NH₃ exchange with grazed grasslands"**

We thank the editor for the comment. Our response is provided below followed by an updated version of the manuscript, in which we marked all the modifications we carried out.

**Comment:** *"After reviewing the comments made by the referees and your response letters, I find that your manuscript requires major revisions to address the main concerns of the referees. Your responses to the referee comments and suggested changes to the paper are very detailed and complete. Please incorporate these into the revised version of the manuscript."*

> **Our response:** As suggested, we have incorporated all the modifications we indicated in our responses to the referee comments. Please see the modified version of the manuscript together with all these modifications below.

[revised manuscript text omitted]